# Deep Neural Networks and Brain Alignment: Brain Encoding and Decoding (Survey)

**Subba Reddy Oota**                                    *subba.reddy.oota@tu-berlin.de*
*Inria Bordeaux, France*
*LaBRI, Université de Bordeaux, Bordeaux, France*
*Université de Bordeaux, Bordeaux, France*
*TU Berlin, Germany*

**Zijiao Chen**                                         *zijiao.chen@u.nus.edu*
*National University of Singapore, Singapore*

**Manish Gupta**                                        *gmanish@microsoft.com*
*Microsoft, India*
*International Institute of Information Technology Hyderabad, India*

**Bapi Raju Surampudi**                                 *raju.bapi@iiit.ac.in*
*International Institute of Information Technology Hyderabad, India*

**Gael Jobard**                                         *gael.jobard@u-bordeaux.fr*
*GIN, IMN-UMR5293, Université de Bordeaux,*
*CEA, CNRS, Bordeaux, France*

**Frederic Alexandre**                                  *frederic.alexandre@inria.fr*
*Inria Bordeaux, France*
*LaBRI, Université de Bordeaux, Bordeaux, France*
*Université de Bordeaux, Bordeaux, France*

**Xavier Hinaut**                                       *xavier.hinaut@inria.fr*
*Inria Bordeaux, France*
*LaBRI, Université de Bordeaux, Bordeaux, France*
*Université de Bordeaux, Bordeaux, France*

**Reviewed on OpenReview:** *https://openreview.net/forum?id=YxKJihRcby*

## Abstract

Can artificial intelligence unlock the secrets of the human brain? How do the inner mechanisms of deep learning models relate to our neural circuits? Is it possible to enhance AI by tapping into the power of brain recordings? These captivating questions lie at the heart of an emerging field at the intersection of neuroscience and artificial intelligence. Our survey dives into this exciting domain, focusing on human brain recording studies and cutting-edge cognitive neuroscience datasets that capture brain activity during natural language processing, visual perception, and auditory experiences. We explore two fundamental approaches: encoding models, which attempt to generate brain activity patterns from sensory inputs; and decoding models, which aim to reconstruct our thoughts and perceptions from neural signals. These techniques not only promise breakthroughs in neurological diagnostics and brain-computer interfaces but also offer a window into the very nature of cognition. In this survey, we first discuss popular representations of language, vision, and speech stimuli, and present a summary of neuroscience datasets. We then review how the recent advances in deep learning transformed this field, by investigating the popular deep learning based encoding and decoding architectures, noting their benefits and limitations across different

sensory modalities. From text to images, speech to videos, we investigate how these models capture the brain's response to our complex, multimodal world. While our primary focus is on human studies, we also highlight the crucial role of animal models in advancing our understanding of neural mechanisms. Throughout, we mention the ethical implications of these powerful technologies, addressing concerns about privacy and cognitive liberty. We conclude with a summary and discussion of future trends in this rapidly evolving field. Given the large amount of recently published work in the computational cognitive neuroscience (CCN) community, we believe that this survey provides an invaluable entry point for deep neural network (DNN) researchers looking to diversify into CCN research, inviting them to join in unraveling the ultimate puzzle: the human brain.

# 1 Introduction

The central aim of neuroscience is to unravel how the brain represents information and processes it to carry out various tasks (visual, linguistic, auditory, etc.). Two critical paradigms in this field are encoding and decoding. The **encoding model** predicts neural responses from external stimuli, while the **decoding model** reconstructs or classifies stimuli from observed neural activity. Recent advancements in deep neural networks (DNNs) for processing visual, auditory, linguistic, and multimodal stimuli have raised intriguing questions about their potential to elucidate brain function. DNNs may offer a computational framework to capture the unprecedented complexity and richness of brain activities, potentially leading to more accurate encoding and decoding solutions. Building on this, previous surveys have addressed various aspects of brain encoding and decoding studies. For example, Cao et al. (2021) provided a general overview of mechanistic modeling in systems neuroscience, while Karamolegkou et al. (2023) focused on brain encoding and decoding studies for language stimuli. However, recent research in cognitive neuroscience has expanded to modeling for naturalistic and multimodal stimuli using DNNs. By integrating insights from studies involving diverse naturalistic stimuli, we can better understand the full potential of DNNs in modeling complex brain responses. Therefore, this survey aims to fill this gap by systematically summarizing the latest encoding and decoding efforts, focusing on:

1. How DNNs explain underlying information processing in the brain for naturalistic stimuli across various modalities.

2. Potential improvements to DNN models using brain data.

3. Exploration of shared characteristics between artificial and biological neural systems.

**The general context of computational models of the brain function.** The interdisciplinary field of Cognitive Computational Neuroscience (CCN) aims to combine findings from cognitive science, neuroscience, and computational modeling to understand how the brain represents and mediates cognitive processes. Computational models are pitched at three different scales: microscale, mesoscale, and macroscale. Microscale computational models are typically based on detailed biophysical modeling of the neurons and are aimed at deciphering the fundamental principles of neural computation (Dayan et al., 2003). Mesoscale models are designed to link the detailed dynamics of individual neurons and the large-scale organization of brain networks, typically using data from neuroimaging studies of the whole brain (Shine et al., 2021). However, macroscale computational models focus on understanding brain function at a larger systems-level, typically involving networks of brain regions and their interactions. The models discussed in the review belong to the mesoscale category. The cognitive architecture of the brain exhibits an intriguing interplay of structure and function that underpins human cognition, allowing the integration of sensory information (vision, auditory and other senses), perceptual processing, learning, memory, attention, motor processes, decision-making, emotions, and other higher-order cognitive processes. From the panoply of cognitive processes, the models reported here address only a small subset comprising perceptual processes related to vision and auditory domains, as well as behaviors related to reading, listening, speech, and language. Thus the new field of neuro-AI is still in its infancy, modeling the brain function in limited cognitive domains using the latest

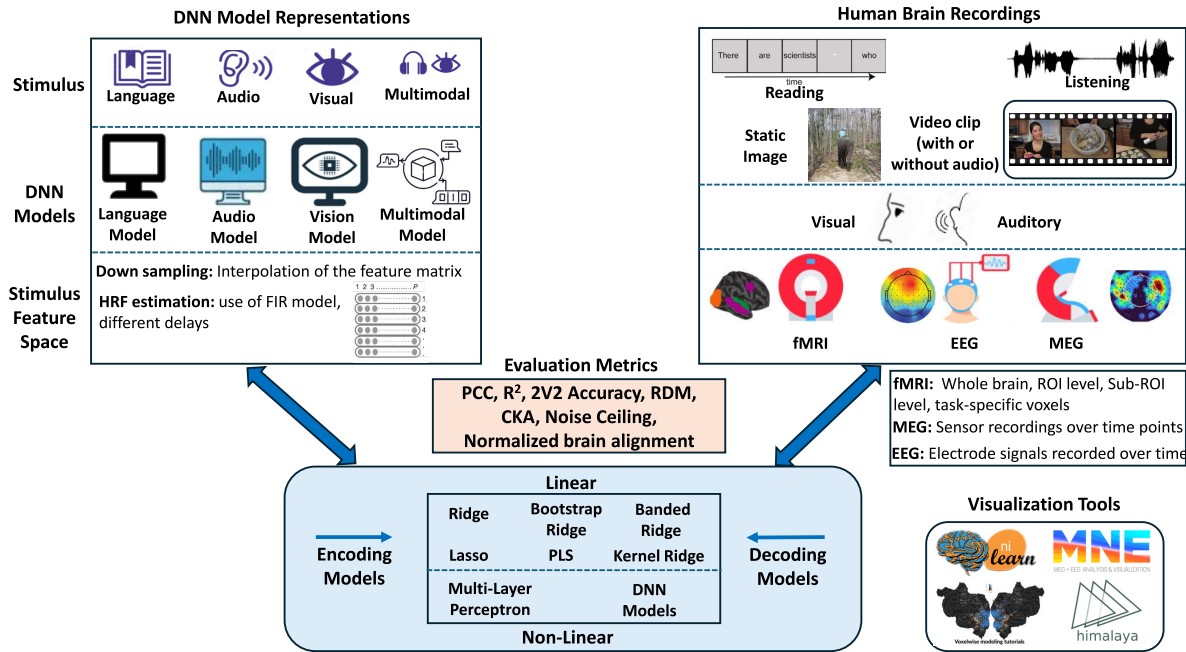

Figure 1: This figure summarizes overall encoding and decoding pipelines with different neuroimaging modalities (fMRI, MEG and EEG), Stimulus modalities (language, audio, visual, and multimodal), and Tasks (reading, listening, watching static images or videos, with or without audio). Further, the pipelines also incorporate stimulus representations obtained from different types of DNN, mapping the DNN and Brain representations via linear or non-linear models, and evaluation measures estimating the performance of encoding/decoding models. Visualization tools facilitate intuitive presentation of the results.

advances in deep learning for understanding how the brain encodes information and how to decode this back!

This survey aims to introduce the challenges in Computational Cognitive Neuroscience (CCN) to AI researchers familiar with recent advances in DNNs. Rather than delving into architectural details and learning procedures for DNNs, we highlight how these advances are applied to CCN problems, providing an entry point for DNN researchers to diversify into CCN research.

Specifically, DNN researchers can benefit from this interdisciplinary approach in several ways:

1. Interpreting DNN models and evaluating their capabilities using naturalistic brain datasets.
2. Using open-source brain datasets as evaluation benchmarks to enhance DNN model capabilities.
3. Incorporating brain recordings in DNN training.
4. Developing advanced brain-computer interfaces (BCIs) for decoding brain patterns.
5. Accessing curated ecological stimuli datasets and a GitHub repository for efficient study initiation [1].
6. Understanding the taxonomy of CCN models and approaches.
7. Exploring open research problems in this rapidly evolving field.

Overall, Figure 1 provides an overview of how we move from human brain recordings to claims about computational principles using neural network models. Specifically, Figure 1 highlights various neuroimaging modalities, stimulus modalities, and tasks, incorporating stimulus representations derived from different types of DNNs. It also shows how DNN and brain representations are mapped through linear or non-linear models, the evaluation measures used to assess the performance of encoding/decoding models, and visualization tools that enable intuitive presentation of the results.

---

[1] https://github.com/subbareddy248/Awesome-Brain-Encoding--Decoding

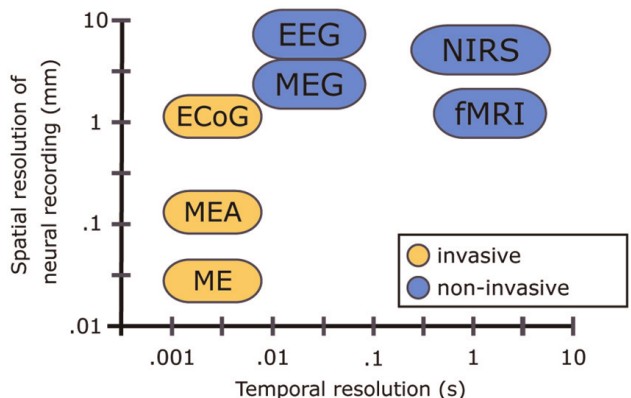

Figure 2: Overview of different brain–machine interfacing methods and their spatial and temporal resolution. Methods included: electroencephalography (EEG), magnetoencephalography (MEG), near-infrared spectroscopy (NIRS), functional magnetic resonance imaging (fMRI), electrocorticography (ECoG), microelectrode array (MEA) recordings and single microelectrode (ME) recordings. *Figure is adapted from (Van Gerven et al., 2009), and used with permission from the respective authors.*

**Brain encoding and decoding.** Figure 1 illustrates the two main paradigms studied in cognitive neuroscience: brain encoding and decoding. Encoding involves learning a mapping $f_e$ from stimulus representations $\mathbf{X}$ to neural brain activation $\mathbf{Y}$, where $\mathbf{X}$ can be achieved through feature engineering or DNNs. Decoding, conversely, involves learning a mapping $f_d$ that predicts stimuli $\mathbf{X}$ from brain activation $\mathbf{Y}$. Often, brain decoding aims to predict a stimulus representation $\mathbf{X}$ rather than reconstructing stimuli $\mathcal{S}$ directly (see Table 1 for notation clarification). The process typically involves learning a semantic representation $\mathbf{X}$ of stimuli $\mathcal{S}$ during training, followed by training regression functions $f_e : \mathbf{X} \rightarrow \mathbf{Y}$ for encoding and $f_d : \mathbf{Y} \rightarrow \mathbf{X}$ for decoding. These functions can then be applied to new stimuli and brain activations during testing. Ridge regression is commonly used for both $f_e$ and $f_d$. To study brain responses to various stimulus modalities, neuroscientists have collected datasets comprising stimuli and corresponding brain activity. Participants interact with these stimuli, often performing tasks such as language comprehension or visual and auditory processing.

Table 1: Notations

| Symbol | Description |
|---|---|
| $\mathcal{S}$ | Set of stimuli used in the experiment. |
| $\mathbf{X}$ | Matrix representing stimulus data. |
| $\mathbf{Y}$ | Matrix representing neural data. |
| $f_e$ | encoding: function mapping from $\mathbf{X} \rightarrow \mathbf{Y}$ |
| $f_d$ | decoding: function mapping from $\mathbf{Y} \rightarrow \mathbf{X}$ |
| $f_e(\mathbf{X})$ | Matrix of predicted neural responses. |
| $f_d(\mathbf{Y})$ | Matrix of predicted stimulus representation. |

**Techniques for recording brain activations.** Popular techniques for recording brain activations can be broadly classified into invasive and non-invasive techniques, as illustrated in Figure 2. Invasive techniques include single Micro-Electrode (ME), Micro-Electrode array (MEA), and Electro-Corticography (ECoG). The non-invasive recording techniques include functional magnetic resonance imaging (fMRI), Magneto-encephalography (MEG), Electro-encephalography (EEG) and Near-Infrared Spectroscopy (NIRS). These techniques differ not only in their invasiveness but also in their spatial and temporal resolution. fMRI offers high spatial resolution but low temporal resolution, making it suitable for identifying which brain regions handle specific functions. A typical whole-brain fMRI scan takes 1-4 seconds, which is considerably slower than the speed of human cognitive processes. In contrast, MEG and EEG provide high temporal but low spatial resolution, capable of preserving rich syntactic information (Hale et al., 2018) but limited in source

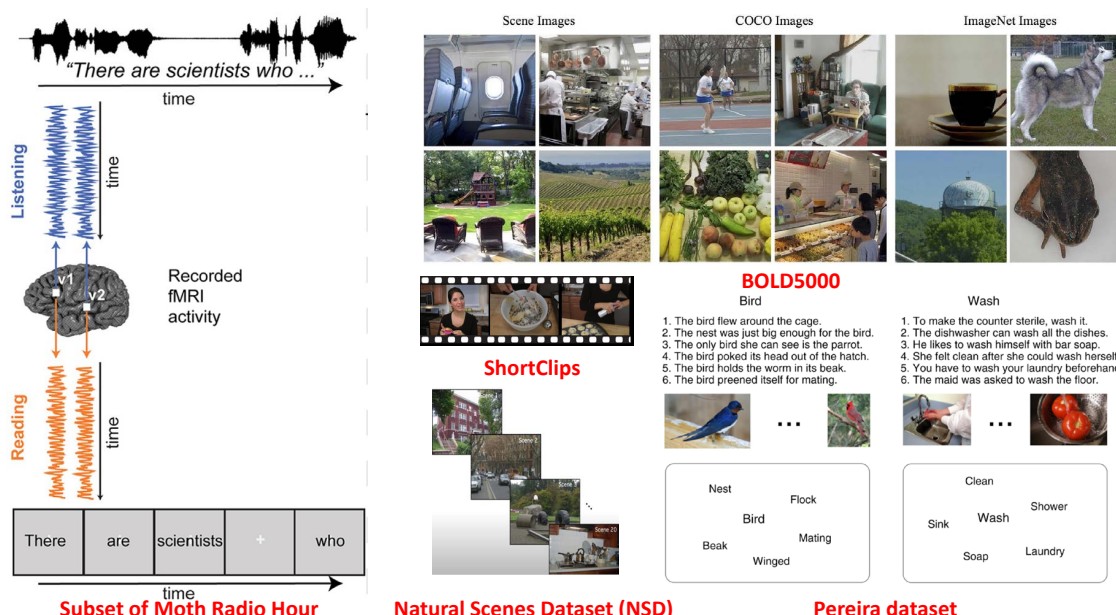

Figure 3: Representative Samples of Naturalistic Brain Datasets. (Left) Comparison of brain activity patterns recorded during reading and listening to the same narrative, illustrating modality-specific and shared neural responses (Deniz et al., 2019). (Right) Examples of diverse naturalistic stimuli used in various public neuroimaging repositories: complex visual scenes from BOLD5000 (Chang et al., 2019), video frames from ShortClips (Huth et al., 2022), natural images from the Natural Scenes Dataset (NSD) (Allen et al., 2022), and multimodal stimuli (text and images) from the Pereira dataset (Pereira et al., 2018). *Adapted with permission from the respective authors.*

analysis. fNIRS offers a compromise, with better temporal resolution than fMRI and better spatial resolution than EEG. However, it is restricted to recording cortical activity and cannot capture signals from deeper brain structures such as the basal ganglia, amygdala, or hippocampus. Further details on the data collection of brain recordings for specific brain regions are discussed in Section 4.

**Stimulus representations.** Neuroscience datasets encompass stimuli across various modalities, including text, visual, audio, video, and other multimodal forms. We briefly discuss the extraction of stimulus representations from DNN models according to the following criteria:

1. Traditional and advanced models for text-based stimulus representations.

2. Image-based representations from deep vision models.

3. Extraction of low-level speech to Transformer-based speech auditory representations.

4. Multimodal stimulus representations, exploring early fusion (combining information across modalities at initial processing stages) and late fusion (combining at the end) deep learning methods.

Further details on different stimulus representation methods are discussed in Section 2.

**Naturalistic neuroscience datasets involving human participants.** Numerous neuroscience datasets have been proposed across modalities (see Figure 3). These datasets vary in terms of: (1) Method for recording activations (fMRI, EEG, MEG, etc.); (2) Repetition time (TR), i.e., the sampling rate; (3) Characteristics of fixation points (location, color, shape); (4) Form of stimuli presentation (text, video, audio, images, or multimodal); (5) Task (question answering, property generation, rating quality, etc.) performed by participants during recording sessions; (6) Time allocated for task completion, e.g., 1 minute to list given

object properties; (7) Participant demographics, e.g. males or females, sighted or blind; (8) Number of repetitions for stimuli response recording; (9) Natural language associated with the stimuli. We discuss details of these datasets in Section 3.

**From Non-Human Models to Human Brain-Machine Alignment.** While this review focuses on brain-machine alignment using human neuroimaging datasets, it is important to acknowledge the substantial contributions made by research involving non-human subjects. Neuro-AI and cognitive-AI methods have made significant strides in modeling the primate visual system and hippocampal navigation system by leveraging advanced machine learning and deep learning techniques to mimic neural processes. Since high-resolution data related to hierarchically organized maps of stimulus properties such as orientation, spatial frequency, and color, are not available for human visual system, measurements from the primate (macaque V1) data have been used to design and validate deep learning models of visual function (Margalit et al., 2024). Several non-human primate datasets, like the primate microelectrode recording dataset proposed in Majaj et al. (2015), are highly valuable and widely used in vision research. The other area where tremendous progress has been made is in the hippocampal memory formation and spatial navigation using rodent electrophysiology data. These efforts led to newer understanding of how place cells and grid cells in the hippocampus and entorhinal cortex encode spatial information and brain's ability to build cognitive maps for spatial navigation (Bellmund et al., 2018). The emerging insights of brain encoding and decoding enabled successful design and testing of brain-machine interfaces for primates (Lebedev & Nicolelis, 2017). Successful application of deep neural networks in mimicking primate perceptual functions have led to formulation of newer frameworks for systems neuroscience research (Richards et al., 2019). In contrast, this review focuses exclusively on neuroscience datasets involving human participants, particularly those with non-invasive recordings such as fMRI, MEG, and EEG. Given the scope of our current review paper, our focus is limited to human brain datasets with an emphasis on the language function that is difficult, if impossible, to investigate in rodents and non-human primates.

**Evaluation of brain encoding and decoding methods.** For brain encoding models, 2-versus-2 (2V2) accuracy, Pearson Correlation and $R^2$ score are commonly used evaluation metrics. Brain decoding models are typically evaluated using metrics such as pairwise accuracy, rank accuracy, $R^2$ score, and mean squared error. Additionally, representational similarity metrics frequently used in brain encoding and decoding studies include representational dissimilarity matrices (RDM), centered kernel alignment (CKA), and normalized distribution similarity (NDS). Detailed definitions of these metrics are provided in Section 5.

**Interpreting brain recordings through DNN model representations.** To interpret stimulus representations obtained from DNN models and examine their impact on brain alignment, prior studies have proposed several methods: variance partitioning (de Heer et al., 2017), the residual approach (Toneva et al., 2022a; Oota et al., 2024b), an indirect approach (Schrimpf et al., 2021; Goldstein et al., 2022), and the stacked regression approach (Lin et al., 2023). We discuss these methods in detail in Section 6.3.

**Computational Cognitive Neuroscience (CCN) research goals.** CCN researchers have primarily focused on two main areas (Doerig et al., 2023):

1. Improving predictive accuracy, addressing questions such as:
   - Which feature set best reflects the neural representational space?
   - Does neural data Y contain information about features X?
   - How can we build accurate models of brain data to enable neuroscience experiment simulation?

2. Enhancing interpretability, exploring questions like:
   - Which features contribute most to neural activity?
   - How do representational spaces correspond, for example, between CNNs and the ventral visual stream, or among text representations?
   - Do features X, generated by a known process, accurately describe the space of neural responses Y?
   - Do voxels respond to single features or exhibit mixed selectivity?
   - How does the mapping relate to other brain function models or theories?

Overall, the CCN community tries to understand the brain in computational terms which also involves falsifying existing models, and making new (model-guided) predictions (Tuckute et al., 2024). We discuss some of these questions in Sections 6 and 7.

Brain encoding literature (Mitchell et al., 2008; Wehbe et al., 2014; Huth et al., 2016) has focused on studying several important aspects: (1) Which models lead to better predictive accuracy across modalities? (Toneva & Wehbe, 2019; Deniz et al., 2019; Schrimpf et al., 2021) (2) How can we disentangle the contributions of syntax and semantics from language model representations to the alignment between brain recordings and language models? (Lopopolo et al., 2017; Reddy & Wehbe, 2021) (3) Why do some representations lead to better brain predictions? How are deep learning models and brains aligned in terms of their information processing pipelines? (Merlin & Toneva, 2022; Aw & Toneva, 2023) (4) Does joint encoding of task and stimulus representation help? (Oota et al., 2024b). We discuss these details of encoding methods in Section 6.

Brain decoding models aim to understand what a subject is thinking, seeing, and perceiving by analyzing neural recordings. Over the past decades, the brain-computer interface (BCI) has made significant progress in decoding stimuli (language/images/speech) from the brain using non-invasive recordings. Like brain encoding literature, decoding literature focuses on studying a few important aspects: (1) In the context of language, how do we compose the linguistic meaning from different stimuli such as text, images, videos, or speech by analyzing the evoked brain activity (Pereira et al., 2016; 2018). (2) Given brain activations corresponding to visual stimuli, how accurately can we decode a sentence representing the visual stimuli? (Nishimoto et al., 2011; Beliy et al., 2019) (3) How can we decode natural speech processing from non-invasive brain recordings using a single architecture and a data-driven approach? (Défossez et al., 2023) (4) How accurately can we reconstruct perceived natural images or decode their semantic contents from non-invasive recording data using popular deep learning models? (Takagi & Nishimoto, 2022). We discuss these details of decoding methods in Section 7.

## 2 Stimulus Representations

This section discusses various stimulus representations proposed in the literature across different modalities: text, visual, audio, video, and multimodal stimuli.

**Text stimulus representations.** The evolution of text-based stimulus representations reflects the rapid advancements in natural language processing. Early methods relied on text corpus co-occurrence counts (Mitchell et al., 2008; Pereira et al., 2013; Huth et al., 2016), topic models (Pereira et al., 2013), and syntactic and discourse features (Wehbe et al., 2014). In recent years, both semantic models and experiential attribute models have gained prominence. Semantic representation models encompass a wide range of techniques, from word embeddings (Pereira et al., 2018; Wang et al., 2020; Pereira et al., 2016; Toneva & Wehbe, 2019; Anderson et al., 2017a; Oota et al., 2018) to more sophisticated sentence representation models (Sun et al., 2020; 2019; Toneva & Wehbe, 2019). The field has seen a shift from static word embeddings to contextualized representations, with models like RNNs (Jain & Huth, 2018; Oota et al., 2019) and Transformer-based methods (Gauthier & Levy, 2019; Toneva & Wehbe, 2019; Schwartz et al., 2019; Schrimpf et al., 2021; Antonello et al., 2021; Oota et al., 2022b; Aw & Toneva, 2023) gaining traction. Experiential attribute models offer a different perspective, representing words based on human ratings of their associations with various experiential attributes (Anderson et al., 2019; 2020; Berezutskaya et al., 2020; Just et al., 2010; Anderson et al., 2017b). These models provide insights into how humans conceptualize and experience language.

Recent brain encoding research has increasingly focused on contextualized word representations, examining how the amount of context affects brain predictivity (Jain & Huth, 2018; Toneva & Wehbe, 2019). Figure 4 (a) illustrates the construction of past/future context for these representations, demonstrating how researchers consider words preceding and succeeding the current word to create rich, context-aware embeddings.

In practice, the application of contextualized word representations often involves a constrained context length. This approach balances computational efficiency with the need for rich contextual information. For example, when processing a narrative of $M$ words with a context window of 20, the representation for each word is computed using up to 20 preceding words. Specifically, the vector for the third word would be derived

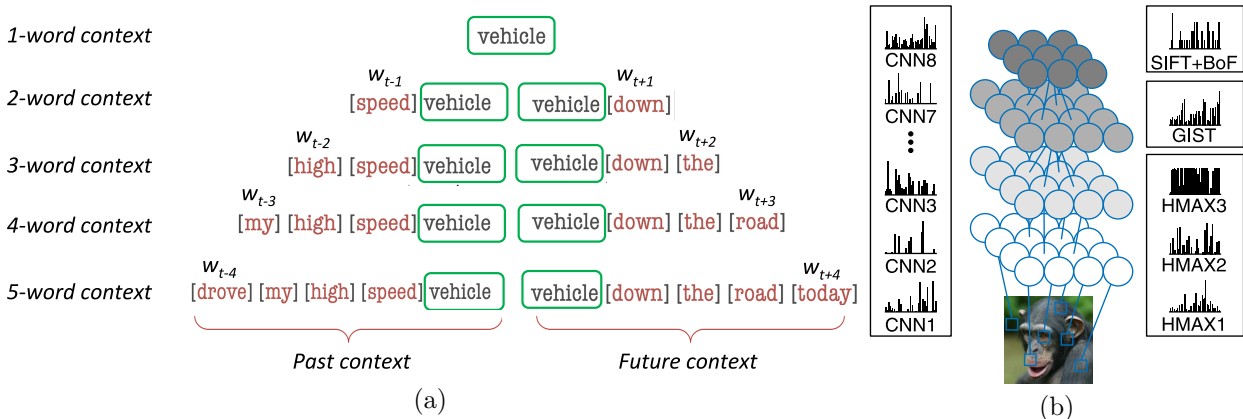

Figure 4: *(a)* Context Representation of Words in Language Models. This figure illustrates how past and future context is constructed for different word orders. Using the word "vehicle" as an example, we demonstrate how preceding words (past context) and succeeding words (future context) are considered for various context lengths. *(b)* Extraction of Image Representations for Brain-Computer Interface Models. This figure illustrates the process of extracting layer-wise image representations from Convolutional Neural Network (CNN) models. These representations have been extensively studied in prior research (Yamins et al., 2014; Horikawa & Kamitani, 2017) for their effectiveness in both brain encoding (predicting neural responses from visual stimuli) and decoding (reconstructing visual stimuli from brain activity) models. The figure demonstrates how different layers of a CNN, from early layers capturing low-level features to deeper layers representing more abstract concepts, can be utilized to understand and predict brain responses to visual stimuli. *Figure is adapted from (Horikawa & Kamitani, 2017), and used with permission from the respective authors.*

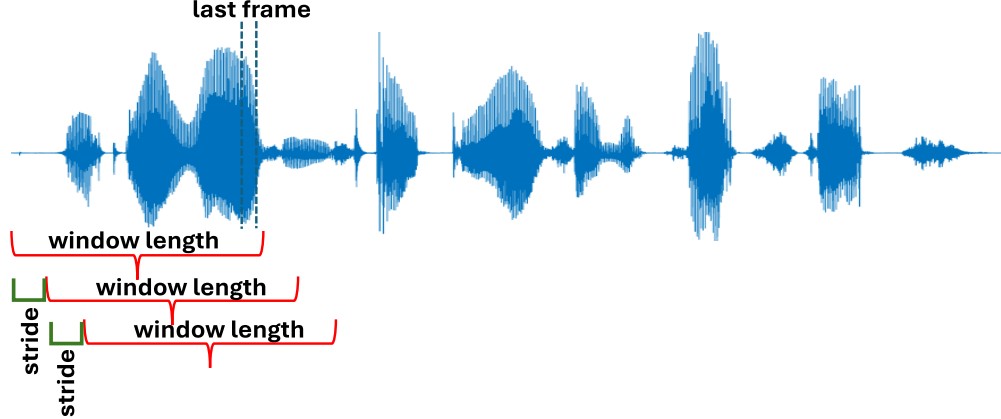

Figure 5: *Extraction of contextualized speech representations:* Representation of the last frame within each window allows for the capture of temporal dynamics and contextual nuances in the speech signal. The length of the time window is typically varied from 16 to 64 secs, with strides ranging from 10 to 100 milliseconds.

from the input sequence $(w_1, w_2, w_3)$, while the vector for the final word, $w_M$, would be based on the input $(w_{M-20}, \ldots, w_M)$. This sliding window approach allows the model to capture local context effectively while maintaining a manageable computational load, even for longer texts. Such methods enable researchers to investigate how varying amounts of context influence the model's ability to predict brain activity, providing insights into the temporal aspects of language processing in the brain.

**Visual stimulus representations.** The field of visual stimulus representation has similarly progressed from simpler methods to more complex, neural network-based approaches. Early techniques employed visual field filter banks (Thirion et al., 2006; Nishimoto et al., 2011) and Gabor wavelet pyramids (Kay et al., 2008;

Naselaris et al., 2009). These methods provided a foundation for understanding how the brain processes visual information.

More recent approaches leverage the power of Convolutional Neural Networks (CNNs) (Du et al., 2020; Beliy et al., 2019; Anderson et al., 2017a; Yamins et al., 2014; Nishida et al., 2020) and concept recognition models (Anderson et al., 2020). CNNs, in particular, have revolutionized visual stimulus representation by automatically learning hierarchical features from raw image data. Figure 4 (b) illustrates how these CNN models are utilized for visual stimulus representation, showcasing their ability to capture increasingly abstract visual concepts across layers.

**Audio stimuli representations.** Audio stimulus representations have evolved from basic acoustic features to sophisticated deep learning models. Initial approaches focused on phoneme-level features (Huth et al., 2016) and low-level speech characteristics such as filter banks, Mel spectrograms, and MFCCs (Deniz et al., 2019). These methods provided a solid foundation for understanding speech processing in the brain.

Recent advancements have seen the integration of deep learning models specifically designed for audio processing. For instance, Nishida et al. (2020) employed features from SoundNet, a deep neural network trained on large-scale audiovisual data. The field has further progressed with the adoption of Transformer-based speech models like Wav2Vec2.0, HuBERT, and Whisper (Vaidya et al., 2022; Antonello et al., 2024b; Oota et al., 2024a). These models offer rich, contextual representations of speech, capturing nuances that were previously difficult to model. Figure 5 demonstrates how researchers extract representations from these models using various time window lengths and strides, allowing for a more comprehensive analysis of temporal dynamics in speech processing.

**Multimodal stimulus representations.** The increasing focus on naturalistic stimuli has led to significant developments in multimodal representations. These approaches aim to jointly model information from multiple sensory inputs, mirroring the integrative nature of human perception. Recent studies have explored various combinations of modalities, each offering unique insights into how the brain processes complex, real-world stimuli.

For video processing, researchers have combined audio and image representations, such as using VGG for visual features alongside SoundNet for audio features (Nishida et al., 2020). This approach allows for a more comprehensive understanding of how the brain integrates audiovisual information. In the realm of image and text combinations, models like GloVe+VGG and ELMo+VGG (Wang et al., 2020) have been employed, leveraging the strengths of both visual and linguistic representations.

The latest advancements in multimodal representations include sophisticated models that are pretrained on large-scale multimodal datasets. These include Contrastive Language-Image Pretraining (CLIP) (Radford et al., 2021), Learning Cross-Modality Encoder Representations from Transformers (LXMERT) (Tan & Bansal, 2019), and VisualBERT (Li et al., 2020; Oota et al., 2022e). These models offer powerful, joint representations of visual and linguistic information, potentially providing deeper insights into how the brain integrates information across modalities.

## 3 Naturalistic Neuroscience Datasets

In this section, we discuss popular neuroscience datasets that involve audio, visual, and other multimodal stimuli that have been utilized in the literature. Tables 2, 3 and 4 provide detailed overview of types of brain recording, languages used, stimuli presented, number of subjects (|S|), and tasks across datasets of different modalities. The audio datasets consist of listening to narrative stories or spoken utterances, while the vision-based datasets are categorized into reading stories and watching static images or silent videos. The multimodal stimulus datasets include the presence of visual input along with other modalities, such as audio. Figure 3 illustrates examples from selected datasets.

### 3.1 Audio datasets

Most proposed audio datasets are in English (Huth et al., 2016; Brennan & Hale, 2019; Anderson et al., 2020; Nastase et al., 2020), with one in Italian (Handjaras et al., 2016) and another in Chinese and French (Li et al.,

Table 2: Naturalistic Neuroscience Audio Datasets. Publicly available datasets are linked to their sources in the Dataset column. In this table, |S| represents the number of participants in each dataset.

| | Dataset | Authors | Type | Lang. | Stimulus | \|S\| | Task |
|---|---|---|---|---|---|---|---|
| Audio | - | (Handjaras et al., 2016) | fMRI | Italian | Verbal, pictorial or auditory presentation of 40 concrete nouns, 4 times | 20 | Property generation |
| | Blank2014 | (Blank et al., 2014) | fMRI | English | Listening naturalistic stories | 10 | Passive listening |
| | The Moth Radio Hour | (Huth et al., 2016) | fMRI | English | Listening eleven 10-minute stories | 7 | Passive listening |
| | Narrative Brain Dataset | (Lopopolo et al., 2018) | fMRI | Dutch | Spoken presentation of short excerpts of three stories | 24 | Passive listening |
| | Alice | (Brennan & Hale, 2019) | EEG | English | Listening Chapter one of Alice's Adventures in Wonderland (2,129 words in 84 sentences) as read by Kristen McQuillan | 33 | Question answering |
| | - | (Anderson et al., 2020) | fMRI | English | Listening one of 20 scenario names, 5 times | 26 | Imagine personal experiences |
| | Narratives | (Nastase et al., 2020) | fMRI | English | Listening 27 diverse naturalistic spoken stories. 891 functional scans | 345 | Passive listening |
| | Natural Stories | (Zhang et al., 2020) | fMRI | English | Listening Moth-Radio-Hour naturalistic spoken stories. | 19 | Passive listening |
| | The Little Prince | (Li et al., 2021) | fMRI | English, Chinese, French | Listening audio book for about 100 minutes. | 112 | Passive listening |
| | MEG-MASC | (Gwilliams et al., 2023) | MEG | English | Listening two hours of naturalistic stories. 208 MEG sensors | 27 | Passive listening |
| | Music Genre | (Nakai et al., 2022) | fMRI | English | Listening 540 music pieces from 10 music genres | 5 | Passive listening |
| | SMN4Lang | (Wang et al., 2022) | fMRI, MEG | Chinese | Listening 6 hours of naturalistic stories | 12 | Passive listening |
| | Martin2023 | (Martin, 2023) | MEG | Dutch, French | Listening 70 mins of naturalistic stories | 25 | Passive listening |
| | Giovanni2023 | (Liberto et al., 2023) | EEG | English | Listening an audio book: "The Old Man and the Sea" | 19 | Passive listening |
| | Little Prince Hong Kong | (Momenian et al., 2024) | fMRI, EEG | Cantonese | Listening audiobook for about 20 minutes. | 52 | Passive listening |
| | SCNU-Mandarin-Cantonese | (Yuan & Yang, 2024) | fMRI | Mandarin, Cantonese | Listening three narratives | 27 | Passive listening |

2021). Participants engaged in various tasks while their brain activations were measured, including property generation (Handjaras et al., 2016), passive listening (Huth et al., 2016; Nastase et al., 2020), question answering (Brennan & Hale, 2019), and imagining themselves personally experiencing common scenarios (Anderson et al., 2020). In the latter, participants underwent fMRI as they reimagined scenarios (e.g., resting, reading, writing, bathing) when prompted by standardized cues. The Narratives dataset (Nastase et al., 2020) used 27 different stories as stimuli, totaling 6.4 days worth of recordings across subjects.

## 3.2 Vision datasets

**Text datasets: Reading.** These datasets are created by presenting words, sentences, passages, or chapters as stimuli. Examples include the Harry Potter Story dataset (Wehbe et al., 2014), ZuCo EEG dataset (Hollenstein et al., 2018), and datasets proposed in (Handjaras et al., 2016; Anderson et al., 2017a; 2019; Wehbe et al., 2014). In (Handjaras et al., 2016), participants were asked to verbally enumerate, within one minute, the properties (features) describing the entities the words refer to. The study involved four groups of participants: 5 sighted individuals presented with pictorial forms of nouns, 5 sighted individuals with written Italian words, 5 sighted individuals with spoken Italian words, and 5 congenitally blind individuals with spoken Italian words. The dataset proposed by Anderson et al. (2017a) contains 70 Italian words from seven taxonomic categories (abstract, attribute, communication, event/action, person/social role, location, object/tool) in the law and music domains, including both concrete and abstract words. The Pereira dataset includes pictures, sentences, and word clouds, for which fMRI recordings were obtained during three viewing conditions (Pereira et al., 2018). The ZuCo dataset (Hollenstein et al., 2018) contains sentences for which EEG recordings were obtained during three tasks: normal reading of movie reviews, normal reading of Wikipedia sentences, and task-specific reading of Wikipedia sentences. For this dataset, sentences were presented to subjects in a naturalistic reading scenario, with each complete sentence displayed on the screen.

Table 3: Naturalistic Neuroscience Vision-based Datasets (Reading text, Watching images and videos (without audio). Publicly available datasets are linked to their sources in the Dataset column. In this table, |S| represents the number of participants in each dataset.

| | Dataset | Authors | Type | Lang. | Stimulus | |S| | Task |
|---|---|---|---|---|---|---|---|
| **Text** | Harry Potter | (Wehbe et al., 2014) | fMRI, MEG | English | Reading Chapter 9 of Harry Potter and the Sorcerer's Stone | 9 | Story understanding |
| | Fedorenko2016 | (Fedorenko et al., 2016) | ECoG | English | 8-word-long sentences | 5 | Passive reading |
| | - | (Handjaras et al., 2016) | fMRI | Italian | Verbal, pictorial or auditory presentation of 40 concrete nouns, four times | 20 | Property generation |
| | - | (Anderson et al., 2017a) | fMRI | Italian | Reading 70 concrete and abstract nouns from law/music, five times | 7 | Imagine a situation with noun |
| | ZuCo | (Hollenstein et al., 2018) | EEG | English | Reading 1107 sentences with 21,629 words from movie reviews | 12 | Rate movie quality |
| | Pereira | (Pereira et al., 2018) | fMRI | English | Viewing 180 Words with Picture, Sentences, word clouds; reading 96 text passages; 72 passages. 3 times repeated. | 16 | Passive reading |
| | - | (Anderson et al., 2019) | fMRI | English | Reading 240 active voice sentences describing everyday situations | 14 | Passive reading |
| | BCCWJ-EEG | (Oseki & Asahara, 2020) | EEG | Japanese | Reading 20 newspaper articles for ∼30-40 minutes | 40 | Passive reading |
| | Subset Moth Radio Hour | (Deniz et al., 2019) | fMRI | English | Reading 11 stories | 9 | Passive reading and listening |
| | Greta2024 | (Tuckute et al., 2024) | fMRI | English | Reading 1,000 sentences | 9 | Passive reading |
| **Static Images** | Inverse retinotopy | (Thirion et al., 2006) | fMRI | - | Viewing rotating wedges (8 times), expanding/contracting rings (8 times), rotating 36 Gabor filters (4 times), grid (36 times) | 9 | Passive viewing |
| | Vim-1 | (Kay et al., 2008) | fMRI | - | Viewing sequences of 1870 natural photos | 2 | Passive viewing |
| | Generic Object Decoder | (Horikawa & Kamitani, 2017) | fMRI | - | Viewing 1,200 images from 150 object categories; 50 images from 50 object categories; imagery 10 times | 5 | Repetition detection |
| | BOLD5000 | (Chang et al., 2019) | fMRI | - | Viewing 5254 images depicting real-world scenes | 4 | Passive viewing |
| | Algonauts | (Cichy et al., 2019) | fMRI, MEG | - | Viewing 92 silhouette object images and 118 images of objects on natural background | 15 | Passive viewing |
| | NSD | (Allen et al., 2022) | fMRI | - | Viewing 73000 natural scenes | 8 | Passive viewing |
| | THINGS | (Hebart et al., 2023) | fMRI, MEG | - | Viewing 31188 natural images | 8 | Oddball Detection |
| | Gifford2022 EEG | (Gifford et al., 2022) | EEG | - | Viewing 82160 natural images | 8 | Passive viewing |
| | Infants EEG | (Quek et al., 2024) | EEG | - | Viewing 200 natural objects | 42 | Passive viewing |
| | NOD | (Gong et al., 2023) | fMRI | - | Viewing 57,120 natural images | 30 | Passive viewing |
| **Videos (without audio)** | BBC's Doctor Who | (Seeliger et al., 2019) | fMRI | English | Viewing spatiotemporal visual and auditory videos (30 episodes). 120.8 whole-brain volumes (∼23 h) of single-presentation data, and 1.2 volumes (11 min) of repeated narrative short episodes. 22 repetitions | 1 | Passive viewing |
| | Japanese Ads | (Nishida et al., 2020) | fMRI | Japanese | Viewing 368 web and 2452 TV Japanese ad movies (15-30s). 7200 train and 1200 test fMRIs for web; fMRIs from 420 ads. | 52 | Passive viewing |
| | Pippi Langkous | (Berezutskaya et al., 2020) | ECoG | Swedish, Dutch | Viewing 30 s excerpts of a feature film (in total, 6.5 min long), edited together for a coherent story | 37 | Passive viewing |
| | Algonauts | (Cichy et al., 2021) | fMRI | English | Viewing 1000 short video clips (3 sec each) | 10 | Passive viewing |
| | Natural Short Clips | (Huth et al., 2022) | fMRI | English | Watching natural short movie clips | 5 | Passive viewing |
| | Natural Short Clips | (Lahner et al., 2023) | fMRI | English | Watching 1102 natural short video clips | 10 | Passive viewing |
| | NNDb | (Aliko et al., 2020) | fMRI | English | Watching 10 full-length movies | 84 | Passive viewing |
| | NATVIEW_EEGFMRI | (Telesford et al., 2023) | fMRI, EEG | English | Watching 5 short-length movies | 22 | Passive viewing |
| | Mind captioning | (Horikawa, 2024) | fMRI | English | Watching total of 2,196 videos | 6 | Passive viewing |

Table 4: Naturalistic Neuroscience Multimodal Datasets. Publicly available datasets are linked to their sources in the Dataset column. In this table, |S| represents the number of participants in each dataset.

| | Dataset | Authors | Type | Lang. | Stimulus | \|S\| | Task |
|---|---|---|---|---|---|---|---|
| Multimodal | 60 Concrete Nouns | (Mitchell et al., 2008) | fMRI | English | Viewing 60 different word-picture pairs from 12 categories, 6 times each | 9 | Passive viewing |
| | - | (Sudre et al., 2012) | MEG | English | Reading 60 concrete nouns along with line drawings. 20 questions per noun lead to 1200 examples. | 9 | Question answering |
| | - | (Zinszer et al., 2018) | fNIRS | English | 8 concrete nouns (audiovisual word and picture stimuli): bunny, bear, kitty, dog, mouth, foot, hand, and nose; 12 times repeated. | 24 | Passive viewing and listening |
| | - | (Cao et al., 2021) | fNIRS | Chinese | Viewing and listening 50 concrete nouns from 10 semantic categories. | 7 | Passive viewing and listening |
| | Neuromod | (Boyle et al., 2020) | fMRI | English | Watching TV series and movies (Friends, Movie10) | 6 | Passive viewing and listening |
| | Multimodal fMRI | (Jung et al., 2024) | fMRI | English | Watching movies, dynamic faces task | 101 | Passive viewing and listening |
| | fMRI Narrative movie | (Yamaguchi et al., 2024) | fMRI | English | Watching nine narrative movies | 6 | Passive viewing and listening |
| | Game of Thrones | (Watson & Andrews, 2024) | fMRI | English | Watching short audiovisual clips | 73 | Passive viewing and listening |
| | AVID | (Sava-Segal et al., 2023) | fMRI | English | Watching short audiovisual clips | 48 | Passive viewing and listening |

Subjects read each sentence at their own pace, determining how long to fixate on each word and which word to fixate on next.

**Static Image datasets.** Earlier visual datasets were based on binary visual patterns (Thirion et al., 2006). Recent datasets contain natural images, including Vim-1 (Kay et al., 2008), BOLD5000 (Chang et al., 2019), Algonauts (Cichy et al., 2019), NSD (Allen et al., 2022), Things-data (Hebart et al., 2023), NOD (Gong et al., 2023), and the dataset proposed in (Horikawa & Kamitani, 2017). BOLD5000 includes approximately 20 hours of MRI scans per participant for four participants, using 4,916 unique images as stimuli from three image sources. The Algonauts dataset contains two sets of training data, each consisting of an image set and brain activity in RDM format (for fMRI and MEG). Training set 1 has 92 silhouette object images, while training set 2 has 118 object images with natural backgrounds. The testing data consists of 78 images of objects on natural backgrounds. Most visual datasets involve passive viewing, but the dataset in (Horikawa & Kamitani, 2017) involved participants performing a one-back repetition detection task.

**Video datasets.** Recently, video neuroscience datasets have also been proposed. These include BBC's Doctor Who (Seeliger et al., 2019), Japanese Ads (Nishida et al., 2020), Pippi Langkous (Anderson et al., 2020), and Algonauts (Cichy et al., 2021). The Japanese Ads dataset contains data for two sets of movies provided by NTT DATA Corp: web and TV ads. It also includes four types of cognitive labels associated with the movie datasets: scene descriptions, impression ratings, ad effectiveness indices, and ad preference votes. The Algonauts 2021 dataset contains fMRIs from 10 human subjects who watched over 1,000 short (3-second) video clips.

### 3.3 Multimodal datasets.

Beyond audio and vision datasets, other multimodal datasets have been proposed. These include audiovisual datasets (Zinszer et al., 2018; Cao et al., 2021; Boyle et al., 2020; Jung et al., 2024), and words associated with line drawings (Mitchell et al., 2008; Sudre et al., 2012). These datasets have been collected using various methods such as fMRI (Mitchell et al., 2008; Pereira et al., 2018), MEG (Sudre et al., 2012), and fNIRS (Zinszer et al., 2018; Cao et al., 2021). Specifically, in Sudre et al. (2012), subjects were asked to perform a question answering (QA) task while their brain activity was recorded using MEG. Subjects were first presented with a question (e.g., "Is it manmade?"), followed by 60 concrete nouns along with their line drawings, in a random order. For all other datasets, subjects performed passive viewing and/or listening tasks. The recent multimodal brain datasets, Neuromod (Boyle et al., 2020) and Multimodal fMRI (Jung

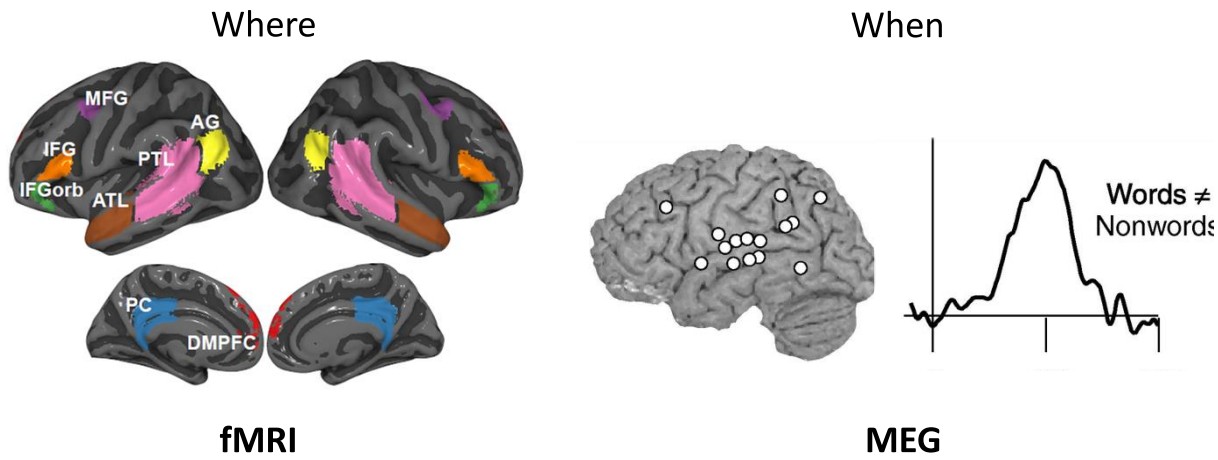

Figure 6: Non-invasive Brain Recording Techniques: fMRI and MEG. The fMRI plot illustrates the language network, highlighting the brain regions involved in language processing. In contrast, the MEG plot demonstrates the timing of peak responses to words, emphasizing the temporal aspects of neural activity. *Figure is adapted from (Toneva et al., 2022a), used with permission from the respective authors.*

et al., 2024), involve experiments with multimodal naturalistic stimuli, where participants engage in watching videos accompanied by audio.

## 4   Brain Regions

This section discusses the mapping of brain recordings to stimulus-specific brain regions as presented in the literature. We focus on the regions of the language network, auditory cortex, and visual cortex, along with their sub-regions. To use brain recordings from preprocessed naturalistic neuroscience datasets, follow these steps: (i) Use brain activation of voxels directly if either of the next two steps is not applicable. (ii) Apply a brain mask to the brain volume to obtain the activation of voxels. or (iii) Project the brain volume onto the surface space (such as "fsaverage5", "fsaverage6", or "fsaverage"). To visualize the brain maps, popular libraries such as Nilearn [2] or Pycortex [3] are useful for fMRI recordings, while MNE-Python [4] is suitable for both MEG and EEG datasets.

**Language network.** The language network refers to brain regions involved in language processing. Studies such as those by Fedorenko & Thompson-Schill (2014); Lipkin et al. (2022) use contrasts like words versus non-words to functionally select language-specific voxels, highlighting the variability of anatomical regions across individuals. As a result, the language network is typically defined functionally rather than anatomically. Based on the Fedorenko lab's language parcels (Fedorenko et al., 2010; Fedorenko & Thompson-Schill, 2014), eight language-relevant regions encompass broader language areas: angular gyrus (AG), anterior temporal lobe (ATL), posterior temporal lobe (PTL), inferior frontal gyrus (IFG), inferior frontal gyrus orbital (IFGOrb), middle frontal gyrus (MFG), posterior cingulate cortex (PCC), and dorsal medial prefrontal cortex (dmPFC), as shown in Figure 6 (left). These eight language networks are used in several recent studies (Toneva & Wehbe, 2019; Toneva et al., 2022a; Aw & Toneva, 2023; Oota et al., 2024b; 2023d;a; Dong & Toneva, 2023).

To map brain activations to these eight language regions, prior studies use the multimodal parcellation of the human cerebral cortex based on the Glasser Atlas (Glasser et al., 2016), which consists of 180 regions of interest in each hemisphere. The data covers eight language brain ROIs with the following subdivisions: (i) AG: PFm, PGs, PGi, TPOJ2, and TPOJ3; (ii) ATL: STSda, STSva, STGa, TE1a, TE2a, TGv, and TGd;

---

[2]https://nilearn.github.io/stable/index.html
[3]https://gallantlab.org/pycortex/
[4]https://mne.tools/stable/index.html

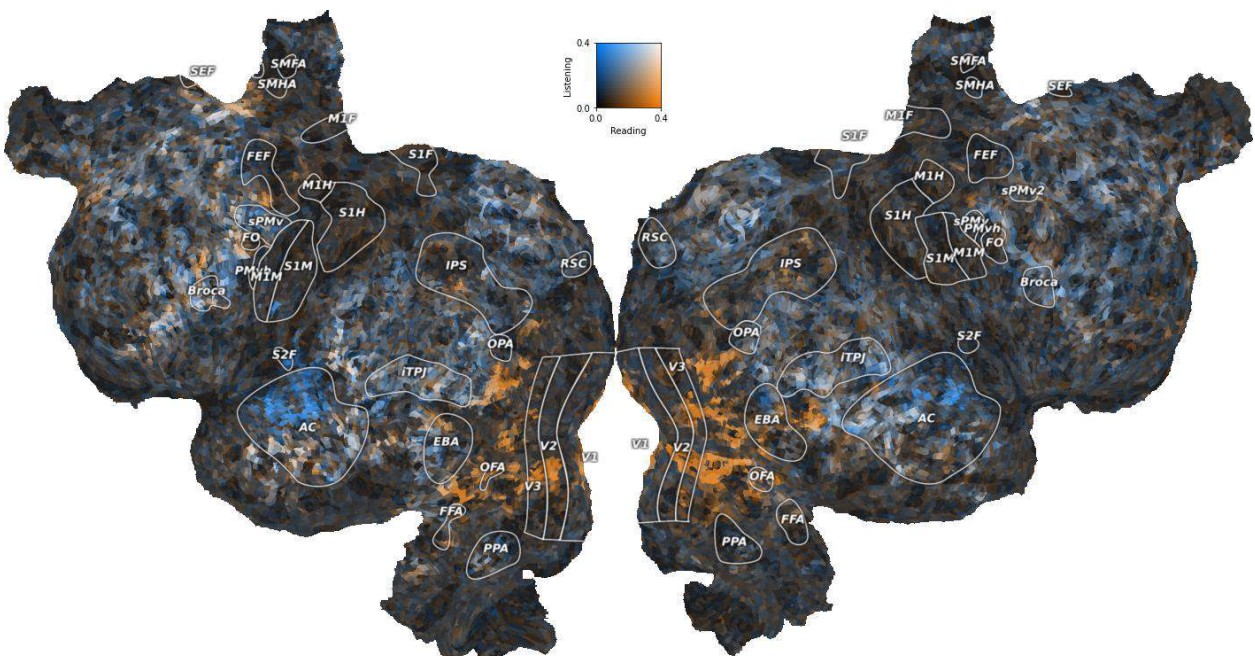

Figure 7: Contrast of estimated cross-subject prediction accuracy for reading and listening for a representative subject (subject-5 from the subset-moth-radio-hour dataset). Blue and Orange voxels depict higher cross-subject prediction accuracy estimates during listening and reading, respectively. Voxels that have similar cross-subject prediction accuracy during reading and listening appear white, and are distributed across language regions. *This figure is adapted from (Oota et al., 2024a), and used with permission from the respective authors.*

(iii) PTL: A5, STSdp, STSvp, PSL, STV, TPOJ1; (iv) IFG: 44, 45, IFJa, IFSp; (v) MFG: 55b; (vi) IFGOrb: a47r, p47r, a9-46v; (vii) PCC: 31pv, 31pd, PCV, 7m, 23, RSC; and (viii) dmPFC: 9m, 10d, d32.

Figure 7 displays the cross-subject prediction accuracy for reading and listening for a representative sample subject from the subset-moth-radio-hour dataset (Oota et al., 2024a). It illustrates that, irrespective of text-evoked or speech-evoked brain activity, high-level information processing occurs in the language regions (indicated by white voxels).

**Auditory cortex.** The auditory cortex (AC) is a specific brain region responsible for processing auditory information, including the perception of sound, speech, music, and other auditory stimuli. Figure 7 illustrates that during speech-evoked brain activity, the early auditory cortex (EAC) has higher prediction accuracy (indicated by blue voxels), signifying early sensory information processing, while high-level information processing occurs in the language regions (indicated by white voxels). The auditory cortex is divided into the following subdivisions (Nastase et al., 2020): (i) EAC (early auditory cortex): A1 (Primary Auditory Cortex), the Lateral Belt (LBelt), Posterior Belt (PBelt), Medial Belt (MBelt), Rostral Intermediate (RI), and (ii) AAC (auditory association cortex): A4 and A5.

Together, these distinct areas work in concert to form a sophisticated system for perceiving and interpreting diverse aspects of auditory stimuli.

**Visual cortex.** The visual cortex is a critical part of the brain responsible for visual information processing, allowing us to see and understand the world around us. Many experiments contrast brain activity elicited by specific image categories. This functional localizer approach has been used to identify many regions of interest (ROIs) in the visual pathways (dorsal and ventral streams), representing information from low-level visual (early) to high-level semantic information.

The early visual cortex (EVC) is primarily responsible for processing basic visual information, including the detection of simple features like edges, colors, shapes, and motion. After initial processing in the early visual cortex, visual information is transmitted along two pathways (dorsal and ventral streams) for higher-level interpretation. The ventral pathway, often referred to as the "what" pathway, extends from the EVC to the inferior temporal cortex (IT). This pathway is crucial for object recognition and form representation. It handles the processing of detailed visual information, enabling us to identify and understand complex stimuli such as faces, objects, and scenes. In contrast, the dorsal pathway, known as the "where" or "how" pathway, projects from the EVC to the posterior parietal cortex. It is primarily responsible for spatial awareness and motion detection. The dorsal stream processes information about the location and movement of objects in space, facilitating the coordination of actions like reaching, grasping, and navigating through the environment. In summary, the Higher visual cortex (HVC) includes regions from both the dorsal and ventral pathways, each specializing in different aspects of visual processing.

Overall, the visual cortex is divided into the following subdivisions: (i) PVC (primary visual cortex): V1, (ii) EVC (early visual cortex): V2, V3 and V4, (iii) VWFA (visual word form area), and (iv) HVC (high-level visual cortex): Ventral visual regions - the extrastriate body area (EBA), occipital face area (OFA), and the fusiform face area (FFA), the occipital place area (OPA), the parahippocampal place area (PPA), and the retrosplenial cortex (RSC); Dorsal visual regions - V3A and V3B (associated with motion processing), V6 and V6A (integrate visual and somatic inputs to inform motor actions), V7 and IPS1 (involved in visual attention and mapping objects in space), and MT & MST (play roles in detecting and analyzing moving objects in our environment).

## 5 Evaluation Metrics

This section discusses popular metrics for evaluating brain encoding and decoding models.

### 5.1 Metrics for Brain Encoding Models

Given a subject and a brain region, let $N$ be the number of samples. Let $Y_i$ and $\hat{Y}_i$ denote the actual and predicted voxel value vectors for the $i^{th}$ sample. Thus, $Y \in \mathbb{R}^{N \times V}$ and $\hat{Y} \in \mathbb{R}^{N \times V}$, where $V$ is the number of voxels in that region. The process comprises two steps, one estimating the predicted value ($\hat{Y}$) and then comparing with the ground truth value ($Y$). Predictions are typically estimated from either linear models such as partial least squares regression (PLS) (Yamins et al., 2014; Schrimpf et al., 2020), ridge regression or other nonlinear models. The predictions from brain encoding models are compared with ground truth values using two common metrics: 2V2 accuracy (Toneva et al., 2020) and Pearson Correlation. They are defined as follows:

**2V2 classification accuracy.** This metric evaluates how close the brain activity prediction is to ground truth, using measures such as Euclidean distance or cosine similarity. It assesses fMRI predictions by using them in a classification task on held-out data in a cross-validation setting. The task is to match predicted left-out brain responses to their corresponding ground truth, as introduced in (Mitchell et al., 2008; Wehbe et al., 2014; Toneva et al., 2020; Aw & Toneva, 2023). Given two sets of brain predictions $\hat{Y}_i$ and $\hat{Y}_j$, and corresponding ground truth $Y_i$ and $Y_j$, the 2V2 classification accuracy is computed as:

$$\frac{1}{\binom{N}{2}} \sum_{i=1}^{N-1} \sum_{j=i+1}^{N} \mathbb{I}[\cos(Y_i, \hat{Y}_i) + \cos(Y_j, \hat{Y}_j) > \cos(Y_i, \hat{Y}_j) + \cos(Y_j, \hat{Y}_i)] \tag{1}$$

where $\cos(A, B)$ denotes the cosine similarity between vectors A and B, and $\mathbb{I}[c]$ is an indicator function such that $\mathbb{I}[c] = 1$ if $c$ is true, else it is 0. Higher 2V2 accuracy indicates better performance. Figure 8 (left) illustrates the computation of 2V2 Accuracy for the case where samples $i$ and $j$ correspond to the brain activity of concepts "dog" and "house", respectively. This metric was proposed to boost the signal-to-noise ratio in estimating brain alignment for single-trial data (Aw & Toneva, 2023). Under this metric, chance performance is 50%.

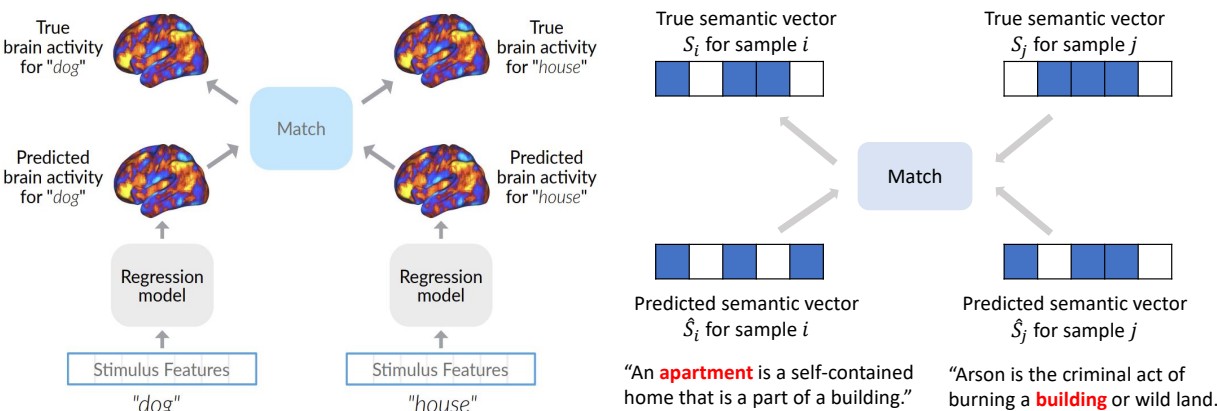

Figure 8: Evaluation Metrics for Brain Encoding and Decoding. (Left) Illustration of 2V2 Accuracy computation, a metric used in brain encoding models to evaluate prediction accuracy. (Right) Demonstration of Pairwise Accuracy calculation, commonly used in brain decoding models. *This figure is from Pereira et al. (2018), and used with permission from the authors.*

**Pearson correlation.** This metric evaluates the similarity between fMRI predictions ($\hat{Y}_i$) and the corresponding true fMRI data ($Y_i$) by computing the Pearson correlation for each voxel $i$. The Pearson correlation for voxel $i$ is computed as:

$$\text{PC}_i = \text{corr}[Y_i, \hat{Y}_i] \tag{2}$$

where corr is the correlation function. The average Pearson correlation across all voxels, Pearson Correlation Coefficient (PCC), is then computed as:

$$\text{PCC} = \frac{1}{V} \sum_{i=1}^{V} \text{corr}[Y_i, \hat{Y}_i] \tag{3}$$

where $V$ denotes the number of voxels. This metric is widely used in cognitive neuroscience (Jain & Huth, 2018; Toneva & Wehbe, 2019; Caucheteux et al., 2021a; Goldstein et al., 2022; Aw & Toneva, 2023; Oota et al., 2022b; 2024b).

**Cross-subject prediction accuracy.** To account for intrinsic noise in biological measurements and obtain a more accurate estimate of model performance, Schrimpf et al. (2021) proposed an approach to estimate cross-subject prediction accuracy. This is achieved by estimating the amount of brain response in one subject that can be predicted using only data from a combination of other subjects using an encoding model. For instance, consider the *Harry Potter* dataset with $n = 8$ participants. The process begins by subsampling the data with $n$ participants into all possible combinations of $s$ participants for all $s \in [2, 8]$ (e.g., 2, 3, 4, 5, 6, 7, 8 for $n = 8$). For each of these subsamples, a random participant is selected as the target to be predicted from the remaining $s-1$ participants. This approach allows for predictions ranging from 1 subject based on 1 other subject, to 1 subject based on 2 subjects, and so on up to 1 subject based on 7 subjects, ultimately yielding a mean score for each voxel in that subsample. To extrapolate these results to infinitely many humans and obtain the highest possible (most conservative) estimate, the following equation is fitted:

$$v = v_0 \times \left(1 - e^{-\frac{x}{\tau_0}}\right) \tag{4}$$

In this equation, $x$ represents each subsample's number of participants, $v$ denotes each subsample's correlation score, and $v_0$ and $\tau_0$ are the fitted parameters. This fitting procedure is performed independently for each voxel, using 100 bootstraps to estimate variance. Each bootstrap iteration draws $x$ and $v$ with replacement. The final ceiling value is determined by taking the median of the per-voxel ceilings $v_0$. This approach provides

a robust estimate of cross-subject prediction accuracy, accounting for the variability inherent in biological measurements.

**Normalized brain alignment.** The neural model predictivity values can be normalized by their respective subject estimated cross-subject prediction accuracies, as proposed by Schrimpf et al. (2021). Thus, normalized brain alignment on a dataset is computed as Pearson's correlation between model predictions and neural recordings divided by the estimated ceiling and averaged across voxel locations and participants.

## 5.2 Metrics for Brain Decoding Models

Brain decoding methods are evaluated using several popular metrics, including pairwise accuracy and rank accuracy (Pereira et al., 2018; Sun et al., 2019; 2020; Oota et al., 2022c). Additionally, other metrics such as $R^2$ score, mean squared error, and Representational Similarity Matrix are also employed (Cichy et al., 2019; 2021).

**Pairwise accuracy.** This metric is computed as follows:

1. Predict all test stimulus vector representations using a trained decoder model.

2. Let $X = [X_0, X_1, \cdots, X_n]$ and $\hat{X} = [\hat{X}_0, \hat{X}_1, \cdots, \hat{X}_n]$ denote the "true" (stimuli-derived) and predicted stimulus representations for $n$ test instances, respectively.

3. For a given pair $(i, j)$ where $0 \leq i, j \leq n$, the score is 1 if:

$$\text{corr}(X_i, \hat{X}_i) + \text{corr}(X_j, \hat{X}_j) > \text{corr}(X_i, \hat{X}_j) + \text{corr}(X_j, \hat{X}_i) \tag{5}$$

Otherwise, the score is 0. Here, corr denotes the Pearson correlation.

Figure 8 (right) illustrates the computation of Pairwise Accuracy for the case where samples $i$ and $j$ correspond to the brain activations for text stimuli "apartment" and "building" respectively. The final pairwise matching accuracy per participant is the average of scores across all pairs of test instances.

**Rank accuracy.** This metric is computed as follows:

1. Compare each decoded vector to all the "true" stimuli-derived semantic vectors and rank them by their correlation.

2. The classification performance reflects the rank $r$ of the stimuli-derived vector for the correct word or picture stimuli:

$$\text{Rank accuracy} = 1 - \frac{r - 1}{\#\text{instances} - 1} \tag{6}$$

3. The final accuracy value for each participant is the average rank accuracy across all instances.

These metrics provide comprehensive evaluation of brain decoding models, assessing their ability to accurately reconstruct stimuli from brain activity patterns.

## 5.3 Representational Similarity Metrics for Brain Encoding and Decoding Models

Three popular representational similarity metrics commonly used in brain encoding and decoding studies (Anderson et al., 2017b; Marques et al., 2021; AlKhamissi et al., 2024) are representational dissimilarity matrices (RDM) (Kriegeskorte et al., 2008), centered kernel alignment (CKA) (Kornblith et al., 2019) and normalized distribution similarity (NDS). Unlike brain encoding or decoding models that rely on linear regression, both of these metrics are nonparametric and do not require any additional training.

**Representational Dissimilarity Matrices (RDMs)** Kriegeskorte et al. (2008) introduced RDMs as a solution to the challenge of integrating brain-activity measurements, behavioral observations, and computational models in systems neuroscience. RDMs are part of a broader analytical framework referred to as representational similarity analysis (RSA). In practical terms, to compute the dissimilarity matrix for an $N$-dimensional network's responses to $\mathcal{S}$ different stimuli, an $S \times S$ matrix of distances between all pairs of evoked responses is generated for both the brain activity and the activations of the DNN model (Harvey et al., 2024). The correlation between these two matrices is then used as a measure of brain alignment.

1. To construct the RDM, compute the pairwise dissimilarities between all pairs of responses for the $M$ different stimuli.

2. This results in a matrix $S \times S : D$, where each element $D_{ij}$ represents the dissimilarity between the representation vectors ($\vec{r}_i$ and $\vec{r}_j$) corresponding to the stimuli $i$ and $j$.

$$D_{ij} = 1 - \frac{\text{cov}(\vec{r}_i, \vec{r}_j)}{\sqrt{\text{var}(\vec{r}_i) \cdot \text{var}(\vec{r}_j)}} \tag{7}$$

3. Similarly, construct a matrix $S \times S : B$, where each element $B_{ij}$ represents the dissimilarity between brain responses ($\vec{b}_i$ and $\vec{b}_j$) corresponding to stimuli $i$ and $j$.

$$B_{ij} = 1 - \frac{\text{cov}(\vec{b}_i, \vec{b}_j)}{\sqrt{\text{var}(\vec{b}_i) \cdot \text{var}(\vec{b}_j)}} \tag{8}$$

4. Finally, RSA involves calculating the correlation between the two RDMs D and B. This correlation quantifies the similarity between the representational spaces of the two systems (brain activity and the model).

**Centered Kernel Alignment (CKA).** Kornblith et al. (2019) introduced CKA as a substitute for Canonical Correlation Analysis (CCA) to assess the similarity between neural network representations. Unlike linear predictivity, it is a nonparameteric metric and therefore does not require any additional training. CKA is particularly effective with high-dimensional representations, and is reliabile in identifying correspondences between representations in networks trained from different initializations (Kornblith et al., 2019). The linear CKA metric is computed as follows:

1. CKA between input matrices X and Y is computed as

$$\text{CKA}(X, Y) = \frac{\text{HSIC}(K, L)}{\sqrt{\text{HSIC}(K, K)\text{HSIC}(L, L)}} \tag{9}$$

where K and L are the similarity matrices computed as $K = XX^T$ and $L = YY^T$, respectively.

2. The terms $\text{HSIC}(K, L)$, $\text{HSIC}(K, K)$, and $\text{HSIC}(L, L)$ represent the Hilbert-Schmidt Independence Criterion (HSIC) (Gretton et al., 2005) values.

3. CKA values range from 0 to 1, where CKA = 1 indicates perfect similarity, and CKA = 0 indicates no similarity.

**Normalized Distribution Similarity (NDS).** Marques et al. (2021) introduced normalized distribution similarity. The normalized distribution similarity score is calculated as follows:

$$\text{Score} = \frac{D_m}{D_b} \tag{10}$$

$$D_m = 1 - \text{KS}_c^{\text{M}-\text{E}}$$

$$\mathrm{D}_b = 1 - \mathrm{KS}_c^{\mathrm{E-E}}$$

where $\mathrm{D}_m$ is the distance Kolmogorov-Smirnov (KS) between the empirical model distribution (M) and the empirical biological distribution (E), and $\mathrm{D}_b$ is an estimate of the expected distance from KS and c denotes the ceiling. A low score indicates that the model distribution does not match the biological distribution, while a score of 1 means that the model distribution is indistinguishable from the biological distribution when considering experimental variability.

### 5.4 Noise Ceiling Metric for Quantifying Model Performance

In linguistic or auditory or vision brain encoding studies, researchers aim to better interpret the accuracy of model-level predictions by quantifying the maximal possible prediction performance. This is often achieved by treating inter-participant variability as a "noise ceiling" representing the portion of neural response variability that cannot be predicted by a computational model without access to individual participant measurements (Schrimpf et al., 2020; 2021; Tuckute et al., 2024; Oota et al., 2024a; AlKhamissi et al., 2024).

Several studies compute the noise ceiling by using repeated trials of stimuli to obtain the maximum explained variance (Schoppe et al., 2016; Deniz et al., 2019; Popham et al., 2021). By presenting the same stimulus multiple times and recording the neural responses, researchers can estimate the consistency of responses within and between participants. This approach helps distinguish the variance attributable to the stimulus from the variance due to noise or individual differences. Incorporating noise ceiling into evaluation metrics provides a benchmark for interpreting model performance, allowing for a more accurate assessment of how much of the neural data the model can potentially explain.

This consideration is crucial when comparing models to neural data because it sets an upper limit on the expected performance. Without accounting for the noise ceiling, a model's predictive accuracy might be underestimated due to the inherent variability in neural responses that no model could capture. By normalizing model performance relative to the noise ceiling, researchers can more fairly compare different models and better understand the extent to which a model approximates the true neural processing of linguistic, auditory or visual information.

## 6 Brain Encoding

Encoding is the process of learning the mapping from the stimulus domain to neural activation. The quest in brain encoding is for "reverse engineering" the algorithms that the brain uses for sensation, perception, and higher-level cognition. The foundational approach to constructing a brain encoder, illustrated in Figure 10(a), adopts a general brain alignment strategy previously implemented in several notable studies (Jain & Huth, 2018; Toneva & Wehbe, 2019; Aw & Toneva, 2023; Oota et al., 2024b). This method predicts brain recordings at every voxel for each participant, utilizing DNN representations that mirror the participant's engagement in tasks such as reading or listening.

Building on this foundation, recent advancements in neuroimaging technologies have enhanced our ability to closely approximate how the brain responds to different stimuli, thereby deepening our understanding of the brain's information processing mechanisms. Concurrently, advancements in deep neural network (DNN) models have led to the development of highly efficient models across different modalities, including language, vision, speech, and multimodal interactions. These models have set new benchmarks in performance for a wide range of applications. Leveraging cutting-edge neuroimaging techniques and DNN models, this section offers a comprehensive review of the task settings for brain encoding and the latest achievements in understanding language processing, visual object recognition, auditory perception, and multimodal processing in the brain.

In the discussion on encoding task settings, we present stimulus downsampling, TR (Repetition Time) alignment, and voxelwise encoding models. In linguistic brain encoding, we explore recent breakthroughs in applied Natural Language Processing (NLP) that facilitate the reverse engineering of the language function of the brain. In the realm of vision brain encoding, pioneering results have been achieved in reverse engineering the function of the ventral visual stream for object recognition, thanks to the advancements and impressive

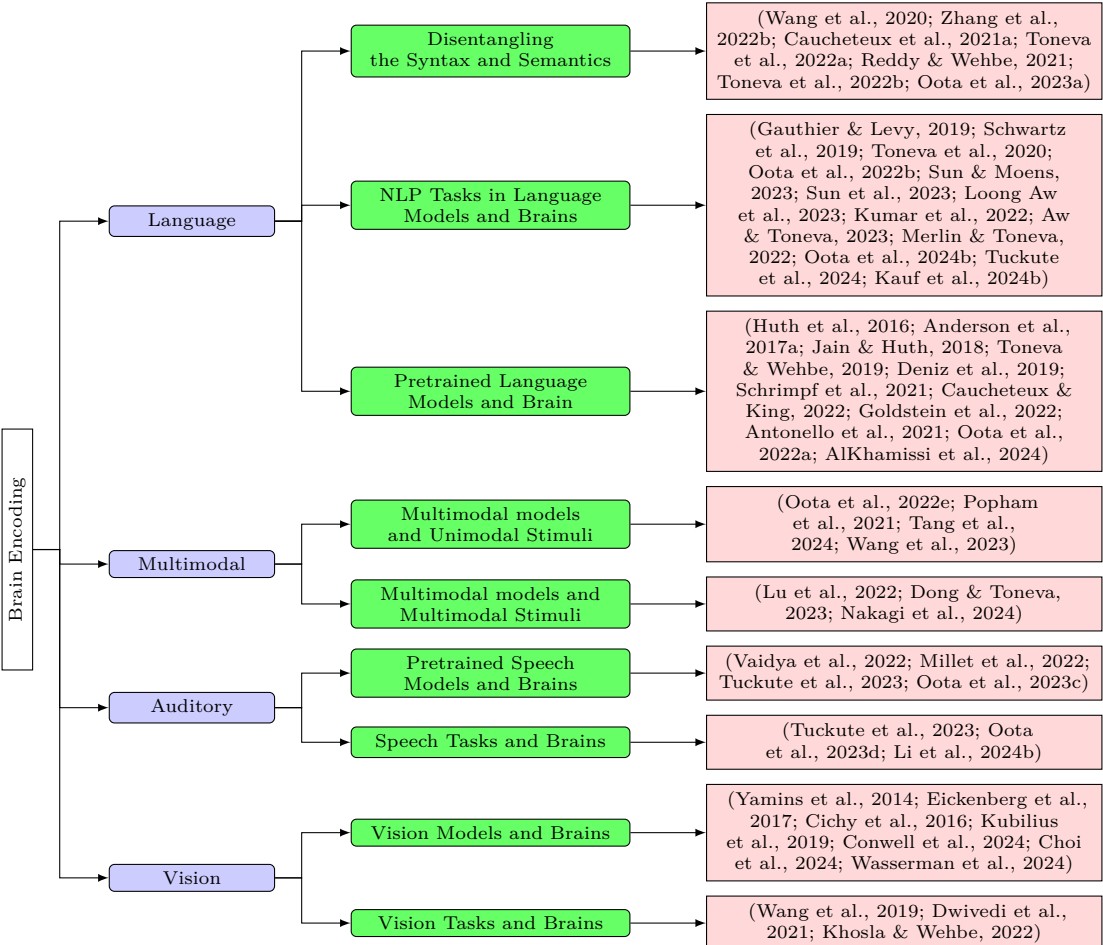

Figure 9: Categorization of Brain Encoding Studies Across Different Modalities and Tasks. The categorization is organized by stimulus modalities (e.g., language, vision, auditory, and multimodal) and the nature of the representations used—either from pretrained models or from task-specific models.

successes of deep Convolutional Neural Networks (CNNs) and Vision Transformers. Additionally, we present the latest insights into auditory and multimodal brain encoding. This systematic approach informs the organization of this section. Overall, Figure 9 classifies the encoding literature along various stimulus domains such as vision, auditory, multimodal, and language and the corresponding tasks in each domain. Finally, Table 6 summarizes various encoding models proposed in the literature related to textual, audio, visual, and multimodal stimuli.

## 6.1 Encoding Task Settings

**Stimulus downsampling.** In the context of narrative story reading or listening, the rate of fMRI data acquisition is lower than the rate at which the text stimulus is presented to the subjects; several words fall under the same TR in a single acquisition. Therefore, previous studies match the stimulus acquisition rate to fMRI data recording by downsampling the stimulus features using a 3-lobed Lanczos filter (Huth et al., 2016; Jain & Huth, 2018; Toneva & Wehbe, 2019; Antonello et al., 2021; Oota et al., 2024b). After downsampling, word embeddings corresponding to each TR are obtained.

For naturalistic audio, Vaidya et al. (2022); Antonello et al. (2024b) windowed the stimulus waveform with a sliding window of size 16s and stride 100 ms before feeding it into the model. Further, the features are

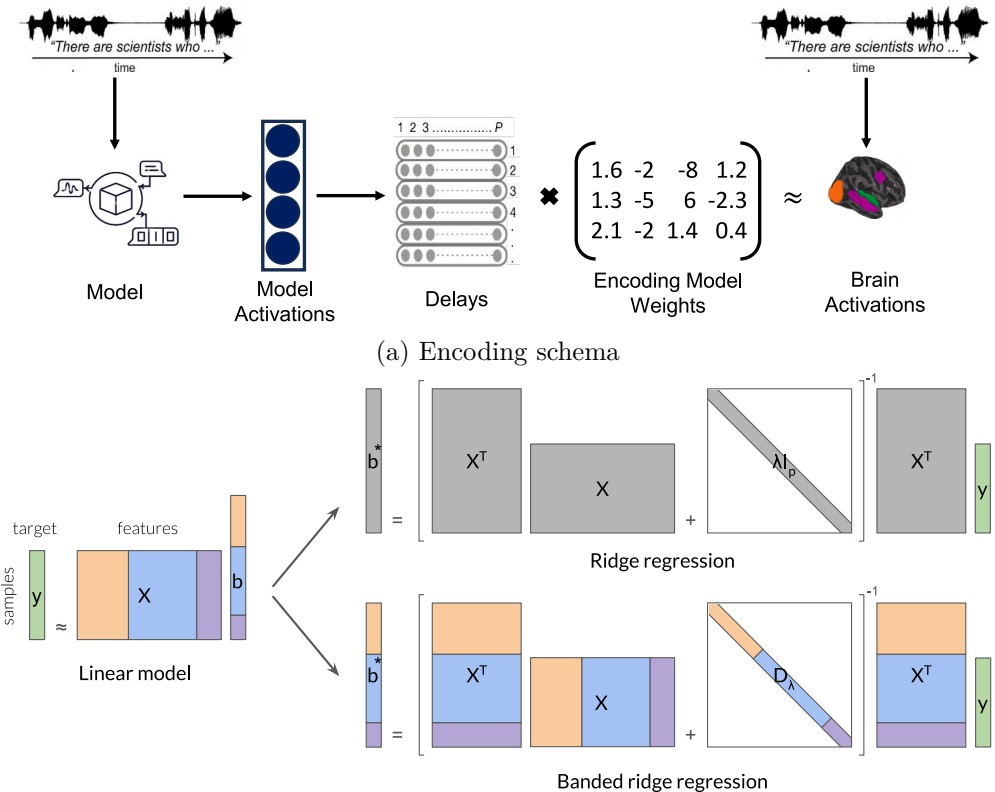

(a) Encoding schema

(b) Ridge regression vs. Banded ridge regression

Figure 10: (a) Scheme for Brain Encoding (top): This approach learns a function to predict the brain recordings at every voxel of each participant using the model representations that correspond to the same text read or listened by the participant. (b) Ridge regression vs. Banded ridge regression (bottom) plot is *adapted from (la Tour et al., 2022), and used with permission from the respective authors.* Each color (or band) represents a different feature space.

downsampled as previously described, using Lanczos interpolation, to match the sampling rate of fMRI recordings.

Similarly, for naturalistic videos, the rate of fMRI data acquisition (TR = 2 seconds) in the shortclips dataset (Huth et al., 2022) is lower than the rate at which the stimulus was presented to the subjects (15 frames per second). Consequently, 30 frames of a video were viewed during the same TR for a single fMRI acquisition (Popham et al., 2021). This synchronization between the stimulus presentation rate and fMRI data recording is then leveraged to train our encoding models.

**fMRI preprocessing.** The minimal preprocessing steps described in *fMRIPrep* framework (Esteban et al., 2019) to obtain the preprocessed fMRI data from raw BOLD fMRI recordings are as follows. For raw fMRI data and for each subject separately, using *fMRIPrep*, the following steps should be performed:

- Take the raw functional MRI data in either DICOM or NIfTI format, and ensure that all DICOM or NIfTI files from the fMRI scan are organized in a single folder. DICOM files typically consist of a series of 2D images for each slice and time point, whereas a NIfTI file is a 4D file containing all the volumes (time points) of the scan. It is optional to start directly with DICOM files or first convert DICOM files to NIfTI format for preprocessing.
- BOLD reference image estimation:
    1. Mean Functional Image: The mean of all functional volumes is computed to create a stable and representative reference image.

2. Single Functional Volume: Alternatively, a single high-quality volume (e.g., the first or a middle time point) may be selected as the reference.

- Motion Correction (Realignment): All functional images are aligned to the reference image that correct for translational and rotational movements.
- Coregistration: The reference functional image is aligned with the participant's anatomical (structural) MRI scan. This step bridges functional and anatomical data, allowing for accurate localization of brain activity.
- Normalization: Images are transformed into a standardized space (e.g., MNI space) to allow for group analyses across participants.
- Smoothing (Optional): Apply a Gaussian kernel to smooth the data, enhancing the signal-to-noise ratio (SNR).
- Confound regression: Perform confound regression on both smoothed and unsmoothed data to mitigate artifacts from head motion and physiological noise.

**fMRI Repetition Time (TR) alignment.** To account for the slowness of the hemodynamic response, previous studies generally model the Hemodynamic Response Function (HRF) using a finite impulse response (FIR) filter per voxel and for each subject separately, with a delay of 8 to 12 seconds (Jain & Huth, 2018; Toneva & Wehbe, 2019; Popham et al., 2021; Oota et al., 2024b; Antonello et al., 2024b). Table 5 summarizes current brain encoding studies with a fixed HRF delay.

**MEG preprocessing and alignment.** Several encoding and decoding studies using MEG recordings Toneva et al. (2020; 2022a); Gwilliams et al. (2023); Hebart et al. (2023); Benchetrit et al. (2024) describe the following minimal preprocessing steps. For raw MEG data and for each subject separately, using *MNE-Python library (Gramfort et al., 2013)*[5], the following steps should be executed:

- **Bandpass filtering in MEG analysis** involves allowing only a specific range of frequencies to pass through, filtering out lower frequencies (such as slow drifts) and higher frequencies (like noise or muscle artifacts). Specifically, in language ERP (Event-Related Potential) studies (Widmann & Schröger, 2012; Tanner et al., 2015), using a high-pass filter above 0.3 Hz introduced significant effects too early compared to unfiltered data. Based on this, these studies suggest an optimal high-pass value of 0.1 Hz for language processing. Therefore, the choice of frequency range for bandpass filters depends on the brain signals being studied, with prior studies typically using a range between 0.1 and 40 Hz (Toneva et al., 2020; Gwilliams et al., 2023; Hebart et al., 2023).
- **Temporally decimating the data or downsampling it over time** reduces the computational load for downstream analyses. For example, if MEG data is originally sampled at 1,000 Hz (1,000 data points per second) and is decimated by a factor of 10, the resulting data will have a sampling rate of 100 Hz (100 data points per second).
- **Segmenting the continuous signals** into smaller, distinct time intervals or epochs is often done to isolate specific events or time windows of interest for further analysis. For instance, Toneva et al. (2022a) analyzed the data from the beginning of the stimulus word presentation (i.e., 0 ms) to 800 ms, while Hebart et al. (2023) used the signal from an optical sensor to epoch the continuous data from –100 ms to 1,300 ms relative to stimulus onset.
- **Applying baseline correction** adjusts the signal relative to the pre-stimulus period, removing any potential drift or noise.
- **Clipping the MEG data** between the fifth and ninety-fifth percentiles across channels reduces the impact of extreme values or outliers.

## 6.2 Voxelwise Encoding Model

The main goal of the voxel-wise encoding model is to predict brain responses associated with each brain voxel given a stimulus. To estimate the brain alignment of a DNN model of stimulus representations, researchers

---

[5]https://mne.tools/stable/index.html

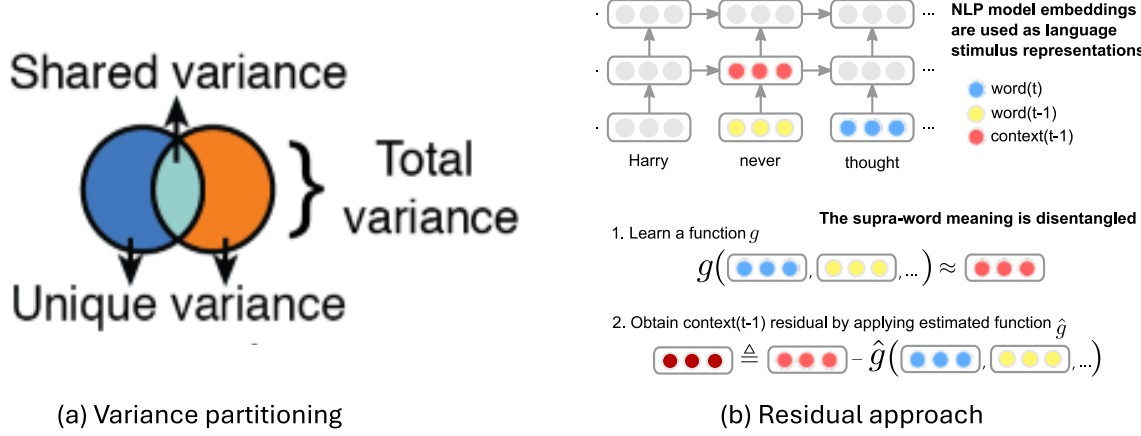

(a) Variance partitioning  (b) Residual approach

Figure 11: Methods for Interpreting Neural Representations in Brain Encoding Models. This figure illustrates two key approaches used to analyze the contribution of different features in neural encoding: (Left) Variance partitioning approach (Lescroart, 2017) and (Right) Residual approach (Toneva et al., 2022a). *Adapted from (Toneva et al., 2022a), and used with permission from the respective authors.*

train standard voxel-wise encoding models (Deniz et al., 2019; Toneva & Wehbe, 2019). Specifically, for each voxel and participant, prior studies train fMRI encoding models using ridge regression to predict the fMRI recording associated with this voxel as a function of the stimulus representations obtained from DNN models. To simultaneously accommodate different feature spaces, which may necessitate varying levels of regularization, Nunez-Elizalde et al. (2019) proposed a voxel-wise encoding model that utilizes an advanced form of ridge regression. This method, known as banded ridge regression, introduces individual regularization parameters for each feature space, as illustrated in Figure 10 (b). Before performing ridge regression or banded ridge regression, we first z-score normalize each feature channel separately for training and testing. This is done to match the features to the fMRI responses, which are also z-scored for training and testing. Formally, at time step (t), we encode the stimuli as $X_t \in \mathbb{R}^{N \times D}$ and brain region voxels $Y_t \in \mathbb{R}^{N \times V}$, where $N$ is the number of training examples, $D$ denotes the dimension of the concatenation of delayed TRs, and $V$ denotes the number of voxels. To find the optimal regularization parameter for each feature space, we use a range of regularization parameters that is explored using cross-validation.

To automate the voxelwise encoding pipeline, several popular libraries have recently been introduced to perform voxelwise brain encoding: (1) Voxelwise tutorials[6] and (2) Himalaya[7]. These libraries are specifically designed for fMRI encoding models.

### 6.3 Interpreting brain recordings through the robustness of DNN model representations

In this section, we discuss four popular robustness methods to interpret the contribution of stimulus representations obtained from DNN models to brain alignment.

**Variance partitioning.** Variance partitioning quantifies the unique contribution of different stimulus features to BOLD responses. For variance partitioning (Lescroart, 2017; Deniz et al., 2019; Vaidya et al., 2022), set theory is used to calculate the common variance (as the intersection of various combinations of feature spaces) and the unique variance (as the set difference for each individual feature space), as shown in Figure 11 (a). Overall, the total variance explained by each model is computed as the unique variance explained by each model and the shared variance across models. This variance partition approach is computed separately for each voxel, then averaged across ROIs and across subjects.

---

[6]https://github.com/gallantlab/voxelwise_tutorials
[7]https://github.com/gallantlab/himalaya

Table 5: Summary of Brain Encoding Studies with HRF delays. Here, |S| denotes number of participants. These are studies on English text using fMRI activations.

| Authors | Stimulus Representations | |S| | Dataset | Delays |
|---|---|---|---|---|
| Wehbe et al. (2014) | Non-negative sparse embedding | 8 | Harry-Potter | 8secs (4 TRs) |
| Jain et al. (2020) | LSTM | 6 | Moth-Radio-Hour | 8secs (4 TRs) |
| Jain & Huth (2018) | LSTM | 6 | Moth-Radio-Hour | 8secs (4 TRs) |
| Caucheteux et al. (2021a) | GPT-2 | 345 | Narratives | 7.5secs (5 TRs) |
| Reddy & Wehbe (2021) | Syntax Parsers, BERT | 8 | Harry-Potter | 8secs (4 TRs) |
| Merlin & Toneva (2022) | GPT2 | 8 | Harry-Potter | 8secs (4 TRs) |
| Aw & Toneva (2023) | BART, LongT5, LED | 8 | Harry-Potter | 8secs (4TRs) |
| Antonello et al. (2021) | 100 Language Models | 7 | Moth-Radio-Hor | 8secs (4 TRs) |
| Chen et al. (2024a) | BERT | 9 | Moth-Radio-Hour | 8secs (4 TRs) |
| Chen et al. (2024b) | mBERT | 6 | Bilingual Moth-Radio-Hour | 8secs (4 TRs) |
| Oota et al. (2024b) | BERT and Probing Tasks | 18 | Narratives 21$st$-Year | 9secs (6 TRs) |
| Oota et al. (2023d) | BERT, GPT-2, Wav2Vec2.0 | 6 | Moth-Radio-Hour | 12secs (6 TRs) |

**Task-Performance aligned approach.** The task-Performance aligned approach first relates model representations to the human brain data, followed by an independent examination of the related model to some task performance or behavioral output. For instance, Schrimpf et al. (2021) tests the computations of a language model that may underlie human language understanding. This is accomplished by an independent examination of the relationship between the models' ability to predict an upcoming word and their brain predictivity. Similarly, Goldstein et al. (2022) provides empirical evidence that both the human brain and language models engage in continuous next-word prediction before word onset. Analogous to linguistic encoding studies, many vision encoding studies adopt this approach to validate the significance of relating task performance to encoding performance (Yamins & DiCarlo, 2016; Schrimpf et al., 2020; Cao et al., 2021).

**Residual approach.** In contrast to the indirect approach, the method proposed by Toneva et al. (2022a) can directly estimate the impact of a specific feature on the alignment between the model and the brain recordings by observing the difference in alignment before and after the specific feature is computationally removed from the model representations. This method, used to remove the linear contribution of a feature to a model's representation, is one way to implement such a direct approach, as shown in Figure 11 (b). This is why the residual approach is also referred to as direct. Another method was investigated by previous work Oota et al. (2024b); Dong & Toneva (2023) and was shown to yield very similar results.

Other direct approaches have also been proposed in the literature. Most notably, work by Ramakrishnan & Deniz (2021) studies the impact of removing information related to word embeddings directly from brain responses on a downstream task. Conceptually, the results obtained from this approach and ours should be similar because the feature is completely removed from either the brain alignment input, target, or both and thus cannot further impact the observed alignment.

**Stacked regression.** The stacked regression approach, proposed by Lin et al. (2023), follows a two-level pipeline. The first level consists of different linear regressors, each using a different stimulus feature space as input. At the second level, the parameters $\alpha_j$ are learned for a convex combination of first-level predictors. Overall, the entire stacked model is estimated separately at each voxel. This method is useful when building different encoding models where input feature spaces are correlated (e.g., visual and semantic features of natural images) and for demonstrating the importance of each feature space in predicting a voxel's response.

**Hierarchical processing in DNNs and the Human Brain.** Hierarchical processing is a fundamental characteristic observed in both biological neural systems and deep neural networks (DNNs). To investigate the hierarchy of information processing, researchers primarily focus on an in-depth analysis of how hierarchical structures in deep neural networks (DNNs) mirror the hierarchical processing observed in the human visual cortex, as well as in speech and language comprehension. For example, Yamins & DiCarlo (2016) demonstrate that early layers in convolutional neural networks (CNNs) correspond to primary visual areas (e.g., V1, V2) that process basic features, while deeper layers align with higher visual areas (e.g., IT cortex) responsible for complex object recognition. Similarly, Caucheteux et al. (2021b) employ a model-based approach to successfully replicate the seminal study by Lerner et al. (2011), which revealed the hierarchy of language areas by comparing fMRI data of subjects listening to both regular and scrambled narratives.

Table 6: Summary of Representative Brain Encoding Studies. In this table, |S| represents the number of participants in each dataset.

| | Authors | Dataset Type | Lang. | Stimulus Representations | \|S\| | Dataset |
|---|---|---|---|---|---|---|
| **Text** | (Jain & Huth, 2018) | fMRI | English | LSTM | 6 | Subset Moth Radio Hour |
| | (Toneva & Wehbe, 2019) | fMRI, MEG | English | ELMo, BERT, Transformer-XL | 9 | Harry Potter |
| | (Jat et al., 2020) | MEG | English | ELMo, BERT | 8 | Rafidi2016 MEG data |
| | (Schrimpf et al., 2021) | fMRI, ECoG | English | 43 language models (e.g. GloVe, ELMo, BERT, GPT-2, XLNET) | 20 | Blank2014, Fedorenko2016, Pereira2018 |
| | (Gauthier & Levy, 2019) | fMRI | English | BERT, finetuned NLP tasks (Sentiment, Natural language inference), Scrambling language model | 8 | Pereira2018 |
| | (Deniz et al., 2019) | fMRI | English | GloVe | 9 | Subset Moth Radio Hour |
| | (Jain et al., 2020) | fMRI | English | LSTM | 6 | Subset Moth Radio Hour |
| | (Zhang et al., 2020) | fMRI | English | Word2Vec | 19 | Moth Radio Hour |
| | (Caucheteux et al., 2021a) | fMRI | English | GPT-2, Basic syntax features | 345 | Narratives |
| | (Antonello et al., 2021) | fMRI | English | GloVe, BERT, GPT-2, Machine Translation, POS tasks | 6 | Moth Radio Hour |
| | (Reddy & Wehbe, 2021) | fMRI | English | Constituency, Basic syntax features and BERT | 8 | Harry Potter |
| | (Goldstein et al., 2022) | fMRI | English | GloVe, GPT-2 next word, pre-onset, post-onset word surprise | 8 | Goldstein2022 ECoG |
| | (Oota et al., 2022b) | fMRI | English | BERT and GLUE tasks | 82 | Pereira & Narratives |
| | (Lamarre et al., 2022) | fMRI | English | BERT, Attention Heads | 9 | Subset Moth Radio Hour |
| | (Toneva et al., 2022a) | fMRI, MEG | English | ELMo, BERT, Context Residuals | 8 | Harry Potter |
| | (Aw & Toneva, 2023) | fMRI | English | BART, Longformer, Long-T5, BigBird, and corresponding Booksum models as well | 8 | Harry Potter |
| | (Zhang et al., 2022c) | fMRI | English, Chinese | Node Count | 19, 12 | Zhang |
| | (Oota et al., 2023a) | fMRI | English | Constituency, Dependency trees, Basic syntax features and BERT | 82 | Narratives |
| | (Chen et al., 2024a) | fMRI | English | Time scale features of BERT | 9 | Subset Moth Radio Hour |
| | (Tuckute et al., 2024) | fMRI | English | BERT-Large, GPT-2 XL | 12 | Tuckute2024 |
| | (Kauf et al., 2024b) | fMRI | English | BERT-Large, GPT-2 XL | 12 | Pereira |
| | (Singh et al., 2023) | fMRI | English | BERT-Large, GPT-2 XL, Text Perturbations | 5 | Pereira2018 |
| **Visual** | (Wang et al., 2019) | fMRI | - | 21 downstream vision tasks | 4 | BOLD 5000 |
| | (Kubilius et al., 2019) | fMRI | - | CNN models AlexNet, ResNet, DenseNet | 1472 | Human behavioral data |
| | (Dwivedi et al., 2021) | fMRI | - | 21 downstream vision tasks | 4 | BOLD 5000 |
| | (Khosla & Wehbe, 2022) | fMRI | - | CNN models AlexNet | 4 | BOLD 5000 |
| | (Conwell et al., 2024) | fMRI | - | CNN models AlexNet | 4 | BOLD 5000 |
| **Audio** | (Millet et al., 2022) | fMRI | English | Wav2Vec2.0 | 345 | Narratives |
| | (Vaidya et al., 2022) | fMRI | English | APC, AST, Wav2Vec2.0, and HuBERT | 7 | Moth Radio Hour |
| | (Tuckute et al., 2023) | fMRI | English | 19 Speech Models (e.g. DeepSpeech, Wav2Vec2.0, VQ-VAE) | 19 | Passive listening |
| | (Oota et al., 2023c) | fMRI | English | 5 basic and 25 deep learning based speech models (Tera, CPC, APC, Wav2Vec2.0, HuBERT, DistilHuBERT, Data2Vec | 6 | Moth Radio Hour |
| | (Li et al., 2024b) | fMRI | English | Wav2Vec2.0 and SUPERB tasks | 21 | Narratives |
| **Multi Modal** | (Dong & Toneva, 2023) | fMRI | English | Merlo Reseve | 5 | Neuromod |
| | (Popham et al., 2021) | fMRI | English | 985D Semantic Vector | 5 | Moth Radio Hour & Short Movie Clips |
| | (Wang et al., 2023) | fMRI | English | CLIP | 5 | BOLD5000 |
| | (Lu et al., 2022) | fMRI | English | BriVL | 5 | Pereira & Short Movie Clips |
| | (Tang et al., 2024) | fMRI | English | BridgeTower | 5 | Moth Radio Hour & Short Movie Clips |
| | (Nakagi et al., 2024) | fMRI | English | BERT, GPT-2, LLaMa | 5 | Moth Radio Hour & Short Movie Clips |

## 6.4 Linguistic Encoding

### 6.4.1 Alignment Between Pretrained Language Models (LMs) and the Brain

Previous works have investigated the alignment between pretrained language models and brain recordings of people comprehending language. Huth et al. (2016) were able to identify brain ROIs (Regions of Interest) that respond to words with similar meanings, thus building a "semantic atlas" of how the human brain organizes language. Many studies have shown accurate results in mapping brain activity using neural distributed word embeddings for linguistic stimuli (Anderson et al., 2017a; Pereira et al., 2018; Oota et al., 2018; Nishida & Nishimoto, 2018; Sun et al., 2019). Unlike earlier models, where each word is represented as an independent vector in an embedding space, Jain & Huth (2018) built encoding models using rich contextual representations

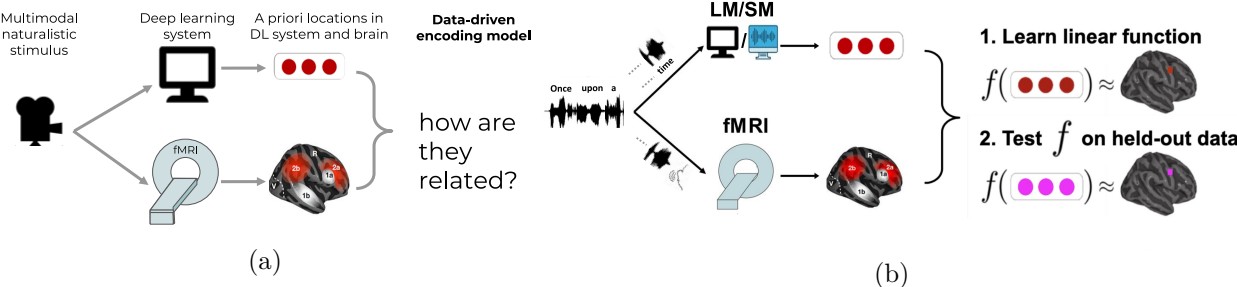

Figure 12: (a) Alignment of representations between deep learning systems and human brains (Toneva & Wehbe, 2019). (b) For instance, a narrative story provided to both the Language model as well as human participants. For the Language model, we extract its representations for every word in the text. For the human participants, we record their brain activity using fMRI. Next, we train a linear function that uses the extracted Language model representations to predict human brain activity. Finally, we test this function on unseen data, and evaluate its accuracy as the amount of "brain alignment" (Toneva & Wehbe, 2019). Here, LM refers to the Language Model, and SM refers to the Speech Model. *These two images are sourced from Cogsci-22 tutorial slides Oota et al. (2022d), and used with permission from the respective authors.*

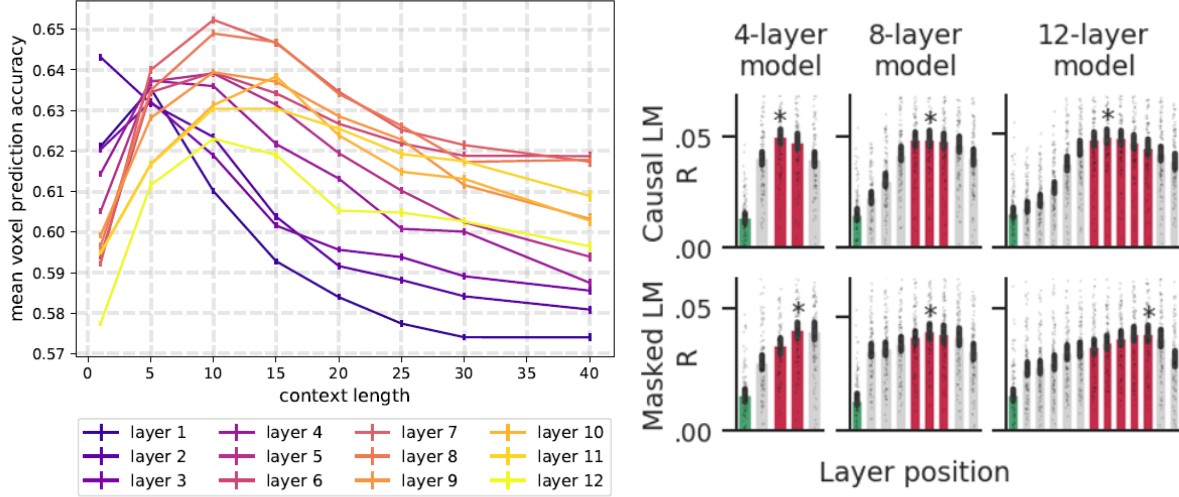

Figure 13: The strongest alignment with high-level language brain regions has consistently been observed in the middle layers. Left: Performance of BERT encoding model for all hidden layers as the amount of context provided to the network is increased (Toneva & Wehbe, 2019). Right: fMRI encoding score (averaged across time and channels) of 6 representative transformers varying in tasks (CLM vs MLM) and depth (4-12 layers) (Caucheteux & King, 2022). *The left Figure is adapted from (Toneva & Wehbe, 2019) and the right Figure is adapted from (Caucheteux & King, 2022). Adapted with permission from the respective authors. Note: Please see a cautionary note about the robustness of middle-layer alignment in the main text.*

derived from an LSTM language model in a story listening task. Using these contextual representations, their approach revealed a clear dissociation dissociation in brain activation, with the auditory cortex (AC) and Broca's area processing shorter contexts, whereas the left Temporo-Parietal junction (TPJ) became more engaged with longer contexts. Hollenstein et al. (2019) presents the first multimodal framework for evaluating six types of word embeddings (Word2Vec, WordNet2Vec (Bartusiak et al., 2019), GloVe, fastText, ELMo, and BERT) on 15 datasets, including eye-tracking, EEG and fMRI signals recorded during language processing. With the recent advances in contextual representations in NLP, few studies incorporated them in relating sentence embeddings with brain activity patterns (Sun et al., 2020; Gauthier & Levy, 2019; Jat et al., 2020).

More recently, researchers have begun to study the alignment of language regions of the brain with the layers of language models (broadly following the method described in Figure 12) and found that the best alignment was achieved in the middle layers of these models (Jain & Huth, 2018; Toneva & Wehbe, 2019; Caucheteux & King, 2022), as shown in Figure 13. Toneva & Wehbe (2019) study how representations of various Transformer models differ across layer depth, context length, and attention type. The results demonstrated that across several larger NLP models, middle layers of language models are well aligned with brain language regions. However, this higher brain alignment in the middle layers is influenced by the model's architecture type (encoder, decoder, or encoder-decoder) rather than being a universal characteristic. Schrimpf et al. (2021) examined the relationship between 43 diverse state-of-the-art language models. They also studied the behavioral signatures of human language processing in the form of self-paced reading times and a range of linguistic functions assessed via standard engineering tasks from NLP. They found that Transformer-based models perform better than RNNs or word-level embedding models. Larger-capacity models perform better than smaller models. Models initialized with random weights (prior to training) perform surprisingly similarly in neural predictivity compared to final trained models, suggesting that network architecture contributes as much or more than experience dependent learning to a model's match to the brain. Antonello et al. (2021) proposed a "language representation embedding space" and demonstrated the effectiveness of the features from this embedding in predicting fMRI responses to linguistic stimuli. Very recent work by Antonello et al. (2024b) tested whether larger open-source models, such as those from the text-based model (OPT (Zhang et al., 2022a) and LLaMA (Touvron et al., 2023)) families, are better at predicting brain responses recorded using fMRI. The results demonstrate that encoding performance improvements scale well with both model size and dataset size, and large datasets will no doubt be necessary in producing useful encoding models.

### 6.4.2 Disentangling Syntax and Semantics

The representations of transformer models like BERT and GPT-2 have been shown to linearly map onto brain activity during language comprehension. Several studies have attempted to disentangle the contributions of different types of information from word representations to the alignment between brain recordings and language models (Lopopolo et al., 2017; Wang et al., 2020; Caucheteux et al., 2021a; Reddy & Wehbe, 2021; Zhang et al., 2022b; Toneva et al., 2022a; Oota et al., 2023a). Wang et al. (2020) proposed a two-channel variational autoencoder model to dissociate sentences into semantic and syntactic representations and separately associate them with brain imaging data to find feature-correlated brain regions. Similarly, Zhang et al. (2022b) separated different syntactic features from pretrained BERT representations, to explore the potential for distinct syntactic and semantic processing language regions in the brain. Compared to lexical word representations, word syntactic features (parts-of-speech, named entities) and word-relation features (semantic roles, dependencies) are distributed across brain networks instead of a local brain region. The previous two studies could not conclude whether all or any of these representations effectively drive the linear mapping between language models (LMs) and the brain. Toneva et al. (2022a) presented an approach to disentangle supra-word meaning from lexical meaning in language models and showed that supra-word meaning is predictive of fMRI recordings in two language regions (anterior and posterior temporal lobes). Similar to the approach presented in Toneva et al. (2022a), Oota et al. (2023b) disentangle the past and future context meaning from word meaning in language models and showed that past context is crucial in obtaining significant results while predicting MEG brain recordings. Caucheteux et al. (2021a) proposed a taxonomy to factorize the high-dimensional activations of language models into four combinatorial classes: lexical, compositional, syntactic, and semantic representations. They found that (1) Compositional representations recruit a more widespread cortical network than lexical ones and encompass the bilateral temporal, parietal, and prefrontal cortices. (2) Contrary to previous claims, syntax and semantics are not associated with separated modules, but, instead, appear to share a common and distributed neural substrate.

While previous works studied syntactic processing as captured through complexity measures (syntactic surprisal, node count, word length, and word frequency) (Zhang et al., 2020; 2022b), very few have studied the syntactic representations themselves (Caucheteux et al., 2021a; Reddy & Wehbe, 2021; Oota et al., 2023a). Studying syntactic representations using fMRI is difficult because (1) representing syntactic structure in an embedding space is a non-trivial computational problem, and (2) the fMRI signal is noisy. To overcome these limitations, Reddy & Wehbe (2021) proposed syntactic structure embeddings that encode the syntactic information inherent in the natural text that subjects read in the scanner. The results reveal that syntactic

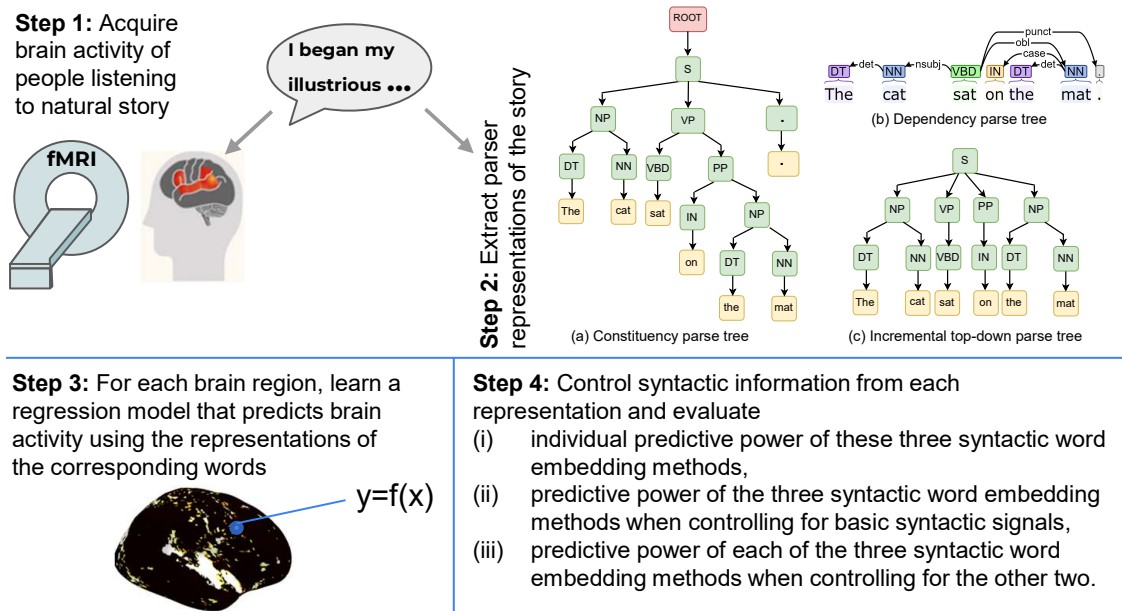

Figure 14: Four steps proposed in Oota et al. (2023a): (1) fMRI acquisition, (2) Syntactic parsing, (3) Regression model training, and (4) Predictive power analysis of the three embedding methods. *This Figure is adapted from (Oota et al., 2023a), and used with permission from the respective authors.*

structure-based features explain additional variance in the brain activity of various parts of the language system, even after controlling for complexity metrics that capture the processing load. Toneva et al. (2022b) further examined whether the representations obtained from a language model align with different language processing regions in a similar or different way. While Reddy & Wehbe (2021) focused on constituency parsing mainly including incremental top-down parsing, Oota et al. (2023a) leverage dependency information more systematically by learning the dependency representations using graph convolutional networks, using the four step recipe as illustrated in Figure 14. The results reveal that constituency tree structure is better encoded in language regions such as bilateral temporal cortex (ATL and PTL) and MFG, while dependency structure is better encoded in AG and PCC language regions.

While previous studies focused on narrative English language stories and have shown that several brain regions are involved in building the hierarchical syntactic structure, a recent study by Zhang et al. (2022c) analyzes the neural basis of such structures between two diverse languages: Chinese and English. The results demonstrate that the brain may use different parsing strategies for different language structures to reduce the cognitive load.

### 6.4.3 NLP Tasks and Linguistic Properties in LMs and Brains

Understanding the reasons behind the observed similarities between language comprehension in language models and brains can lead to more insights into both systems. Further, it is unclear what type of information in the finetuned language models leads to high encoding accuracy. It is unclear whether and how the two systems align in their information processing pipeline. Recent work (Schwartz et al., 2019; Schrimpf et al., 2021; Kumar et al., 2022; Goldstein et al., 2022; Aw & Toneva, 2023; Merlin & Toneva, 2022; Oota et al., 2022b; 2024b; Sun & Moens, 2023; Sun et al., 2023; Loong Aw et al., 2023) addressed this question either by tuning the pretrained language model on downstream NLP tasks or inducing the brain relevant information into the language model.

Several researchers have suggested that one contributor to the alignment is the LM's ability to predict the next word, with a positive relationship between next-word prediction ability and brain alignment across LMs (Schrimpf et al., 2021; Goldstein et al., 2022). However, more recent work shows no simple relationship

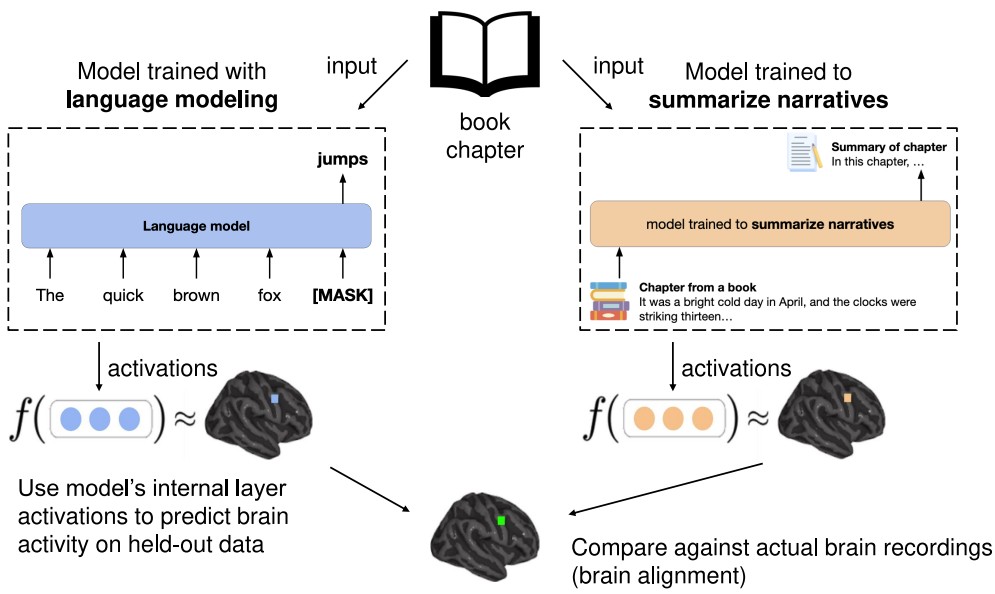

Figure 15: Comparison of brain recordings with language models trained on web corpora (Left) and language models trained on book stories (Right). *This Figure is adapted from (Aw & Toneva, 2023), and used with permission from the respective authors.*

exists, and language modeling loss is not a perfect predictor of brain alignment (Pasquiou et al., 2022; Antonello et al., 2021). Schwartz et al. (2019) finetuned pretrained BERT model to predict brain activity and found that finetuned BERT has modified language representations to better encode the information that is relevant for the prediction of brain activity. Rather than finetuning BERT model on brain data, Oota et al. (2022b) finetuned BERT model on 10 GLUE (General Language Understanding Evaluation) (Wang et al., 2018) tasks to check whether task supervision leads to better encoding models to account for the brain's language representation. Oota et al. (2022b) found that using a finetuned BERT on downstream NLP tasks led to improved brain predictions. The results reveal that reading fMRI was best explained by Co-reference Resolution, NER (Named Entity Recognition), shallow syntax parsing; and listening fMRI was best explained by paraphrasing, summarization, NLI. Since full finetuning generally updates the entire parameter space of the model which has been proven to distort the pretrained features (Kumar et al., 2022), Sun & Moens (2023) explore prompt-tuning that generates representations which better account for the brain's language representations than finetuning. They find that prompt-tuning on tasks dealing with fine-grained concept meaning including Word Sense Disambiguation and Co-reference Resolution yields representations that are better at neural decoding than tuning on other tasks with both finetuning and prompt-tuning. Further, Sun et al. (2023) extended similar prompt-tuning to bridge the gap between human brain and supervised DNN representations of the Chinese language. With the recent success of instruction-tuned large language models, Loong Aw et al. (2023) investigated the effect of instruction-tuning on large language models and alignment with the human brain's language representations. The results demonstrate that instruction-tuning of large language models (LLMs) improves both world knowledge representations and brain alignment, suggesting that mechanisms that encode world knowledge in LLMs also improve representational alignment to the human brain.

To investigate whether large language models with longer context are learning a deeper understanding of the text, Aw & Toneva (2023) used four pretrained large language models (BART (Lewis et al., 2020), Longformer Encoder Decoder (Beltagy et al., 2020), BigBird (Zaheer et al., 2020), and LongT5 (Guo et al., 2022)) and also trained them to improve their narrative understanding, using the method detailed in Figure 15. They find that the improvements in brain alignment are larger for character names than for other discourse features, which indicates that these models are learning important narrative elements. However, it is not understood whether language models with the prediction of the next word are necessary for the observed

brain alignment or simply sufficient, and whether there are other shared mechanisms or information that is similarly important. Merlin & Toneva (2022) proposed two perturbations to pretrained language models that, when used together, can control for the effects of next word prediction and word-level semantics on the alignment with brain recordings. Specifically, they found that improvements in alignment with brain recordings in two language processing regions–Inferior Frontal Gyrus (IFG) and Angular Gyrus (AG)–are due to next word prediction and word-level semantics. However, what linguistic information actually underlies the observed alignment between brains and language models was not clear. Recently, Oota et al. (2024b) tested the effect of a range of linguistic properties (surface, syntactic and semantic) and found that the elimination of each linguistic property results in a significant decrease in brain alignment across all layers of BERT. Further, syntactic properties are more responsible and have the largest effect on the trend of brain alignment across model layers. To further understand what aspects of linguistic stimuli contribute to ANN-to-brain similarity, Kauf et al. (2024b) systematically manipulated the stimuli (i.e., perturbed sentences' word order, removed different subsets of words, or replaced sentences with other sentences of varying semantic similarity) and found that lexical semantic content rather than the sentence's syntactic form is primarily responsible for the DNN-to-brain similarity. Similar to studies on pretrained models and brain similarity, AlKhamissi et al. (2024) investigated the reasons for the similarity of untrained language models and brain alignment by performing mechanistic interpretability of the models. By isolating components of the Transformer architecture (GPT-2 XL), they found that tokenization strategy and multihead attention are the two major components driving this better brain alignment.

Previous studies (Oota et al., 2024b; Kauf et al., 2024b) on brain alignment with language models have shown mixed results, with some finding that syntactic tasks are more responsible and others emphasizing lexical semantic content. To explore this further, Kauf et al. (2024a) investigated the extent to which language comprehension relies on syntactic versus semantic cues by manipulating the grammaticality and meaningfulness of linguistic inputs. Their findings support a strong reliance on syntactic processing rather than shallow, semantics-based processing in the language network.

### 6.4.4 Key Takeaways

- **Alignment with Language Models:**

  1. Language models initialized with random weights (untrained models), the representations induced by architectural priors can exhibit reasonable alignment to brain data.
  2. Across several language models (like ELMo and Transformers), the middle layers of language models align well with brain language regions.
  3. Encoding performance improvements scale well with both model size and dataset size, indicating that large datasets will be essential for producing effective encoding models.

- **Semantic and Syntactic Processing:**

  1. Word syntactic and relation features are distributed across brain networks, unlike lexical word representations, which are localized to specific brain regions.
  2. Contrary to previous claims in Shain et al. (2024), syntax and semantics are not associated with separate modules but instead share common brain language regions and are distributed across the language network.

- **Contextual Representations:**

  1. Brain regions like the auditory cortex and Broca's area are involved in processing shorter contexts, while regions like the left temporo-parietal junction handle longer contexts.
  2. Contextual representations from language models improve the prediction of brain activity compared to traditional word embeddings.
  3. Long past contexts enable better encoding than future or short-scale present contexts.

- **Reasons for DNN-to-Brain similarity**

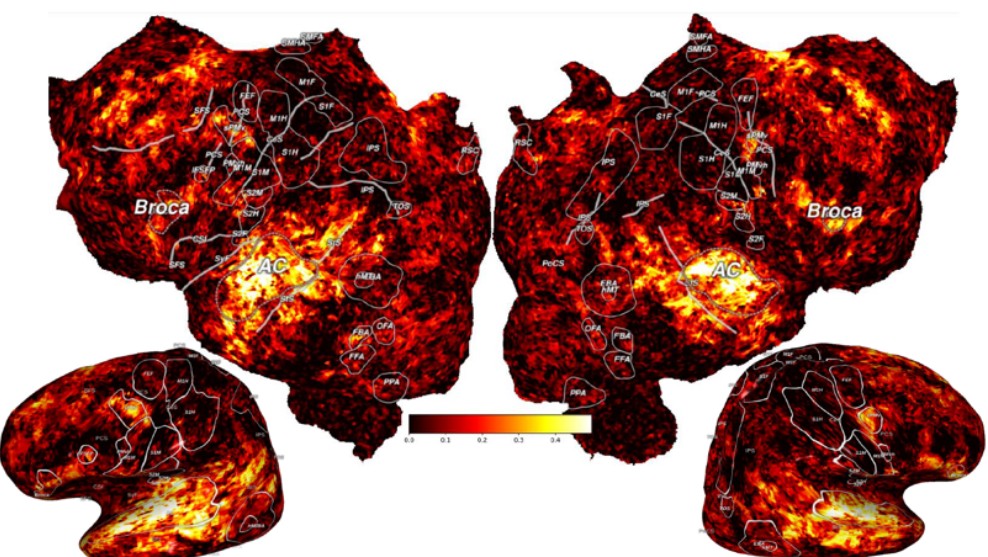

Figure 16: Brain prediction using self-supervised speech model: Data2Vec. The plot from Oota et al. (2023c) shows that speech-based models better predict early auditory cortex. *Adapted with permission from the respective authors.*

1. For untrained language models, mechanistic interpretability of models by isolating critical components of the Transformer architecture reveals that tokenization strategy and multihead attention are the two major components driving brain alignment.
2. For pretrained language models, representational interpretability of models reveals that syntactic properties have the largest effect on the trend of brain alignment across model layers.
3. Strong reliance of syntactic properties rather than semantic-based processing in the language network.

## 6.5 Auditory Encoding

To study auditory processing in the human brain, earlier studies focused on using hand-constructed features such as a number of phonemes, MFCC (Mel Frequency Cepstral Coefficients), spectrotemporal modulations for auditory brain encoding (de Heer et al., 2017). These basic acoustic features are part of a standard model of primary auditory cortex responses to sound encoding (Norman-Haignere & McDermott, 2018; Venezia et al., 2019; Mesgarani et al., 2014). In several other studies, speech stimuli have predominantly been represented as text transcriptions (Huth et al., 2016), or basic features like phoneme rate and the sum of squared FFT (Fast Fourier Transform) coefficients have been employed when constructing encoding models (Pandey et al., 2022). However, text transcription-based methods ignore the raw audio-sensory information completely. The basic speech feature engineering method misses the benefits of transfer learning from rigorously pretrained speech deep learning (DL) models. The benefits of using pretrained speech models include: (i) efficient contextual speech representations, (ii) enhanced accuracy and (iii) flexibility in fine-tuning.

### 6.5.1 Alignment Between Pretrained Speech Models and Brains

Recently, several researchers have used popular deep learning models such as APC (Chung et al., 2020), Wav2Vec2.0 (Baevski et al., 2020), HuBERT (Hsu et al., 2021), and Data2Vec (Baevski et al., 2022) for encoding speech stimuli. Millet et al. (2022) used a self-supervised learning model, Wav2Vec2.0, to learn latent representations of the speech waveform similar to human brain. They find that the functional hierarchy of its transformer layers aligns with the cortical hierarchy of speech in the brain and reveals the whole-brain organisation of speech processing with unprecedented clarity. This means that the first transformer

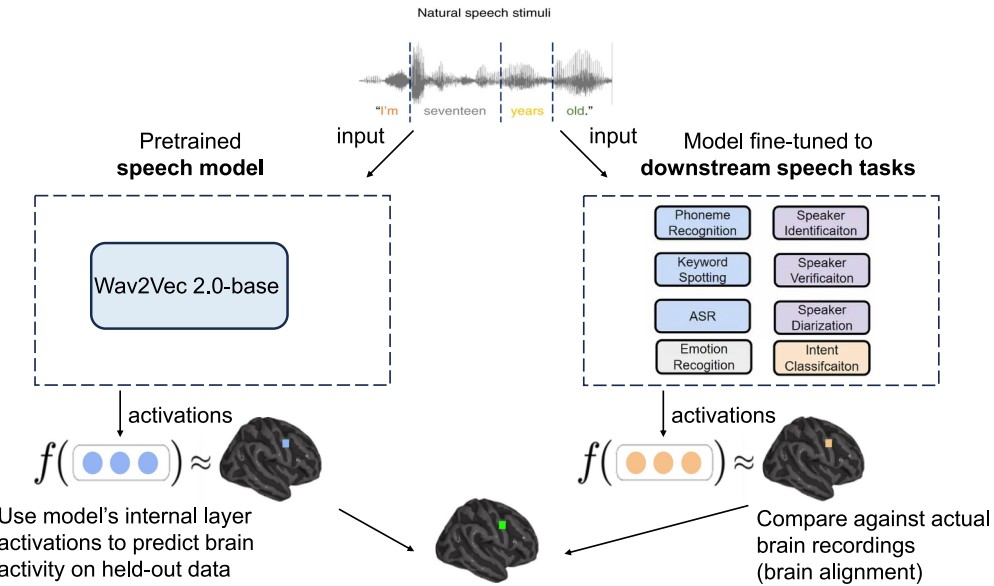

Figure 17: The pretrained Wav2Vec2.0 model and finetuned to eight different downstream speech tasks and their brain alignment. *This figure is adapted from Oota et al. (2023d), and used with permission from the respective authors.*

layers map onto the low-level auditory cortices (A1 and A2), the deeper layers map onto brain regions associated with higher-level processes (e.g. STS and IFG). Vaidya et al. (2022) present the first systematic study to bridge the gap between recent four self-supervised speech representation methods (APC, Wav2Vec, Wav2Vec2.0, and HuBERT) and computational models of the human auditory system. Similar to Millet et al. (2022), they find that self-supervised speech models are the best models of auditory areas. Lower layers best modeled low-level areas, and upper-middle layers were most predictive of phonetic and semantic areas, while layer representations follow the accepted hierarchy of speech processing. Tuckute et al. (2023) analyzed 19 different speech models and found that some audio models derived in engineering contexts (model applications ranging from speech recognition and speech enhancement to audio captioning and audio source separation) produce poor predictions of auditory cortical responses, many task-optimized audio speech deep learning models outpredict a standard spectrotemporal model of the auditory cortex and exhibit hierarchical layer-region correspondence with auditory cortex. Further, Oota et al. (2023c) extended this analysis to more such deep learning based speech models (30 self-supervised speech models). They found that both language and auditory brain areas, are best aligned with intermediate layers in deep learning models. As shown in Figure 16, they also found that speech models better predict early auditory cortex than late language regions. Although pretrained speech models can understand broad aspects of speech in general, the implications of finetuning speech pretrained models for various speech-processing tasks for speech encoding in the brain, remain underexplored.

### 6.5.2 Underlying Speech Properties in Speech Models and Brains

Understanding the reasons behind the observed similarities between speech processing in speech models and brains can lead to more insights into both systems. Recent work Oota et al. (2023d) has found that using a finetuned Wav2Vec2.0 leads to improved brain alignment. In particular, as shown in Figure 17, Oota et al. (2023d) build neural speech taskonomy models for brain encoding and aim to find speech-processing tasks that have the most explanatory capability of brain activation during naturalistic story listening experiments. They find that task-specific (Automated Speech Recognition (ASR), Entity Recognition (ER), Speaker Identification (SID) and Intent Classification (IC)) speech representations lead to a significant improvement in brain alignment compared to the pretrained Wav2Vec2.0 model for specific brain regions. Finetuning on ER, SID and IC leads to the best alignment for the early auditory cortex; finetuning on ASR provides the

best encoding for the auditory associative cortex and language regions. Further, the layer-wise analysis of the effect of each speech task on the alignment with whole brain activity shows that the ASR task is better aligned in middle layers. Similar to language model fine-tuning with brain data in Schwartz et al. (2019), Li et al. (2024b) fine-tuned a pretrained Wav2Vec2.0 model with brain recordings and and found that this induced process modified the language representations, improving the model's performance on downstream tasks from the SUPERB benchmark.

To understand what types of information these language models truly predict in the brain, a very recent study by Oota et al. (2024a) proposes a direct approach by removing a wide range of low-level features from model representations and examining the effect on alignment with both text and speech models. This study reveals that in context of brain reading or listening, both text-based and speech-based models show high brain alignment with late language regions, but speech models trails behind text models. In early visual and auditory regions, both models exhibit high degree of normalized brain alignment. Specifically, text models alignment with late language regions due to brain-relevant semantics, while speech models alignment due to low-level stimulus features. Conversely, text models alignment with early auditory regions mostly due to low-level textual features, while speech models alignment is only partially explained by these features. These findings conclude that speech-based language lack important brain-relevant semantics.

### 6.5.3 Key Takeaways

- **Alignment with Speech Models:**

    1. The functional hierarchy of Transformer layers aligns with the cortical hierarchy of speech processing in the brain.
    2. The lower layers map onto primary auditory areas, while the deeper layers are more predictive of phonetic and semantic information processing.

- **Task-specific speech models lead to improved brain alignment:**

    1. Speech tasks such as emotion recognition (ER), speaker identification (SID), and intent classification (IC) lead to the best alignment in the early auditory cortex.
    2. Fine-tuning on automatic speech recognition (ASR) provides the best alignment in the auditory association cortex and language regions.

- **Speech-based language models lack brain relevant semantics:**

    1. Speech models are useful for modeling early listening: investigate them to learn more about the auditory cortex (AC).
    2. Text models are useful for modeling late language in both listening and reading.

### 6.6 Visual Encoding

### 6.6.1 Alignment Between Vision Models and Brains

Similar to language, in vision, early models focused on independent models of visual processing (object classification) using CNNs (Yamins et al., 2014; Yamins & DiCarlo, 2016). Eickenberg et al. (2017) use CNNs as candidate models to model human brain activity during the viewing of natural images by constructing predictive models based on their different CNN layers and BOLD fMRI activations. They find that there are similarities between the computations of convolutional networks and cognitive vision at the beginning and at the end of the ventral stream object-recognition process. Cichy et al. (2016) further investigate the stages of human visual processing in both time (MEG recordings) and space (fMRI recordings). By comparing these findings with representations derived from deep neural networks (DNNs), the authors demonstrate that DNNs effectively encapsulate the sequential stages of human visual processing. This encompasses the progression from early visual areas towards the specialized pathways of the dorsal and ventral streams, highlighting the DNN's capacity to mirror complex neural processes in both time and space. Despite the effectiveness of CNNs, it is difficult to draw specific inferences about neural information processing using CNN-derived representations from a generic object-classification CNN. Hence, Wang et al. (2019) built

encoding models with individual feature spaces obtained from 21 computer vision tasks. One of the main findings is that features from 3D tasks, compared to those from 2D tasks, predict a distinct part of visual cortex. Recent efforts in visual encoding models, particularly self-supervised models (instance-prototype contrastive learning), operate by taking multiple samples over an image and projecting these through a deep convolutional neural network into a low-dimensional embeddings space (Konkle & Alvarez, 2022). The results show that these self-supervised models achieve parity with the category-supervised models in accounting for the structure of brain responses. Since the human visual system uses two parallel pathways for spatial processing and object recognition, while computer vision systems (CNNs) typically use a single pathway, Choi et al. (2024) developed a dual-stream vision model to mimic human vision. This model uses two branches of CNNs to replicate the dorsal and ventral cortical pathways, aligning with the brain's pathways and suggesting that distinct responses are driven more by visual attention and object recognition goals than by retinal input selectivity.

In a recent study by Matsuyama et al. (2023) on enhancing the precision of models for visual brain encoding, the research focused on two primary questions: (1) How does changing the size of the fMRI training dataset affect prediction accuracy? (2) How does the prediction accuracy across the visual cortex change with the size of the parameters in the vision models? The findings indicate that prediction accuracy improves with increased training sample size, adhering to a scaling law. Similarly, increasing the parameter size of the vision models also leads to improved prediction accuracy, following the same scaling law.

### 6.6.2 Vision Tasks and Brains

How can we push deeper CNN models to capture brain processing more stringently? Continued architectural optimization on ImageNet alone no longer seems like a viable option. Instead of feed-forward deep CNN models, using shallow recurrence enabled better capture of temporal dynamics in the visual encoding models (Kubilius et al., 2019; Schrimpf et al., 2020). Kubilius et al. (2019) proposed a shallow recurrent anatomical network, CORnet, that follows neuro-anatomy more closely than standard CNNs, and achieved the state-of-the-art results on the Brain-score benchmark (Schrimpf et al., 2020). It has four computational areas, conceptualized as analogous to the ventral visual areas V1, V2, V4, and IT, and a linear category decoder that maps from the population of neurons in the model's last visual area to its behavioral choices.

### 6.6.3 Key Takeaways

- **Alignment with Vision Models:** The functional hierarchy of CNN layers aligns with the cortical hierarchy of visual processing in the brain.

- **Task-specific speech models lead to improved brain alignment:** Encoding models using feature spaces from 21 computer vision tasks found that features from 3D tasks predict a distinct part of the visual cortex compared to those from 2D tasks.

- **Brain-Score:** A composite of multiple neural and behavioral benchmarks is used to score any artificial neural network (ANN) based on its similarity to the brain's mechanisms for core object recognition [8].

### 6.7 Multimodal Brain Encoding

Recently Transformer-based multimodal models, which combine pairs of modalities such as language-vision, language-audio, and language-audio-vision, have emerged, offering rich aligned representations compared to single-modality models (i.e. text-only, audio-only or vision-only). Specifically, multimodal Transformers such as CLIP, LXMERT, and VisaulBERT take both image and text stimuli as input and output a joint visio-linguistic representations. Since the human brain perceives the environment using information from multiple modalities, examining the alignment between language and visual representations in the brain by training encoding models on fMRI responses, while extracting joint representations from multimodal models, can offer insights into the relationship between the two modalities.

---

[8]https://www.brain-score.org/

**Single modality stimulus.** Here, participants engage in single modality stimuli, such as watching images or silent videos. Many brain encoding studies have focused on single modality stimuli, while representations are extracted from multimodal models (Oota et al., 2022e; Wang et al., 2023; Tang et al., 2024). Oota et al. (2022e) experimented with multimodal models like CLIP, LXMERT, and VisualBERT and found VisualBERT better predict neural responses than vision-only models such as CNNs and Image Transformers. Similarly, Wang et al. (2023) find that multimodal models like CLIP better predict neural responses in the visual cortex than previous vision-only models like CNNs. This is attributed to the fact that high-level human visual representations encompass semantics and the relational structure of the visual world beyond object identity (Gauthier et al., 2003). Recently, Tang et al. (2024) investigated a multimodal Transformer as the encoder architecture to extract the aligned concept representations for narrative stories and movies to model fMRI responses to naturalistic stories and movies, respectively. Since language and vision rely on similar concept representations, the authors perform a cross-modal experiment in which how well the language encoding models can predict movie-fMRI responses from narrative story features (story → movie) and how well the vision encoding models can predict narrative story-fMRI responses from movie features (movie → story). Overall, the authors find that cross-modality performance was higher for features extracted from multimodal transformers than for linearly aligned features extracted from unimodal transformers.

**Multimodality stimulus.** Here, participants engage with multi-modal stimuli (e.g., watching movies that include audio). Recent studies have built encoding models where multi-modal stimulus representations are extracted using Transformer-based multi-modal models (Dong & Toneva, 2023; Nakagi et al., 2024). Dong & Toneva (2023) present a systematic approach to probe multimodal video Transformer model by leveraging neuro-scientific evidence of multimodal information processing in the brain. The authors find that intermediate layers of a multimodal video transformer are better at predicting multimodal brain activity than other layers, indicating that the intermediate layers encode the most brain-related properties of the video stimuli. A recent study by Nakagi et al. (2024), which used fMRI during the viewing of 8.3 hours of video content, and discovered distinct brain regions associated with different semantic levels, highlighting the significance of modeling various levels of semantic content simultaneously. The video material was meticulously annotated in five distinct semantic categories—speech, object, story, summary, and time/place—employing advanced large language models to derive latent representations. These representations were then used to predict brain activity across the various semantic categories. The authors discovered that the lack of unique variance for Summary and TimePlace is a notable insight, suggesting that merely incorporating these types of information into encoding analyses may not adequately capture higher-level semantic representations in the brain.

### 6.7.1 Key Takeaways

- **Multimodal Integration:** Incorporating linguistic information with other modalities (like vision and auditory) can enhance understanding of how the brain processes complex stimuli.

- **Cross-modal vs. Jointly pretrained models:** Both cross-modal and jointly pretrained multimodal models demonstrate significantly improved brain alignment with language regions and visual regions when analyzed against unimodal video data.

- **Single modality vs. Multimodality stimulus:** Many brain encoding studies have experimented with subjects engaged with single modality stimulus, leaving the full potential of these models in true multi-modal scenarios still unclear.

## 7 Brain Decoding

Brain decoding aims to map neural activations back to the stimulus domain, allowing us to interpret what a person is seeing, hearing, or thinking based on their brain activity, as illustrated in Figure 18. This process is crucial for developing brain-computer interfaces and advancing our understanding of cognitive neuroscience. Unlike brain encoding, which focuses on predicting brain activity from stimuli, brain decoding involves reconstructing the original stimuli from observed neural signals (Glaser et al., 2020).

### 7.1 Problem Formulation

Brain decoding involves learning the mapping between brain activations and stimuli. Early approaches focused on pixel-level mappings using models such as Autoencoders (AEs) and Variational Autoencoders (VAEs), which captured detailed information but often lacked semantic richness. With the advent of large-scale generative models, the focus has shifted to conditional generation, where brain activity representations are used to condition pretrained generative models like generative adversarial networks (GANs), diffusion models, and GPTs. This shift has enhanced the fidelity and meaningfulness of decoded stimuli, enabling more sophisticated and accurate brain decoding systems.

### 7.2 Data Prepossessing

Similar to brain encoding, we utilized several key steps in the data preprocessing phase to ensure robust and accurate brain decoding. Initially, we performed standard preprocessing of the fMRI data, including motion correction, spatial normalization, and smoothing, to mitigate noise and artifacts inherent in the raw recordings.

We utilized paired data from previous sections, consisting of fMRI, Stimuli pairs. This paired data approach ensures that the neural activity is directly aligned with the corresponding stimuli, facilitating more accurate decoding. Following this, we extracted Regions of Interest (ROIs) based on prior neuroanatomical knowledge or functional localization tasks, ensuring that the most informative voxels were selected for subsequent analysis. ROIs focus the analysis on specific brain areas known to be involved in processing the stimuli, reducing the dimensionality and improving the signal-to-noise ratio of the data.

### 7.3 Decoder Architectures

**Pixel-level reconstruction.** Initially, brain decoding was framed as a problem of learning an exact mapping between brain activations and stimuli, often using end-to-end models like Autoencoders (AEs) (Bank et al., 2023; Beliy et al., 2019) and Variational Autoencoders (VAEs) (Kingma & Welling, 2013; Han et al., 2019). These approaches focused on pixel-level mappings, which, while capturing detailed information, were often not semantically meaningful. Early decoding studies employed ridge regression models trained on the most informative voxels or cortex-specific voxels (Pereira et al., 2018; Sun et al., 2019; Oota et al., 2022c), with some using fully connected layers (Beliy et al., 2019) or multi-layered perceptrons (Sun et al., 2019). In some studies where decoding was modeled as multi-class classification, Gaussian Naïve Bayes (Singh et al., 2007; Just et al., 2010) and SVMs (Thirion et al., 2006) were also used. However, despite their ability to recover some detailed information (such as color, shape and location), these methods often fell short of capturing the highly complex non-linear semantic information between the stimulus and the neural responses.

**Semantic reconstruction.** As large-scale generative models evolved, the problem formulation shifted towards conditional generation. In this setup, a representation of brain activity is first obtained and then used as a condition for pretrained generative models, such as GANs (Du et al., 2020; Beliy et al., 2019; Fang et al., 2020), diffusion models (Chen et al., 2023; Takagi & Nishimoto, 2022; Scotti et al., 2024), and GPTs (Tang et al., 2023). This approach emphasizes learning semantic information, effectively capturing high-level information but sometimes lacking fine detail. Conditional generation models leverage vast amounts of pretrained knowledge, allowing them to generate high-quality outputs conditioned on the brain activity representations. This shift has significantly enhanced the fidelity and meaningfulness of the decoded stimuli, paving the way for more sophisticated and accurate brain decoding systems.

**Trade-off between pixel-level and semantic-level reconstruction.** End-to-end methods in brain decoding excel at capturing detailed information such as color, shape, and location due to their direct mapping approach from brain activations to stimuli. These models, often implemented as autoencoders or VAEs, learn a comprehensive transformation that preserves fine-grained details present in the input data. The reconstruction loss functions used during training penalize deviations from the original stimuli, encouraging the model to maintain low-level features like edges and textures.

Figure 18: Scheme for Brain Decoding. Left: Image decoder (Smith, 2013), Right: Language Decoder (Wang et al., 2019). *The left Figure is adapted from (Smith, 2013) and the right Figure is adapted from (Wang et al., 2019). Adapted with permission from the respective authors.*

In contrast, conditional generation frameworks involve a two-stage process where a high-level representation is first extracted from brain activity and then used to condition a pretrained generative model. While this approach leverages powerful generative models like GANs and diffusion models, which are adept at producing realistic and semantically coherent outputs, it tends to abstract away precise pixel-level details in favor of capturing broader semantic information. Consequently, end-to-end methods are particularly suited for tasks requiring detailed reconstructions, whereas conditional generation frameworks excel in generating high-level, semantically accurate representations.

**Hybrid approaches.** Hybrid approaches in brain decoding aim to combine the strengths of both end-to-end methods and conditional generation frameworks to achieve detailed and semantically rich reconstructions. By integrating the direct mapping capabilities of end-to-end models with the high-level semantic generation of conditional frameworks (Scotti et al., 2024; Ferrante et al., 2024; Wang et al., 2024), these approaches can capture fine-grained details while maintaining semantic coherence. Typically, a hybrid approach might first use an end-to-end model to capture detailed low-level features from brain activations and then employ a conditional generative model to refine and enhance these features, ensuring that the final output is both accurate and meaningful. This dual-stage process allows for the preservation of essential details such as color and shape while benefiting from the contextual understanding provided by advanced generative models. Hybrid approaches therefore offer a promising avenue for improving the fidelity and applicability of brain decoding technologies, bridging the gap between detailed reconstruction and high-level semantic interpretation.

## 7.4 Impact of Large Models on Brain Decoding

**Representation learning.** Representation learning has been a crucial step in the evolution of brain decoding. Two primary approaches have been particularly influential: masked autoencoders and contrastive learning.

Masked autoencoders (He et al., 2022) play a vital role in learning low-rank representations by reconstructing missing parts of the input data. In the context of brain decoding, these models are often used for pretraining by masking out some brain voxels and attempting to reconstruct them, thereby learning the underlying representations (Chen et al., 2023; 2024c; Sun et al., 2024). These fMRI representations are then utilized as conditions for downstream conditional generation models, enhancing their ability to produce detailed and accurate reconstructions compared with linear models.

Contrastive learning (Khosla et al., 2020) has emerged as a powerful technique for representation learning by maximizing the similarity between related data points while minimizing the similarity between unrelated ones. This approach has been instrumental in aligning brain activity with corresponding stimuli in a shared embedding space, facilitating more accurate and semantically meaningful decoding. One of the most notable applications of contrastive learning in brain decoding is the CLIP model (Radford et al., 2021). CLIP aligns

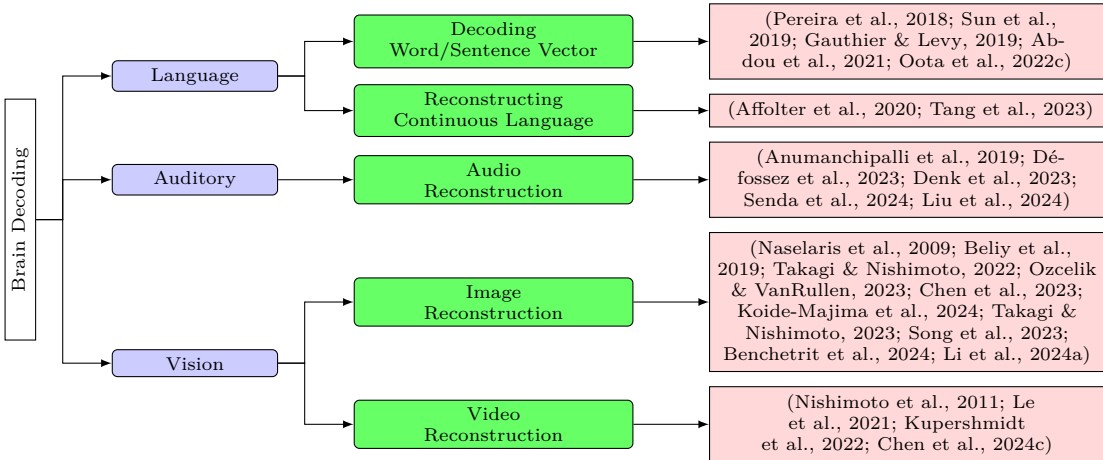

Figure 19: Comprehensive Categorization of Brain Decoding Studies Across Modalities and Tasks. This figure provides a systematic overview of brain decoding research, illustrating the diverse approaches used to reconstruct or classify stimuli from brain activity patterns. The categorization is organized by stimulus modalities (e.g., language, vision, audio, and multimodal) and specific tasks within each modality.

text and images in a shared embedding space, greatly enhancing the decoding of visual stimuli. These models decode brain activity into text descriptions that are then used to generate corresponding images, effectively bridging the gap between linguistic and visual representations. In brain decoding, researchers often align fMRI embeddings with CLIP-based embedding spaces, allowing for more precise and semantically rich reconstructions of visual stimuli from brain data (Chen et al., 2024c; Scotti et al., 2024).

**Large language models (LLMs).** LLMs, particularly models in the GPT series (Brown et al., 2020), have revolutionized language decoding. These models are capable of generating coherent and contextually appropriate text based on brain activity patterns. For instance, instead of merely decoding vector representations of stimuli, recent studies have leveraged LLMs to reconstruct entire sentences or continuous language from fMRI data (Tang et al., 2023; Zhao et al., 2024). This shift from vector-based decoding to full text generation has significantly enhanced the semantic richness and contextual accuracy of the decoded output. The ability of LLMs to model complex language structures and generate text conditioned on neural data has opened new avenues for understanding how the brain processes language, providing practical applications in areas such as communication aids for individuals with speech impairments.

**Diffusion models (Stable Diffusion).** Diffusion models (Ho et al., 2020), particularly those like Stable Diffusion (Rombach et al., 2022), have been pivotal in generating high-fidelity images from brain activity. These models leverage the noise-to-signal transformation process to produce detailed and semantically rich visual outputs. By conditioning these models on brain activity data, researchers have achieved remarkable success in reconstructing images that closely resemble the original stimuli (Scotti et al., 2024; Takagi & Nishimoto, 2022; Chen et al., 2023; 2024c; Ozcelik & VanRullen, 2023; Takagi & Nishimoto, 2023). The high resolution and fidelity of the generated images represent a significant improvement over previous methods, which often struggled to capture fine details and semantic accuracy simultaneously.

**Brain decoding applications.** Figure 19 summarizes the literature on decoding solutions proposed in vision, auditory, and language domains. Table 7 aggregates the brain decoding literature along different stimulus domains such as textual, visual, and audio. The most common setting is to perform decoding to a vector representation using a stimuli of a single mode (visual, text or audio).

Table 7: Summary of Representative Brain Decoding Studies. Here, |S| represents the number of participants in each dataset.

| | Authors | Dataset Type | Lang. | Stimulus Representations | |S| | Dataset |
|---|---|---|---|---|---|---|
| **Text** | (Pereira et al., 2018) | fMRI | English | Word2Vec, GloVe, BERT | 17 | Pereira |
| | (Wang et al., 2020) | fMRI | English | BERT, RoBERTa | 6 | Pereira |
| | (Oota et al., 2022c) | fMRI | English | GloVe, BERT, RoBERTa | 17 | Pereira |
| | (Tang et al., 2023) | fMRI | English | GPT, finetuned GPT on Reddit comments and autobiographical stories | 7 | Moth Radio Hour |
| **Visual** | (Beliy et al., 2019) | fMRI | | End-to-End Encoder-Decoder, Decoder-Encoder, AlexNet | 5 | Generic Object Decoding, ViM-1 |
| | (Takagi & Nishimoto, 2022) | fMRI | | Latent Diffusion Model, CLIP | 4 | NSD |
| | (Ozcelik & VanRullen, 2023) | fMRI | | VDVAE, Latent Diffusion Model | 7 | NSD |
| | (Chen et al., 2024c) | fMRI | | Latent Diffusion Model, CLIP | 3 | HCP fMRI-Video-Dataset |
| | (Li et al., 2024a) | EEG | | CLIP, Diffusion Model | 10 | Things EEG |
| | (Song et al., 2023) | EEG | | CLIP, ViT | 10 | Gifford2022 EEG |
| | (Benchetrit et al., 2024) | MEG | | CLIP | 10 | Things MEG |
| **Audio** | (Défossez et al., 2023) | MEG, EEG | English | MEL Spectrogram, Wav2Vec2.0 | 169 | MEG-MASC |
| | (Gwilliams et al., 2023) | MEG | English | Phonemes | 7 | MEG-MASC |
| | (Denk et al., 2023) | fMRI | English | Music | 5 | Music Genre fMRI |

## 7.5 Linguistic Decoding

Initial brain decoding experiments studied the recovery of simple concrete nouns and verbs from fMRI brain activity (Nishimoto et al., 2011) where the subject watches either a picture or a word. Sun et al. (2019) used several sentence representation models to associate brain activities with sentence stimulus, and found InferSent to perform the best. More work has focused on decoding the text passages instead of individual words (Wehbe et al., 2014). Some studies have focused on multimodal stimuli based decoding where the goal is still to decode the text representation vector. For example, Pereira et al. (2018) trained the decoder on imaging data of individual concepts, and showed that it can decode semantic vector representations from imaging data of sentences about a wide variety of both concrete and abstract topics from two separate datasets. Further, Oota et al. (2022c) propose two novel brain decoding setups: (1) multi-view decoding (MVD) and (2) cross-view decoding (CVD). In MVD, the goal is to build an MV decoder to take brain recordings for any view as input and predict the concept. In CVD, the goal is to train a model that takes brain recordings for one view as input and decodes a semantic vector representation of another view. Specifically, they study practically useful CVD tasks like image captioning, image tagging, keyword extraction, and sentence formation.

To understand application of Transformer models for decoding better, Gauthier & Levy (2019) finetuned a pretrained BERT on a variety of Natural Language Understanding (NLU) tasks to find tasks that lead to improvements in brain-decoding performance. They find that tasks that produce syntax-light representations (representations extracted from a language model trained on randomly shuffled words from corpus samples, thereby eliminating all first-order cues to syntactic structure) yield significant improvements in brain decoding performance.

With the recent development of large language models, rather than decoding stimuli vector representations, some studies have attempted to reconstruct words (Affolter et al., 2020), and continuous language (Tang et al., 2023) from fMRI brain activity.

## 7.6 Auditory Decoding

With the recent advancements of self-supervised speech models and generative AI models, recent studies have largely targeted reconstructing speech/music from brain recordings (Défossez et al., 2023; Denk et al., 2023; Senda et al., 2024). As shown in Figure 20, Défossez et al. (2023) proposed a CLIP-MEG pipeline to align MEG activity onto pretrained speech embeddings and generate speech from a stream of MEG signals. Unlike other methods which are experimented with on narrative speech, Denk et al. (2023) introduced a method for reconstructing music from fMRI brain activity, as shown in Figure 21. Specifically, they proposed a Brain2Music pipeline where the first step involves using fMRI data to predict MuLan$^{music}$

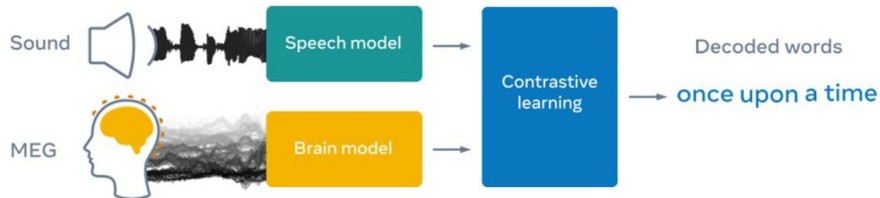

Figure 20: CLIP-MEG pipeline to align MEG activity onto pretrained speech embeddings. *This Figure is adapted from (Défossez et al., 2023), and used with permission from the respective authors.*

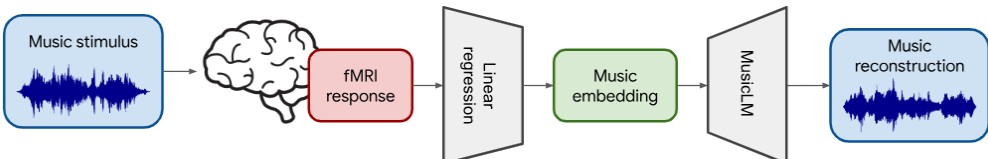

Figure 21: Brain2Music decoding pipeline to reconstruct Music from fMRI brain activity (Denk et al., 2023). *This Figure is adapted from (Denk et al., 2023), and used with permission from the respective authors.*

embeddings (Huang et al., 2022), which are then passed to MusicLM (Agostinelli et al., 2023), is conditioned to generate the music reconstruction, resembling the original music stimulus.

## 7.7 Visual Decoding

A number of methods have been proposed for reconstructing a visual stimulus from brain recordings. Here, we initially address image reconstruction from brain recordings, followed by a discussion on video reconstruction.

### 7.7.1 Image Reconstruction

Before the success of recent generative AI models, researchers have used deep-learning models and algorithms, including generative adversarial networks (GANs) and self-supervised learning models trained on a large number of naturalistic images (Du et al., 2020; Beliy et al., 2019; Fang et al., 2020; Gaziv et al., 2022; Lin et al., 2022). For instance, Beliy et al. (2019) designed a separable autoencoder that enables self-supervised learning in fMRI and images to increase training data. Lin et al. (2022) introduced a framework, Mind Reader, that encoded fMRI signals into a pre-aligned vision-language latent space and used StyleGAN2 (Karras et al., 2020) for image generation. These methods generate more plausible and semantically meaningful images. Several other studies focused on reconstructing personal imagined experiences (Berezutskaya et al., 2020) or application-based decoding like using brain activity scanned during a picture-based mechanical engineering task to predict individuals' physics/engineering exam results (Cetron et al., 2019) and reflecting whether current thoughts are detailed, correspond to the past or future, are verbal or in images (Smallwood & Schooler, 2015).

With the recent success of CLIP and Diffusion models, deep generative models have been gaining attention to generate high-resolution images with high semantic fidelity (Takagi & Nishimoto, 2023; Chen et al., 2023; Scotti et al., 2024; Benchetrit et al., 2024; Song et al., 2023). Takagi & Nishimoto (2023) proposed a method for image reconstruction from fMRI using Stable Diffusion (Rombach et al., 2022), as shown in Figure 22 (left). Their approach involves decoding brain activities to text descriptions and converting them to natural images using Stable Diffusion. Based on a similar philosophy, using a Stable Diffusion model as a generative prior and the pretrained fMRI features as conditions, Chen et al. (2023) reconstructed high-fidelity images with high semantic correspondence to the groundtruth stimuli, as shown in Figure 22 (right). Scotti et al. (2024) proposed a MindEye that can map fMRI brain activity to any high dimensional multimodal latent space, like CLIP image space, enabling image reconstruction using generative models that accept embeddings from this latent space. Different from previous studies, BrainCLIP framework was introduced by Liu et al. (2023) to align fMRI patterns with different modalities (especially from visual and textual modalities)

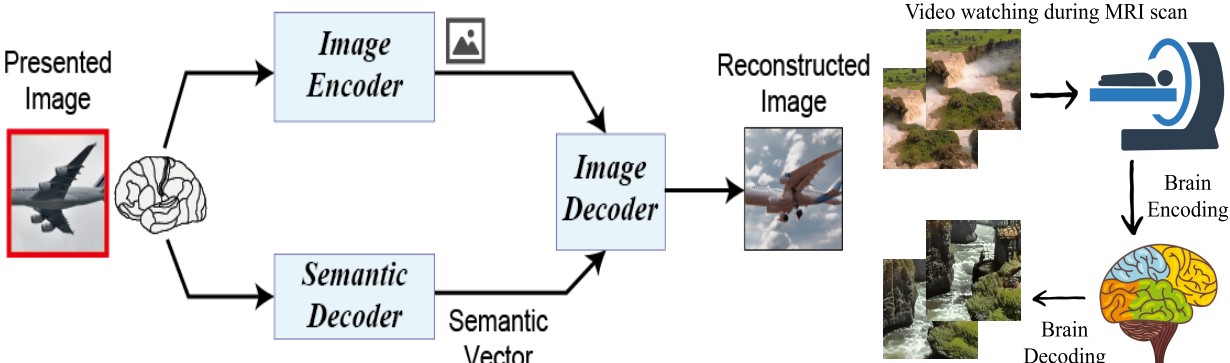

Figure 22: Image reconstruction from fMRI using Stable Diffusion. Left: (Takagi & Nishimoto, 2023), Right: (Chen et al., 2024c). *The Left Figure is adapted from (Takagi & Nishimoto, 2023) and the Right Figure is adapted from (Chen et al., 2024c). Adapted with permission from the respective authors.*

through cross-modal contrastive loss. All these studies have been limited to 2D visual representations. A recent work by Gao et al. (2023) aims to extend the scope of fMRI decoding to 3D representations. Specifically, they introduced Recon3DMind, a groundbreaking task focused on reconstructing 3D visuals from fMRI signals.

Lastly, recent image reconstruction studies have focused on other non-invasive brain recordings such as MEG and EEG rather than fMRI signals. Benchetrit et al. (2024) proposed a CLIP-MEG pipeline to align MEG activity with pretrained visual embeddings and generate images from a stream of MEG signals. Similarly, Song et al. (2023) proposed a CLIP-EEG pipeline to align these two modalities (image and EEG encoders to extract features from paired image stimuli and EEG responses) by constraining their similarity.

### 7.7.2 Video Reconstruction

Unlike static natural images, human visual cortex can process a continuous, diverse flow of scenes, motions, and objects. To recover dynamic visual experience, the challenge lies in the nature of fMRI, which measures blood oxygenation level dependent (BOLD) signals and captures snapshots of brain activity every few seconds. Similar to image reconstruction works, Chen et al. (2024c) present MinD-Video, a two-module pipeline (i.e. CLIP module followed by latent stable diffusion) designed to bridge the gap between image and video brain decoding.

### 7.8 Key Takeaways

- Contrastive learning models like CLIP are popular for aligning stimuli and brain data (fMRI/MEG/EEG) into a common embedding space. This alignment is useful for retrieving or reconstructing the original stimulus from brain data.

- Most decoding studies have focused on the reconstruction of stimuli such as images, videos, text, music, and speech rather than on decoding a subject's imagination. This area remains largely unexplored and would be necessary to achieve an actual mind reading label.

- Unlike brain encoding models, the interpretability of decoding models remains unexplored due to the use of more complex approaches for reconstruction. Addressing this gap is a necessary step to further examine AI models and gain deeper insights into brain functioning.

# 8 Conclusion, Limitations, and Future Trends

In this paper, we surveyed important naturalistic brain datasets, stimulus representations, brain encoding, and brain decoding methods across different modalities. A glimpse of how deep learning solutions throw light on putative brain computations is given. We hope that this systematic organization of recent ideas proposed in the field of cognitive computational neuroscience provides a comprehensive summary to researchers in both the AI and neuroscience communities. Insights gained from recent studies in brain encoding and decoding have significant implications for the fields of AI engineering, neuroscience, and the interpretability of models—some with immediate effects, others with long-term impact.

**AI engineering.** AI engineering generally refers to the design, development, and implementation of artificial intelligence models that simulate human cognitive functions (e.g., vision, language, memory) to enhance our understanding of how these processes are represented and processed in the brain. It also involves designing neural networks inspired by biological neural structures and principles, which can help uncover patterns associated with specific cognitive states or tasks.

The recent brain encoding studies most immediately fit in with the neuro-AI research direction that specifically investigates the relationship between representations in the brain and representations learned by powerful neural network models. This direction has gained recent traction, especially in the domain of language, vision, and speech processing, thanks to advancements in language models (Schrimpf et al., 2021; Goldstein et al., 2022), vision models (Schrimpf et al., 2020) and speech models (Tuckute et al., 2023; Oota et al., 2023c). Furthermore, several recent works most immediately contribute to this line of research by understanding the reasons for the observed similarity in more depth (Merlin & Toneva, 2022; Oota et al., 2024b; Kauf et al., 2024b; Sarch et al., 2024; Oota et al., 2024a). Overall, these studies provide valuable insights for selecting features, enhancing transfer learning, and aiding in the creation of AI architectures that are cognitively plausible.

**Computational modeling in neuroscience.** Researchers have started viewing language models as useful *model organisms* for human language processing (Toneva, 2021) since they implement a language system in a way that may be very different from the human brain but may nonetheless offer insights into the linguistic tasks and computational processes that are sufficient or insufficient to solve them (McCloskey, 1991; Baroni, 2020). These brain encoding studies enable cognitive neuroscientists to have more control over using language models as model organisms of language processing. This approach can also be extended to visual and speech processing, where models in these domains serve as analogous organisms for investigation.

**Model interpretability.** In the long-term, we aspire for these studies on brain encoding and decoding to enhance another research direction that utilizes brain signals to interpret the information processed by neural network models (Toneva & Wehbe, 2019; Aw & Toneva, 2023; Wang et al., 2019; Sarch et al., 2024). Ultimately, our goal is to comprehend the essential and adequate underlying characteristics that result in a meaningful correlation between brain recordings and deep neural network models.

**From humans to animal models.** While this survey focuses on human neuroimaging, it is essential to recognize the significant role of animal models in neuroscience research. Studies involving rodents, primates, and even simpler organisms like C. elegans offer unique insights that complement human studies (Altevogt et al., 2012; Romanova & Sweedler, 2018). Animal models allow for genetic manipulation, invasive techniques, developmental studies, and evolutionary perspectives that are often impossible or unethical in human research. For instance, optogenetics, developed in animal models, enables precise control of neural circuits, providing causal insights into brain function (Romanova & Sweedler, 2018; Metwally et al., 2024).

In the context of encoding and decoding, many fundamental principles were first discovered in animal models. The concept of receptive fields in visual processing, for example, was initially described in cat visual cortex by Hubel & Wiesel (1962). Similarly, our understanding of place cells and grid cells in spatial navigation comes primarily from rodent studies but has implications for human spatial cognition (Moser et al., 2008). When interpreting results, especially in comparative studies, the species-specific nature of neural organization must be considered. However, it is crucial to note that findings from animal studies may not always directly translate to humans due to differences in brain structure and complexity (Kandel et al., 2000). By acknowledging this diversity in neuroscience research, we aim to provide a comprehensive understanding of

neural information processing across species, while maintaining our focus on human neuroimaging studies in this review.

**Ethical considerations and regulatory framework.** It is crucial to understand the ethical landscape surrounding human brain research. The collection and use of neuroimaging data involves significant privacy and ethical considerations. Researchers must adhere to strict protocols to protect participants' personal information, including rigorous anonymization processes to ensure that brain data cannot be linked back to individuals (Eklund et al., 2016). Many funding bodies and journals now mandate open-source data sharing to promote scientific transparency and reproducibility (Poldrack & Gorgolewski, 2014), but this must be balanced with privacy concerns. Before any study begins, it must be approved by Institutional Review Boards (IRBs) or Ethics Committees, which evaluate the potential risks and benefits to participants (Klitzman, 2013). These committees ensure that studies comply with national and international regulations, such as the General Data Protection Regulation (GDPR) in Europe or the Health Insurance Portability and Accountability Act (HIPAA) in the United States (Mourby et al., 2018). Informed consent is a cornerstone of ethical research, requiring that participants fully understand the nature of the study, its risks, and how their data will be used (Nijhawan et al., 2013). As neurotechnology advances, new ethical challenges emerge, such as the potential for brain data to reveal sensitive personal information beyond the scope of the original research question (Ienca & Andorno, 2017). The ongoing dialogue between researchers, ethicists, and policy makers is essential to navigate these complex issues and maintain public trust in neuroscience research.

**Challenges and concerns in neuroscience community.** In the neuroscience community, preprocessing brain data is a critical step in both functional Magnetic Resonance Imaging (fMRI) and Magnetoencephalography (MEG) data analysis. Effective preprocessing ensures that the data are clean, standardized, and suitable for subsequent analyses. However, both fMRI and MEG preprocessing pipelines face a variety of challenges and issues that can impact the quality and reliability of the results. Below is an overview of the common issues encountered in fMRI and MEG preprocessing:

- Issues in fMRI Preprocessing Pipeline:

    1. Motion Artifacts and Correction: Head movements during scanning can introduce significant artifacts, leading to spurious signals that mimic neural activity.
    2. Physiological Noise: Fluctuations in heart rate and breathing can introduce noise into the fMRI signal.
    3. Slice Timing Correction: Variations in the acquisition time of different slices within a single volume can lead to temporal misalignments.
    4. Signal-to-Noise Ratio (SNR): Low SNR can obscure true neural signals, making it difficult to detect meaningful activations.

- Issues in MEG Preprocessing Pipeline:

    1. Environmental Noise and Interference: MEG is highly sensitive to external magnetic and electrical noise, which can contaminate the neural signals.
    2. Source Reconstruction Complexities: Accurately reconstructing the sources of neural activity from sensor data is computationally intensive and prone to errors.
    3. Integration with MRI Data: Combining MEG data with anatomical MRI data for accurate source localization and spatial referencing can be challenging.

Addressing these challenges requires continuous advancements in preprocessing frameworks such as fMRIPrep (Esteban et al., 2019) and MNE (Gramfort et al., 2013), the adoption of standardized preprocessing pipelines, and rigorous quality assurance protocols. By mitigating these preprocessing issues, researchers can improve the accuracy and reproducibility of code, data, and results in neuroimaging studies, thereby deepening our understanding of brain function and its underlying mechanisms. Ultimately, overcoming these preprocessing challenges is essential for advancing the development of cognitively plausible deep neural network (DNN) models and for enhancing practical applications in neuroscience.

**Challenges in selecting evaluation metrics for brain-model alignment.** Selecting the most appropriate metrics for assessing the alignment between computational models and brain data is an active area

of debate in the neuroscience and machine learning communities (Williams et al., 2021; Soni et al., 2024). Each evaluation metric discussed has its own set of assumptions and limitations, which can lead to differing interpretations of model performance.

Recent studies have highlighted the complexities and subjective choices involved in metric selection. For instance, Williams et al. (2021) discuss the limitations of correlation-based metrics, such as PCC and $R^2$ score, suggesting that relying solely on them may provide an incomplete understanding of brain-model alignment. Similarly, debates have arisen over the use of distributional similarity measures like RSA, CKA, and NDS (Marques et al., 2021), which compare the statistical properties of neural and model representations but may not capture fine-grained functional correspondences. Recently, Soni et al. (2024) conducted a comprehensive evaluation of the aforementioned metrics across multiple brain datasets and models to examine the relationships between these measures and the extent to which they support similar conclusions. They found that not only do these measures cause inconsistencies in brain alignment within and across models, but these inconsistencies also undermine some of the conclusions made in the neuro-AI field. Overall, Soni et al. (2024) concluded that evaluation measures were often not chosen based on an analysis of the specific problem at hand but rather for compatibility with the wider field or due to mathematical elegance. However, considerable work may be needed to produce interpretable insights from distinct results obtained by using multiple measures. Further, it is essential to publish complete data and code for reproducibility and transparency. Any claims that rely solely on similarity scores should be revisited with a closer examination of the models.

## 8.1 Future Trends

Some of the future areas of work in this field are as follows.

**Bridging the gap: enhancing deep neural network models for deeper insights into auditory, language and visual processing.** While significant progress has been made in understanding text-based models, understanding the similarity in information processing between visual, speech, and multimodal models versus natural brain systems remains an open area. For instance, Oota et al. (2024a) demonstrate that speech-based language models lack brain relevant semantics in language regions. Therefore, enhancing speech-based language models to align more closely with text-based models could provide valuable insights into language and auditory processing, given that speech is the most ancient form of human language. This suggests a promising direction for future research, aiming to bridge the gap between artificial intelligence models and the complex, multifaceted processes of human cognition.

**Advancing multimodal decoding: the next leap in deep learning accuracy.** Decoding actual multimodal stimuli has become increasingly feasible due to recent advancements in deep learning models dedicated to generation tasks (Rombach et al., 2022; Singer et al., 2022). However, there is still a significant need for further research to enhance the accuracy of these models. This involves not only refining the algorithms and architectures used but also improving the quality and diversity of the datasets on which these models are trained. Advancements in computational power, algorithmic efficiency, and innovative training methodologies are critical for pushing the boundaries of what is possible in multimodal decoding, aiming to achieve more precise, reliable, and nuanced interpretations of complex stimuli.

**Mapping the mind: the effects of brain damage on cognitive capabilities.** We need a deeper understanding of the degree to which damage to different regions of the human brain could lead to the degradation of selective cognitive skills. This exploration requires detailed mapping of cognitive functions to specific brain areas, taking into account the brain's complex network of connections. Studies should investigate not only the immediate effects of brain damage on cognitive skills but also the brain's capacity for reorganization and compensation over time. Ultimately, the goal is to translate these research findings into practical applications, such as more effective cognitive rehabilitation techniques and assistive technologies to improve the quality of life for individuals with brain injuries.

**Towards human-like understanding in ANNs: integrating self-supervised learning and brain-inspired architectures.** How can we train artificial neural networks in novel self-supervised ways such that they compose word meanings or comprehend images and speech like a human brain? Can we model the hierarchical and modular organization of the brain in neural network architectures? This involves cre-

ating networks that reflect the brain's organization, from low-level feature detection to high-level semantic processing, allowing for the integration of information across different modalities. Moreover, how might we integrate dynamic learning strategies, such as curriculum learning, which progressively introduces more intricate tasks to the model? This method emulates how humans naturally progress from understanding straightforward to more complex ideas over time.

**Bridging the language gap in brain-NLP research: the need for multilingual exploration.** An important part of brain-NLP research relies on brain recordings collected from individuals who speak English as their primary language. These studies also utilize experimental stimuli that are presented in the English language. As a result, the majority of neuro-AI research predominantly leverages language models and neural models that have been extensively trained on English text data and brain responses elicited by text or speech in English. However, it is essential to acknowledge the potential variability in study outcomes when extrapolated to languages other than English. For instance, a recent study by Chen et al. (2024b) focuses on recording brain responses while participants fluent in both English and Chinese read several hours of natural narratives in each language. Moreover, the availability of naturalistic stories such as Audiocite.net corpus in French (Felice et al., 2024), and The Little Prince audiobook in Cantonese (Momenian et al., 2024) provides suitable material for recording brain data in other languages. Therefore, it becomes imperative for future research endeavors to delve further into this aspect and investigate how these factors might influence the generalizability of our findings across diverse linguistic contexts.

In addition to the current advancements, there are several potential avenues for future exploration at the intersection of neuroscience and artificial intelligence. One such direction involves leveraging an enhanced understanding of neuroscience to propose modifications to existing artificial neural network architectures, to enhance their robustness and accuracy. Furthermore, an intriguing area for further investigation lies in understanding the brain activity of multilingual, multi-scriptal individuals when processing stimuli in their second language (L2) or script. It remains unclear whether observed brain activity reflects the processing of L2 or the active suppression of their first language (L1) while focusing on L2. This ambiguity underscores the need for further research, particularly in the realm of multilingual multimodal stimuli, to elucidate the underlying mechanisms at play. We hope that this survey motivates research along the above directions.

**Neural population control: the next frontier.** While significant progress has been made in encoding and decoding brain signals, a promising complementary direction involves controlling or stimulating neural activity to influence perception and behavior. This bidirectional brain-computer interface approach could facilitate not only reading but also writing information to the brain, with current methods ranging from optogenetics for precise control of neural circuits (Deisseroth, 2011) to electrical stimulation through neural prosthetics (Basumatary & Hazarika, 2020), and non-invasive techniques such as focused ultrasound and transcranial magnetic stimulation (Polanía et al., 2018). These approaches show particular promise in visual applications, such as restoring sight in blind individuals through direct stimulation of the visual cortex (Beauchamp et al., 2020) or enhancing visual perception through targeted neural modulation (Romei et al., 2010). Additionally, insights from deep-learning-based analyses suggest that targeted stimulation, such as presenting highly activating images, could not only control but also decode neural activity, using brain stimulation as a means to deepen our understanding of brain responses and their similarity to deep network activities (Bashivan et al., 2019; Ponce et al., 2019). Recent efforts using linguistic stimuli to drive language network activity (Tuckute et al., 2024) or use explanation-mediated language stimulus synthesis (Antonello et al., 2024a) show promise in controlling language ROIs. However, several technical challenges persist: the precise targeting of specific neural populations remains difficult, potential tissue damage from invasive methods raises concerns, and our incomplete understanding of neural codes complicates the determination of effective stimulation patterns. Despite these challenges, this direction naturally extends the decoding work discussed in this survey and could eventually lead to fully bidirectional brain-computer interfaces that can both interpret and influence neural activity in real-time (Saha et al., 2021). Such technology holds profound implications for treating neurological conditions, enhancing human capabilities, and advancing our understanding of brain function (Bashivan et al., 2019), provided that future research successfully addresses the needs for more precise and less invasive stimulation methods while carefully considering the ethical implications of direct neural control.

## Acknowledgments

We thank **Dr. Mariya Toneva** from MPI-SWS, Germany, for providing valuable feedback on several versions of this work. We also acknowledge her contributions to the possibility of this survey through her organization of tutorials at various conference venues.

We thank **Prof. Stefan Frank** from Radboud University, Netherlands, for providing review comments on this paper, which helped to further improve its quality after addressing all his suggestions.

We thank all authors for granting approval to use their figures, which has greatly enhanced the quality of this survey.

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
