# OpenReview forum: "Deep Neural Networks and Brain Alignment: Brain Encoding and Decoding (Survey)"
_TMLR — Accepted by TMLR_

### Review · Reviewer_tXDt · 2024-09-08

**Summary Of Contributions:**

This review provides an overview of recent computational neuroscience findings in the fields of especially language, but also audio and vision. Studies are separated into encoding where brain activity is predicted from a stimulus representation, and the inverse decoding where a stimulus representation is predicted from brain activity. The paper provides an overview over several human brain recording datasets in the field, as well as a number of modeling studies working with these datasets along with key results.

**Audience:**

Yes

**Broader Impact Concerns:**

For newcomers to neuroscience, ethical concerns with human data are likely not obvious. The paper should mention the protection of personal information, anonymization of data, open source mandates, institutional review boards, governing laws etc.

**Claims And Evidence:**

Yes

**Requested Changes:**

Please address at least all the Major Weaknesses.

**Strengths And Weaknesses:**

### Strengths:

1. Lists of neural recording datasets, modeling, and tools are undoubtedly useful for newcomers and experts in the computational neuroscience field alike. The tables and figures generally provide a good overview.

2. The paper covers a lot of useful ground in language, audio, and vision as well as encoding and decoding studies.

3. Contents of the paper are generally comprehensible for a machine learning audience, the submission chooses a good level of abstraction, and phrases things in ML-friendly ways (datasets, benchmarks, metrics, etc).



### Major Weaknesses:

1. While I believe most of the paper's content will be useful to an ML audience, the submission could do a much better job of properly contextualizing all of the work.

   1.1 It should be made clear that while vision, audio, and language research are substantial components of neuroscience, they ultimately only make out a small part of the field. Language in particular is a very small part of the current neuroscience research landscape. Decision making, memory, neural disorders, neurogenesis, development, biophysics, glia, somatosensation, and many more avenues of research exist (see e.g. https://www.sfn.org/-/media/SfN/Documents/NEW-SfN/Meetings/Neuroscience-2023/Abstracts/Themes-and-Topics-2023.pdf) and are not mentioned in this review. I recommend an overview paragraph and figure that contextually the contents of this review in the grander neuroscience landscape. It might also be worth mentioning that most of neuroscience does not make use of the kinds of computational models described here; NeuroAI is still a young and small field.

   1.2 Aside from humans, much of neuroscience studies animal brains such as rodents, monkeys, and simpler organisms. This is again not contextualized in the submission but is worth pointing out to a non-specialist audience who might otherwise be confused by confounding claims of how vision works (between e.g. human fMRI and C. elegans papers).

   1.3 The figures are often pieced together from results in other papers which I find useful at later parts of the paper, but some kind of overview figures initially would be quite helpful to ML practicioners I believe. E.g. an overview of how we go from a human to brain recordings to claims about computational principles using neural network models. From there you could also reference the sections in the paper to make clear how everything fits together.


2. Conceptual confusions, incompleteness, and errors.

   2.1 I find it odd that there are "text" datasets. Unlike with language models, we cannot directly beam tokens into the brain. Rather we first have to present them visually or auditorily. Table 1 then does have separate "Audio" datasets which themselves are often spoken text again. What is the distinction here? Similarly, instead of separating "Visual" and "Video" in Table 2, this should imo be "Static images" and "Videos". In the same table, why is the viewing/reading of words a separate category. The organization of datasets is very much unclear to me.

   2.2 I suppose this review focuses on human datasets exclusively, but this is not made clear. For instance, the Majaj & Hong et al. 2015 primate microelectrode recording dataset is widely used in vision research but not listed anywhere. Sometimes the text then only talks about "predict[ing] fMRI recording" (p.14), excluding other data modalities that are present in this review.

   2.3 The language network is typically *not* anatomically defined as implied on page 11, but rather functionally. The Fedorenko studies use the contrast between words and non-words to select language voxels in the brain, with the argument that anatomical regions are not robust between individuals (Lipkin et al. 2022 https://www.nature.com/articles/s41597-022-01645-3).

   2.4 The description of visual cortex focuses only on the ventral stream, completely ignoring dorsal processing (e.g. MT, STS).

   2.5 The evaluation metrics section 5 misses a lot of relevant metrics such as RDM (Kriegeskorte et al. 2008) and CKA (Kornblith et al. 2019) (RDM is only briefly mentioned in the decoding evaluation), as well as distributional comparisons (Marques et al. 2021). The mention of Pearson correlation alone also ignores that there is typically a mapping step in which model activity is regressed to brain recordings (which presents itself with many choices again such as PLS, Ridge, or non-linear mappings altogether).

   2.6 The dichotomy in Figure 10 is not clear to me either. There is probably something I'm not understanding but the text does not clearly describe this, and I don't see how apparently only some modeling studies care about alignment, or how only some studies focus on "Tasks and Brains". E.g., in vision, almost all the papers in the alignment category also focus on some notion of task optimization. Additionally, it would be good to understand how the references are chosen -- these of course have to be incomplete due to the wealth of publications, but is this based on subjective preference or some other criteria?

   2.7 Given the prominence of relating task performance to encoding performance (e.g. Yamins & DiCarlo 2016, Schrimpf et al. 2020, Cao & Yamins 2021 part2), I find it strange that this approach is lumped into the "Indirect approach" group in section 6.3. Lots of papers (cited e.g. in the reviews referenced before) go in this direction and this might be of particular interest to the ML community. There is also little discussion of hierarchical processing between deep nets and brains which has been especially observed in vision.


3. No discussion of shortcomings and disagreements in the community

   3.1 Concerns with fMRI are widely discussed in the neuroscience community, but not at all present in this submission. For instance, using "brain activation of voxels directly" (p. 10) implies that the BOLD signal is a simple hardware readout like the voltage measurement in electrode recordings -- but in reality, this only follows after an intricate, error-prone, and not completely standardized processing pipeline. Efforts are only slowly being made to improve this situation (e.g. Esteban et al. 2019 https://www.nature.com/articles/s41592-018-0235-4). There is a paragraph on MEG preprocessing on p.16, recommend something similar for fMRI, and a discussion of issues for both.

   3.2 It is very much unclear right now which metric(s) we should use for testing models' brain alignment and there is lots of disagreement and discussion about this in the community; some of these issues with metrics and subjective choices should make it into this review (e.g. Williams et al. 2021, Soni et al. 2024 https://www.biorxiv.org/content/10.1101/2024.08.07.607035)



### Minor Weaknesses:

* Several grammar issues and missing words, recommend running a grammar checker

* "Two important models related to how brain represents information are" -> these are model types, not individual models

* "Previous surveys (Cao et al., 2021; Karamolegkou et al., 2023) have primarily focused on brain encoding and decoding studies for language stimuli." -- this is not correct. Cao & Yamins 2021 do not focus on language stimuli but rather take a general view on mechanistic modeling in systems neuroscience

* "A good section of the DNN community is interested in neuroscience and psycholinguistics." -- do you mean "interested in neuroscience and cognitive science", i.e. internal and external measurements of the brain/behavior? Although I am somewhat confused if this review is meant to primarily focus on language, or be more broadly about also sensory processing

* CCN focusing on only improving model accuracy and interpretability (p.4) is a bit narrow. Overall this community tries to understand the brain in computational terms which also involves falsifying existing models, and making new (model-guided) predictions (e.g. Tuckute et al. 2024 for language)

* claiming that "More recently, brain encoding research has shifted [...], examining how the amount of context affects the brain predictivity" is too grand for only a handful of studies. There are many other flavors of studying the human language system using deep nets

* Table 1 is missing several language datasets, even though they are cited elsewhere in text. E.g. Pereira et al. 2018, Fedorenko et al. 2016, Blank et al. 2014, Tuckute et al. 2024

* Figure 7's caption does not really make sense in the context of what the text uses the figure for (showing the locations of the language network)

* Citations for 2V2 accuracy and Pearson correlation (p.12) point to somewhat random papers that *use* those metrics, but they neither define the metrics nor are they the first CCN papers to use them.

* "TR" is first used on page 14, but only described on page 15. What a TR is will not be obvious to the ML community

* parts of Figure 11 look like they are adapted from Schrimpf et al. 2021, recommend acknowledging in the caption

* Table 4 references the wrong datasets for several of the papers. E.g. I'm not sure what dataset "Imagine a situation with the
noun" is for Gauthier & Levy 2019; the dataset for Goldstein et al. 2022 is just called "ECoG"; and Kubilius et al. 2019 according to this submission uses the Algonauts dataset when in reality this was using non-human primate recordings.

* I caution against emphasizing that "middle layers of language models align [best]" -- this result is very dependent on model architecture and looks very different depending on which model you use

* There is quite some bias in the paper towards publications from Oota and the Toneva group (and Huth acolytes broadly). This is of course natural as the author likely comes from that sphere, but especially in a review paper I encourage you to make more of an effort to give a broader overview of the field outside your immediate bubble. Table 3 for instance ignores several language encoding studies with HRF delays that are already cited elsewhere in the paper; similarly for Table 4; especially in pages 20 through 26 the submission reads like a tour of publications from this bubble rather than a broad overview of the field.

---

> ### Author Response · Authors · 2024-10-14
>
> *We thank the reviewer for their strong positive, insightful and valuable comments and suggestions which are crucial for further strengthening our manuscript.*
>
> **Q1.  I recommend an overview paragraph and figure that contextually the contents of this review in the grander neuroscience landscape.**
>
> Thank you for your question. Following the reviewer's suggestion, we have now included an overview paragraph on "The general context of computational models of brain function'' in the Introduction, on page 2.
>
> **The general context of computational models of the brain function.**
> The interdisciplinary field of Cognitive Computational Neuroscience (CCN) aims to combine findings from cognitive science, neuroscience, and computational modeling to understand how the brain represents and mediates cognitive processes. Computational models are pitched at three different scales: microscale, mesoscale, and macroscale. Microscale computational models are typically based on detailed biophysical modeling of the neurons and are aimed at deciphering the fundamental principles of neural computation  *[Dayan et al. 2003]*. Mesoscale models are designed to link the detailed dynamics of individual neurons and the large-scale organization of brain networks, typically using data from neuroimaging studies of the whole brain *[Shine et al. 2021]*. However, macroscale computational models focus on understanding brain function at a larger systems-level, typically involving networks of brain regions and their interactions. The models discussed in the review belong to the mesoscale category. The cognitive architecture of the brain exhibits an intriguing interplay of structure and function that underpins human cognition, allowing the integration of sensory information (vision, auditory and other senses), perceptual processing, learning, memory, attention, motor processes, decision-making, emotions, and other higher-order cognitive processes. From the panoply of cognitive processes, the models reported here address only a small subset comprising perceptual processes related to vision and auditory domains, as well as behaviors related to reading, listening, speech, and language. Thus the new field of neuro-AI is still in its infancy, modeling the brain function in limited cognitive domains using the latest advances in deep learning for understanding how the brain encodes information and how to decode this back!
>
> *[Dayan et al. 2003] Theoretical neuroscience: computational and mathematical modeling of neural systems, Journal of Cognitive Neuroscience*
>
> *[Shine et al. 2021] Computational models link cellular mechanisms of neuromodulation to large-scale neural dynamics, Nature neuroscience*
>
> **Q2. Aside from humans, much of neuroscience studies animal brains such as rodents, monkeys, and simpler organisms. This is again not contextualized in the submission.**
>
> Thank you for pointing this out. We have now included a paragraph on “From Humans to Animal Models” in the discussion, on Page 41.
>
> **From Humans to Animal Models.** While this survey focuses on human neuroimaging, it is essential to recognize the significant role of animal models in neuroscience research. Studies involving rodents, primates, and even simpler organisms like C. elegans offer unique insights that complement human studies *[Altevogt et al. 2012][Romanova et al. 2018]*. Animal models allow for genetic manipulation, invasive techniques, developmental studies, and evolutionary perspectives that are often impossible or unethical in human research. For instance, optogenetics, developed in animal models, enables precise control of neural circuits, providing causal insights into brain function *[Romanova et al. 2018][Metwally et al. 2024]*.
>
> In the context of encoding and decoding, many fundamental principles were first discovered in animal models. The concept of receptive fields in visual processing, for example, was initially described in the cat visual cortex by *[Hubel and Wiese 1962]*. Similarly, our understanding of place cells and grid cells in spatial navigation comes primarily from rodent studies but has implications for human spatial cognition *[Moser et al. 2008]*. When interpreting results, especially in comparative studies, the species-specific nature of neural organization must be considered. However, it is crucial to note that findings from animal studies may not always directly translate to humans due to differences in brain structure and complexity *[Kandel et al. 2000]*. By acknowledging this diversity in neuroscience research, we aim to provide a comprehensive understanding of neural information processing across species, while maintaining our focus on human neuroimaging studies in this review.
>
> *[Altevogt et al. 2012] International animal research regulations: impact on neuroscience research: workshop summary, National Academies Press.*
>
> *[Romanova et al. 2018] Animal model systems in neuroscience, ACS Chemical Neuroscience.*

---

> > ### Author Response · Authors · 2024-10-14
> >
> > **Q3. An overview of how we go from a human to brain recordings to claims about computational principles using neural network models.**
> >
> > Thank you for this valuable suggestion. In response to the reviewer's feedback, we have added a new overview figure (Figure1)  on page 2.
> >
> > * Figure 1: This plot summarizes overall encoding and decoding pipelines with different neuroimaging modalities (fMRI, MEG and EEG), Stimulus modalities (language, audio, visual, and multimodal), and Tasks (reading, listening, watching static images or videos, with or without audio).
> > * Further, the pipelines also incorporate stimulus representations obtained from different types of DNN, mapping the DNN and Brain representations via linear or non-linear models, and evaluation measures estimating the performance of encoding/decoding models. Visualization tools facilitate intuitive presentation of the results.
> >
> > **Q4. The organization of datasets is very much unclear to me.**
> >
> > Thank you for pointing this out. Based on reviewer’s suggestion, we have updated the categorization into audio-based, vision-based, and multimodal datasets. Please refer to Section 3, on page 9.
> >
> > * **Audio-based datasets** involve passive listening, where the primary modality is auditory, such as spoken utterances or narrative stories.
> >
> > * **Vision-based datasets** include passive reading or viewing, with the visual modality encompassing static images, videos (without accompanying audio), and text (lexical items like words or sentences presented on a screen).
> >
> > * **Multimodal datasets** consist of stimuli where both vision and audio modalities are presented together.
> >
> > This new categorization better reflects the sensory modalities involved and provides clearer distinctions between dataset types.
> >
> > **Q5. I suppose this review focuses on human datasets exclusively, but this is not made clear.**
> >
> > Thank you for bringing up this point.
> > * We would like to clarify that in this review, we focus exclusively on neuroscience datasets involving human participants, particularly those with non-invasive recordings such as fMRI, MEG, EEG and ECoG.
> > * Furthermore, We acknowledge that non-human primate datasets, like the primate microelectrode recording dataset proposed in *[Majaj et al. 2015]*, are highly valuable and widely used in vision research. However, given the scope of our current review paper, our focus is limited to human brain datasets.
> > * We have now added this paragraph on page 5 in the revised draft.
> >
> > * We updated statements like "predicting fMRI recordings" to "predicting neural recordings" or "predicting brain activity" to encompass a wider range of data types.
> >
> > *[Majaj et al. 2015] Simple learned weighted sums of inferior temporal neuronal firing rates accurately predict human core object recognition performance, Journal of Neuroscience.*
> >
> > **Q6. The language network is typically not anatomically defined as implied on page 11, but rather functionally.**
> >
> > Thank you for pointing this out.
> > * We have updated the text in line with the reviewer's suggestion to clarify this point. The language network refers to brain regions involved in language processing.
> > * Studies such as those by *[Fedorenko et al. 2014] [Lipkin et al. 2022]* use contrasts like words versus non-words to functionally select language-specific voxels, highlighting the variability of anatomical regions across individuals. As a result, the language network is typically defined functionally rather than anatomically.
> >
> > **Q7. The description of visual cortex focuses only on the ventral stream, completely ignoring dorsal processing**
> >
> > * We acknowledge that our original description of the visual cortex overly focused on the ventral stream and did not sufficiently address the role of dorsal processing regions such as MT and STS. In the revised version, we updated discussion of both streams in the revised draft.
> > * After initial processing in the early visual cortex, visual information is transmitted along two pathways (dorsal and ventral streams) for higher-level interpretation.
> > * The ventral pathway, often referred to as the ``what'' pathway, extends from the EVC to the inferior temporal cortex (IT). This pathway is crucial for object recognition and form representation. It handles the processing of detailed visual information, enabling us to identify and understand complex stimuli such as faces, objects, and scenes.
> > * In contrast, the dorsal pathway, known as the "where" pathway, projects from the EVC to the posterior parietal cortex. It is primarily responsible for spatial awareness and motion detection. The dorsal stream processes information about the location and movement of objects in space, facilitating the coordination of actions like reaching, grasping, and navigating through the environment.
> > * In summary, the Higher visual cortex (HVC) includes regions from both the dorsal and ventral pathways, each specializing in different aspects of visual processing.
> > * We have also updated the details of the visual ROI.

---

> > > ### Author Response · Authors · 2024-10-14
> > >
> > > **Q8. The evaluation metrics section 5 misses a lot of relevant metrics such as RDM and CKA, as well as distributional comparisons.**
> > >
> > > Thank you for highlighting the missing evaluation metrics.
> > > * In the revised version, we have now updated the evaluation metrics section to include important metrics such as Representational Dissimilarity Matrix (RDM), Centered Kernel Alignment (CKA), and Normalized Distribution Similarity, which are crucial for evaluating model-to-brain alignment.
> > > * We have added new subsections 5.3 and 5.4 on pages 17 and 18.
> > > * Additionally, we have expanded the discussion of the Pearson correlation metric, where model activity is typically regressed onto brain recordings using methods such as Partial Least Squares (PLS), Ridge regression, and non-linear mappings. These updates will better reflect the diversity of approaches used in aligning model activity with brain recordings.
> > >
> > > **Q9. The dichotomy in Figure 10 is not clear to me either.**
> > >
> > > Thank you for raising this concern.
> > > * We agree with the reviewer that all of these studies perform alignment, whether based on pretrained model representations or task-based representations. As a result, we have revised Figure 10 by removing the "alignment" term, as every category engages in some form of brain alignment.
> > > * The motivation behind Figure 10 is to illustrate that, while many studies do indeed focus on both task optimization and brain alignment, there exists a conceptual division between research that primarily emphasizes alignment of brain representations (e.g., comparing neural data with pretrained model activations) and research that is more centered on task-based performance (e.g., fine tuned to a specific task without explicitly aligning with neural data).
> > > * Our categorization is based on the nature of the representations used—either from pretrained models or from task-specific models. Most vision encoding studies, for instance, focus on representations derived from CNN models and explore visual hierarchy by correlating CNN layers with brain recordings. On the other hand, several studies specifically concentrate on task-based representations, such as 2D/3D segmentation or depth estimation. Similar approaches are also found in language and speech domains. For this reason, we categorize the studies into two groups: pretrained models and brain alignment, and task-specific models and brain alignment.
> > >
> > > **Q10. Given the prominence of relating task performance to encoding performance, I find it strange that this approach is lumped into the "Indirect approach" group in section 6.3. There is also little discussion of hierarchical processing between deep nets and brains which has been especially observed in vision.**
> > >
> > > Thank you for pointing this out.
> > > * We acknowledge that categorizing the approaches solely under the "Indirect Approach" does not fully capture their significance and methodological distinctiveness. In response to your comment, we have restructured this category to "Task-Performance Aligned Approaches."  We included [Yamins & DiCarlo 2016], [Schrimpf et al. 2020], and [Cao & Yamins 2021], under this new category to highlight their focus on aligning task performance with neural encoding.
> > > * As per your suggestion, we have included a new subsection in Section 6.3 titled "Hierarchical Processing in DNNs and the Human Brain." This subsection delves into how hierarchical structures in deep neural networks (DNNs) mirror the hierarchical processing observed in the human brain, particularly within the visual cortex and during speech and language comprehension.
> > >
> > > **Q11. There is a paragraph on MEG preprocessing on p.16, recommend something similar for fMRI, and a discussion of issues for both.**
> > >
> > > Thank you for raising this concern.
> > >
> > > * We have now included fMRI preprocessing pipeline similar to MEG preprocessing pipeline in the revised draft.
> > > * Based on the reviewer’s suggestion, we have now included a paragraph titled "Challenges and Concerns in the Neuroscience Community" in the discussion section on page 42.
> > >
> > > **Q12. Issues about metrics disagreements and subjective choices should make it into this review**
> > >
> > > Thank you for your question and for suggesting several recent papers.
> > > * We have now included a paragraph titled "Challenges in Selecting Evaluation Metrics for Brain-Model Alignment" in the discussion section on page 43, where we highlight the challenges in selecting complexity metrics. We have also cited these papers along with several others.

---

> > > > ### Author Response · Authors · 2024-10-14
> > > >
> > > > **Minor**
> > > >
> > > > **Q13. Previous surveys (Cao et al., 2021; Karamolegkou et al., 2023) have primarily focused on brain encoding and decoding studies for language stimuli.**
> > > >
> > > > Thank you for pointing this out.
> > > > * We corrected the sentences in the revised draft.
> > > >
> > > > **Q14. CCN focusing on only improving model accuracy and interpretability (p.4) is a bit narrow. Overall this community tries to understand the brain in computational terms which also involves falsifying existing models, and making new (model-guided) predictions**
> > > >
> > > > * We have included this sentence in the revised draft.
> > > >
> > > > **Q15. Table 1 is missing several language datasets, even though they are cited elsewhere in text. E.g. Pereira et al. 2018, Fedorenko et al. 2016, Blank et al. 2014, Tuckute et al. 2024**
> > > >
> > > > * We have now updated Table 2 and Table 3 to include the previously missing datasets, ensuring consistency with the text.
> > > >
> > > > **Q16. Figure 6's caption does not really make sense in the context of what the text uses the figure for**
> > > >
> > > > * We have now updated the caption for Figure 6 in the revised draft.
> > > >
> > > > **Q17. "TR" is first used on page 14, but only described on page 15.**
> > > >
> > > > * We have addressed this typo.
> > > >
> > > > **Q18. parts of Figure 11 look like they are adapted from Schrimpf et al. 2021, recommend acknowledging in the caption**
> > > >
> > > > * We have acknowledged this in the revised draft.
> > > >
> > > > **Q19. Table 4 references the wrong datasets for several of the papers**
> > > >
> > > > * Thank you for pointing this out. We have corrected the table with the relevant datasets and made the following updates:
> > > >
> > > > * Gauthier & Levy (2019) used the Pereira (2018) dataset.
> > > > * The dataset from Goldstein et al. (2022) is now labeled as Goldstein2022 ECoG.
> > > > * Kubilius et al. (2019) utilized both primate and human behavioral data. Since our focus is on human recordings, we have included the dataset as human behavioral data.
> > > >
> > > > **Q20. There is quite some bias in the paper towards publications from Oota and the Toneva group (and Huth acolytes broadly).**
> > > >
> > > > * We have highlighted other publications and removed several from these groups to ensure a balanced representation of papers from all groups.
> > > >
> > > > **Q21. Paper should mention the protection of personal information**
> > > >
> > > > * We have now included a paragraph titled "Ethical Considerations and Regulatory Framework." in the discussion section on page 41

---

> > > > > ### Author Response · Authors · 2024-11-05
> > > > >
> > > > > Dear Reviewer tXDt,
> > > > >
> > > > > We appreciate your feedback and effort you have invested in evaluating our work.
> > > > >
> > > > > In response to your insightful comments, we have addressed both the major and minor weaknesses you highlighted. We believe these revisions significantly enhance the clarity and completeness of the paper.
> > > > >
> > > > > We kindly request you to review our responses based on these revisions. Should you have any further questions or suggestions, we are more than willing to provide additional information or clarification as needed.
> > > > >
> > > > > Thank you for your valuable input and support.

---

> > > > > > ### Author Response · Authors · 2024-11-11
> > > > > >
> > > > > > Dear Reviewer tXDt,
> > > > > >
> > > > > > As the author-reviewer discussion phase is set to close on November 16th, we want to express our gratitude for your engagement and valuable feedback. We kindly request you to review our responses based on recent revisions. Should you have any further questions or suggestions, we are more than willing to provide additional information or clarification as needed.
> > > > > >
> > > > > > Thank you for your valuable input and support.
> > > > > >
> > > > > > Best regards,
> > > > > >
> > > > > > The Authors

---

> > > > > > ### Comment · Reviewer_tXDt · 2024-11-18
> > > > > >
> > > > > > Thank you very much for thoroughly addressing my comments. I wanted to point out a handful of items that would help finalize the paper:
> > > > > >
> > > > > > 1. Thank you for addressing the contextualization of the work. The new Figure 1 is very helpful in this regard. I would just add the word "human" somewhere in the abstract (e.g. "focusing on human fMRI studies").
> > > > > >
> > > > > > 2. Thank you also for addressing the conceptual points. There are a few errors left. For instance on page 14, PLS is attributed to Marques et al. 2021 which very explicitly did *not* use PLS but rather introduce a new set of distributional comparisons. Attributing Pearson correlation to Caucheteux et al. is also very weird to me. I believe PLS (and linear regression more broadly) was first widely introduced for model-to-brain comparisons in the Yamins*, Hong*, et al. 2014 paper and then further popularized by Brain-Score (Schrimpf*, Kubilius*, et al. 2018).
> > > > > > Section 5.3 has a typo ("ad" instead of "and"). I recommend moving the distributional comparisons from Marques et al. 2021 in here.
> > > > > >
> > > > > > 3. I take some issue with Figure 13: the claim is that middle layers consistently best map onto language brain regions, but this very much depends on the model (Schrimpf et al. 2021, Fig. S10, for instance shows a preference towards late layers; from unpublished results I have seen that the correspondence is even more chaotic for different model types). If you insist on keeping this figure, I recommend narrowing the claim, e.g. saying that middle layers work out for this one type of model, but not necessarily in general.
> > > > > >
> > > > > > 4. For the new "Neural population control", you might want to include Ponce*, Xiao*, Schade*, et al. 2019 (https://pubmed.ncbi.nlm.nih.gov/31051108/) and Tuckute et al. 2024 (https://www.nature.com/articles/s41562-023-01783-7).

---

> > > > > > > ### Author Response · Authors · 2024-11-18
> > > > > > >
> > > > > > > *We appreciate the reviewer's feedback and are confident it has enhanced the quality of our paper.*
> > > > > > >
> > > > > > > **Q1. I would just add the word "human" somewhere in the abstract (e.g. "focusing on human fMRI studies").**
> > > > > > >
> > > > > > > Thank you for this suggestion. We have incorporated this text into the abstract in the revised draft.
> > > > > > >
> > > > > > > **Q2. Citations related to PLS and Pearson Correlation**
> > > > > > >
> > > > > > > Thank you for this suggestion. We have now correctly updated the citations for [Yamins et al., 2014] and [Schrimpf et al., 2018] regarding PLS. We have removed the citations related to Pearson correlation, as it is a standard metric.
> > > > > > >
> > > > > > > **Q3. I recommend moving the distributional comparisons from Marques et al. 2021 in here.**
> > > > > > >
> > > > > > > Section 5.1 enumerates metrics for brain encoding. Distributional comparisons-based metrics such as RSA, CKA, NDS are all grouped together in section 5.3 and these are used for evaluating both encoding and decoding models. So we feel keeping them in a separate section would be better. We're not sure if reviewer has any other reason for asking us to move NDS (Marques et al. 2021) into Section 5.1 (Metrics for Encoding models)?
> > > > > > >
> > > > > > > We would appreciate further clarification from the reviewer regarding this suggestion.
> > > > > > >
> > > > > > > **Q4.  I recommend narrowing the claim, e.g. saying that middle layers work out for this one type of model, but not necessarily in general.**
> > > > > > >
> > > > > > > Thank you for this valuable suggestion. We have refined this claim by including the following text: "However, the higher brain alignment in the middle layers is influenced by the model's architecture type (encoder, decoder, or encoder-decoder) rather than being a universal characteristic."
> > > > > > >
> > > > > > > **Q5. Citations regarding Neural Population Control**
> > > > > > >
> > > > > > > We have included the citations [Ponce et al., 2019], [Tuckute et al., 2024], and [Antonello et al., 2024] in the Neural Population Control subsection.

---

### Review · Reviewer_Qyav · 2024-09-10

**Summary Of Contributions:**

The authors (i) survey representations of language, vision, and speech stimuli; (ii) present a summary of neuroscience datasets used to learn deep learning based encoding and decoding models; (iii) provide an outlook about future trends in the connected fields of cognitive and computational neuroscience.

**Audience:**

No

**Claims And Evidence:**

Yes

**Requested Changes:**

If I should recommend this survey for acceptance at TMLR, I would like the authors to perform the following changes:

- Fix all grammar mistakes and improve the writing in general.
- Follow maths notation guidelines: Give better explanations for the notation and clearly define variables. I cannot follow the current notation. It is very sloppy and highly unusual. Alternatively, drop all notation and explain what you are doing in natural language. Math notation should help the reader understand what is going on and not make it more difficult.
- Improve the figures by increasing their resolution and acknowledging copyright ownership! Many figures take parts from other works but neither do the authors mention this anywhere nor is the required copyright information provided. This is concerning and cannot be published as is. Also, it seems that some figure panels are just copied from other papers or blog posts; thus their poor resolution.

**Strengths And Weaknesses:**

**Strenghts**

This paper attempts to comprehensively review the literature regarding encoding and decoding models in the connected fields of cognitive and computational neuroscience. In addition, the authors give a comprehensive overview of the datasets used to learn such encoding and decoding models in the fields of language, vision, and speech neuroscience.

**Weaknesses**

- This paper is very poorly written. In addition to issues with writing style, there are many grammar mistakes.
- The maths notation is sloppy and unusual. Examples: (1) The way cosine distance is denoted on page 13 is highly unusual. I have never seen cosine distance denoted like that elsewhere. (2) The notation in the paragraph on **brain encoding and encoding** is not readable. I don't understand the capital letters that the authors use. What do they refer to? Spaces? Matrices? Vectors? Scalars? Sets? While there is not a single right math notation and researchers have different styles, there are guidelines on how to properly use notation. I encourage the authors to follow those guidelines. If you want to use math notation, then please make sure that any reader can follow. Else, just skip it and explain the maths in natural language.
- Many figures are pixelated and sloppy (e.g., almost every part of Figure 1 is copied from a different paper but neither is this acknowledged anywhere in the main text or the caption nor do the authors provide any copyright information which is concerning; Figure 2 is taking up way too much space, same goes for Figure 4, 6, and 7 of which many parts have poor resolution and consume unnecessarily much space; Figure 23 has very low resolution and is hard to read).
- Some terms to describe the literature are unusual (such as "AI engineering")

---

> ### Author Response · Authors · 2024-10-14
>
> *We thank the reviewer for their strong positive, insightful and valuable comments and suggestions which are crucial for further strengthening our manuscript.*
>
> **Q1. Proofreading**
>
> Thank you for raising this concern. We have thoroughly proofread the paper and corrected the grammatical issues in the revised version.
>
> **Q2. The maths notation is sloppy and unusual.**
>
> Thank you for pointing this out and your valuable suggestions.
> * We have now clearly defined all notations throughout the paper.
> * The cosine distance function is explicitly defined using standard mathematical notation.
> * Additionally, the notation in the paragraph on brain encoding and decoding has been thoroughly clarified.
> * We have also provided a table that specifies what each symbol refers to, such as matrices, sets, or functions.
>
> **Q3. Many figures are pixelated and sloppy. Copyright issues?**
>
> Thank you for raising this concern.
> * We would like to clarify that we contacted the relevant authors and, with their prior permission, included the figures in the draft.
> * In the revised version, we have clearly acknowledged each figure. Following the reviewer's (tXDt) suggestion, we have replaced Figure 1 with an overview figure.
> * To save space, we have merged Figures 4 and 5 into a single figure.
> * Additionally, we have resolved the resolution issues for Figure 23 and other figures.
>
> **Q4. Some terms to describe the literature are unusual (such as "AI engineering")**
>
> Thank you for this question.
>
> * We have now included an additional description of AI engineering in the revised version.
>
> AI engineering generally refers to the design, development, and implementation of artificial intelligence models that simulate human cognitive functions (e.g., vision, language, memory) to enhance our understanding of how these processes are represented and processed in the brain. It also involves designing neural networks inspired by biological neural structures and principles, which can help uncover patterns associated with specific cognitive states or tasks.

---

> > ### Author Response · Authors · 2024-11-05
> >
> > Dear Reviewer Qyav,
> >
> > We appreciate your feedback and effort you have invested in evaluating our work.
> >
> > In response to your insightful comments, we have addressed the requested changes and the weaknesses you highlighted. We believe these revisions significantly enhance the clarity and completeness of the paper.
> >
> > We kindly request you to review our responses based on these revisions. Should you have any further questions or suggestions, we are more than willing to provide additional information or clarification as needed.
> >
> > Thank you for your valuable input and support.

---

> > > ### Author Response · Authors · 2024-11-11
> > >
> > > Dear Reviewer Qyav,
> > >
> > > As the author-reviewer discussion phase is set to close on November 16th, we want to express our gratitude for your engagement and valuable feedback. We kindly request you to review our responses based on recent revisions. Should you have any further questions or suggestions, we are more than willing to provide additional information or clarification as needed.
> > >
> > > Thank you for your valuable input and support.
> > >
> > > Best regards,
> > >
> > > The Authors

---

### Review · Reviewer_HXgw · 2024-10-19

**Summary Of Contributions:**

This manuscript provides an overview over how deep neural networks are used to better understand the brain by training the DNNs to either encode various stimuli (audio, images etc.) into brain activity or, in reverse, decode such stimuli from brain activity (as measured by EEG, MEG, fMRI etc.). It provides a fairly comprehensive overview over the motivation and typical methods for such studies, and discusses a wide range of recent results.

**Audience:**

Yes

**Claims And Evidence:**

Yes

**Requested Changes:**

Please revise the MEG preprocessing sections as described above.

[Audiocite.net: A Large Spoken Read Dataset in French - 2024_Felice_audiocite.pdf](https://hal.science/hal-04533994/document) would be a dataset that I think had not been mentioned?

At least for the discussion part, one thing I would like to see mentioned is the direction of trying to drive or control biological neurons using analyses of artificial deep neural networks, e.g., see [Neural population control via deep image synthesis | Science](https://www.science.org/doi/10.1126/science.aav9436). I don't think this was mentioned, if I missed it, feel free to correct me. Due to the already fairly lengthy manuscript, feel free to keep this to a few sentences.

If you can have a cleaner version of the top part of Figure 10, would be nice.

This paper is duplicated in the references:
Martin Schrimpf, Jonas Kubilius, Ha Hong, Najib J Majaj, Rishi Rajalingham, Elias B Issa, Kohitij Kar,
Pouya Bashivan, Jonathan Prescott-Roy, Franziska Geiger, et al. Brain-score: Which artificial neural
network for object recognition is most brain-like? BioRxiv, pp. 407007, 2018

**Strengths And Weaknesses:**

I found the manuscript well-written and comprehensive, if a bit longwinded at some points. I especially liked the Key Takeaways sections.

Some of the results sections like "Disentangling Syntax and Semantics" may benefit from a bit more shortening, potentially moving some parts to a supplementary.

I found some parts on MEG preprocessing imprecise.
> Addressing these challenges requires continuous advancements in preprocessing algorithms such as fMRIPrep (Esteban et al., 2019) and MNE (Gramfort et al., 2013),

MNE is a software library for MNE data analysis, not a preprocessing algorithm.
Also the MEG preprocessing pipeline on page 21 should be more clearly described as *an* example, choices like a bandpass filter between 0.5 and 30 Hz can and will certainly be done differently by other researchers. Also writing "using MNE-Python (Gramfort et al.,2013) default parameters"  seemed confusing to me, so do you mean there is some MNE default MEG preprocessing pipeline? If so you should like to that specific code pipeline somewhere. but seems unlikely to me that sucha  pipeline would include things like temporally decimating by exactly 10x, this does not make sense as a general default for data with maybe different sampling rates etc.

One thing not mentioned (or I missed) was the direction of trying to drive control biological neurons, see requested changes.

Many figures look quite appealing, even if a bit diverse in visual style throughout the manuscript, but probably hard to avoid when integrating a large number of figures from the field. The top part of Figure 10 I found a little less appealing due to the distorted/strangely scaled font etc.

Overall, I think the manuscript presents a valuable introduction and overview to the field of using DNNs to understand brain activity on natural stimuli.

---

> ### Author Response · Authors · 2024-10-31
>
> *We thank the reviewer for their strong positive, insightful and valuable comments and suggestions which are crucial for further strengthening our manuscript.*
>
> **Q1. Please revise the MEG preprocessing sections as described above.**
>
> Thank you for your valuable suggestion.
> * We have now updated the MEG preprocessing steps to provide a clear and detailed explanation that is applicable to any MEG dataset.
>
> **Q2. Audiocite.net: A Large Spoken Read Dataset in French - 2024_Felice_audiocite.pdf would be a dataset that I think had not been mentioned?**
>
> Thank you for suggesting this dataset.
> * We clarify that the naturalistic brain datasets discussed only include brain recordings paired with associated naturalistic stimuli. Since the Audiocite corpus does not have associated brain data, we have included this dataset in our future studies.
> * Specifically, based on reviewer's suggestion, we have included this dataset in the study titled "Bridging the Language Gap in Brain-NLP Research: The Need for Multilingual Exploration", as collecting brain recordings from this French audiobook offers a promising direction for exploring multilingual language processing in the brain.
>
> **Q3.  I would like to see the direction of neuronal population control using analyses of artificial deep neural networks.**
>
> Thank you for this suggestion. Based on the reviewer's recommendation, we have now included it in the future directions section on page 44, titled "Neural Population Control: The Next Frontier".
>
> While significant progress has been made in encoding and decoding brain signals, a promising complementary direction involves controlling or stimulating neural activity to influence perception and behavior. This bidirectional brain-computer interface approach could enable not only reading but also writing information to the brain, with current methods ranging from optogenetics for precise control of neural circuits *[Deisseroth, 2011]* to electrical stimulation through neural prosthetics *[Basumatary & Hazarika, 2020]*, and non-invasive techniques like focused ultrasound and transcranial magnetic stimulation *[Polanía et al., 2018]*. These approaches show particular promise in visual applications, such as restoring sight in blind individuals through direct stimulation of the visual cortex *[Beauchamp et al., 2020]* or enhancing visual perception through targeted neural modulation *[Romeiet al., 2010]*. However, several technical challenges persist: the precise targeting of specific neural populations remains difficult, potential tissue damage from invasive methods raises concerns, and our incomplete understanding of neural codes complicates the determination of effective stimulation patterns. Despite these challenges, this direction naturally extends the decoding work discussed in this survey and could eventually lead to fully bidirectional brain-computer interfaces that can both interpret and influence neural activity in real-time *[Saha et al., 2021]*. Such technology holds profound implications for treating neurological conditions, enhancing human capabilities, and advancing our understanding of brain function, provided that future research successfully addresses the needs for more precise and less invasive stimulation methods while carefully considering the ethical implications of direct neural control.
>
> *[Deisseroth, 2011] Optogenetics, Nature Methods 2011.*
>
> *[Basumatary & Hazarika, 2020] State of the art in bionic hands, IEEE THMS 2020.*
>
> *[Polanía et al., 2018] Studying and modifying brain function with non-invasive brain stimulation, Nature Neuroscience 2018.*
>
> *[Beauchamp et al., 2020] Dynamic stimulation of visual cortex produces form vision in sighted and blind humans, Cell 2020.*
>
> *[Romeiet al., 2010] On the role of prestimulus alpha rhythms over occipito-parietal areas in visual input regulation: correlation or causation? Journal of Neuroscience 2010.*
>
> *[Saha et al., 2021] Progress in brain computer interface: Challenges and opportunities, Frontiers in systems neuroscience 2021.*
>
> **Q4. If you can have a cleaner version of the top part of Figure 10, would be nice.**
>
> Thank you for pointing this out. We have updated Figure 10 with a cleaner version in the revised draft.
>
> **Q5. This paper is duplicated in the references.**
>
> * We have corrected this typo.

---

> > ### Comment · Reviewer_HXgw · 2024-11-04
> > **Thanks, one remark**
> >
> > Thank you so much for your changes. The MEG preprocessing definitely seems much more general to me now.
> >
> > Thanks for the audiocite comment, I somehow completely missed it does not have brain data, was an oversight on my part.
> >
> > Regarding Q3, maybe I had not been clear enough, I was thinking more in the direction of the paper I cited: Using findings from deep-learning-based analyses to develop a method (e.g. show particular highly activating images) to better *understand* brain responses and similarity between brain activities and deep network activities, so using the stimulation of brain activity not as an end but as a means for understanding. But what you wrote is ccertainly also very interesting and you do mention " and advancing our understanding of brain function" even though you placed the emphasis differently. I would still prefer to also add a citation to  [Neural population control via deep image synthesis | Science](https://www.science.org/doi/10.1126/science.aav9436) as I think it is a very excellent paper that deserves mention in this direction.
> >
> > Thanks for Q4/Q5 improvements and corrections, much appreciated! :)

---

> > > ### Author Response · Authors · 2024-11-05
> > >
> > > *We appreciate the reviewer's feedback and are confident it has enhanced the quality of our paper.*
> > >
> > > Thank you for your valuable suggestion. We have now included a brief description and citation [Bashivan et al. 2019] for the suggested paper under the section ''Neural Population Control: The Next Frontier.''
> > >
> > > [Bashivan et al. 2019] Neural population control via deep image synthesis, Science 2019.

---

> > > > ### Comment · Reviewer_HXgw · 2024-11-05
> > > > **Thanks**
> > > >
> > > > Thanks for the change, I have no further requests at this point.

---

### Author Response · Authors · 2024-10-14

Dear Reviewers,

We are grateful to all the reviewers for their time, strong positive feedback, and constructive suggestions, which will further strengthen the impact of our work.

We have uploaded a revised version of the draft, trying to incorporate all these comments. The revised places are colored in blue. Feel free to let us know if there is anything we could do to further improve the draft.

Thank you.

---

### Decision · Action_Editor_1jpM · 2024-12-06

**Recommendation:** Accept as is

**Comment:**

This submission surveys literature regarding the measurement of alignment between deep neural networks and the brain. The reviewers pointed out several errors in the survey, which the authors addressed during the rebuttal period. In addition, Reviewer tXDt noted that the survey's scope is somewhat restricted compared to the field as a whole. The authors addressed these concerns by correcting the errors identified by the reviewers and making the scope of the paper clearer, e.g. by mentioning in the abstract that the paper focuses on human brain recording studies.

The overview of human neuroimaging studies of brain-machine alignment in the context of language, speech, and vision that this submission provides seems useful; I recommend acceptance. However, readers new to brain-machine alignment may still find it difficult to discern what relevant literature on human-machine alignment falls outside the paper's specific scope. The review is already quite long, so I hesitate to ask the authors to further expand it, but it might be helpful to these readers to slightly expand the paragraph at the start of p. 5 that delineates the paper's scope to provide a slightly more comprehensive (but still one paragraph) high-level description of research topics in brain-machine alignment in non-human subjects, with citations to other review articles that focus primarily on this work.

**Audience:**

Yes.

**Claims And Evidence:**

Yes.

---

> ### Author Response · Authors · 2024-12-09
>
> We are deeply grateful to all the reviewers and the AE for their insightful feedback and support. We greatly appreciate the effort invested in reviewing our work. Finally, we are deeply honored by the proposed survey certification!
>
> Regarding the suggestion to provide a broader high-level description of brain-machine alignment research beyond human studies, we agree this would enhance clarity for readers new to the topic. In the camera-ready version, we will expand the paragraph on p. 5 to include a concise overview of brain-machine alignment research in non-human subjects, along with references to relevant review articles.
>
> Thank you once again for the valuable feedback and the recommendation for acceptance!
>
> Best regards,
> The authors

---

> > ### Author Response · Authors · 2024-12-29
> > **Final Camera-Ready Submission**
> >
> > Dear AE and Reviewers,
> >
> > Thanks so much for your encouragement and suggestions during the paper reviewing period. Based on the AE's suggestion, we have included a paragraph in the final camera-ready version that provides a high-level description of research topics related to brain-machine alignment in non-human subjects. Currently, we have submitted a camera-ready version to the system.
> >
> > Thank you once again for the valuable feedback and the recommendation for acceptance!
> >
> > Best regards,
> >
> > The authors